# T cell memory response to MPXV infection exhibits greater effector function and migratory potential compared to MVA-BN vaccination

Ji-Li Chen[1,2,72], Beibei Wang [1,72], Yongxu Lu [1,3,72], Elie Antoun[1,4,72], Olivia Bird [5,72], Philip G. Drennan[5,6,72], Zixi Yin[1,72], Guihai Liu[1], Xuan Yao[1], Maya Pidoux[1,2], Adam Bates [1,2], Deshni Jayathilaka[1,2], Junyuan Wang[1,2], Brian Angus [7], Sally Beer [5], Alexis Espinosa[5], J. Kenneth Baillie [8,9,10,11], Malcolm G. Semple [12], ISARIC4C Investigators*, Timothy Rostron[13], Craig Waugh[14], Paul Sopp[14], Julian C. Knight [1,4], James N. Fullerton [5,15], Mark Coles [6], Geoffrey L. Smith [1,2,3,73], Alexander J. Mentzer [1,4,73], Yanchun Peng[1,2,73] & Tao Dong [1,2,73] ✉

In 2022, a global mpox outbreak occurred, and remains a concern today. The T cell memory response to MPXV (monkeypox virus) infection has not been fully investigated. In this study, we evaluate this response in convalescent and MVA-BN (Modified Vaccinia Ankara - Bavarian Nordic) vaccinated individuals using VACV-infected cells. Strong CD8$^+$ and CD4$^+$ T cell responses are observed, and T cell responses are biased towards viral early expressed proteins. We identify seven immunodominant HLA-A*02:01 restricted MPXV-specific epitopes and focus our detailed phenotypic and scRNAseq analysis on the immunodominant HLA-A*02:01-G5R$_{18-26}$-specific CD8$^+$ T cell response. While tetramer$^+$CD8$^+$ T cells share similar differentiation and activation phenotypes, T cells from convalescent individuals show greater cytotoxicity, migratory potential to site of infection and TCR clonal expansion. Our data suggest that effective functional profiles of MPXV-specific memory T cells induced by Mpox infection may have an implication on the long-term protective responses to future infection.

On 23 July 2022, the World Health Organisation (WHO) declared the outbreak of monkeypox virus (MPXV) infections a Public Health Emergency of International Concern (PHEIC), following the identification and spread in multiple countries outside of the natural MPXV reservoirs in Africa[1]. The disease mpox is caused by MPXV, an enveloped double-stranded DNA virus of the *Orthopoxvirus* genus in the *Poxviridae*, which includes variola, cowpox, vaccinia, and other viruses[2]. There are two clades of MPXV that have different geographic origins and different case fatality rates (CFR) in humans. Clade I viruses originate from central Africa and have a CFR of 5–8%, whereas Clade II viruses originate from West Africa and have a CFR of <0.2%. Fortunately, the 2022-4 global epidemic has been caused by clade IIb viruses and, as of 29 March 2024, resulted in nearly 95,000 confirmed cases with 178 deaths (CFR 0.19%). Although there has been an overall

A full list of affiliations appears at the end of the paper. *A list of authors and their affiliations appears at the end of the paper.
✉e-mail: tao.dong@ndm.ox.ac.uk

decline in cases globally from the peak reached in 2022, MPXV transmission continues in many nations. Moreover, since June 2023, the WHO noted increased mpox cases in China and the Western Pacific and South-East Asian regions, as well as in Europe and the Americas[3,4]. Of greater concern has been a large outbreak of clade I MPXV infections in the Democratic Republic of Congo (DRC) from 2023 onwards[5], which has killed >1000 people and spread to other nations. Given smallpox vaccination ceased in most nations during the 1970s, as smallpox was controlled and then eradicated, a large proportion of the population has little or no immunity to MPXV or variola virus, the cause of smallpox. Furthermore, severe MPXV disease is more common in very young children, pregnant women, or those with compromised immune systems[6,7].

Although MPXV was identified over 50 years ago[8], human T cell immunity to MPXV infection has not been characterized extensively, with only a few studies demonstrating a T cell response against MPXV in humans during the 2003 US mpox outbreak[9-12]. More recently, epitope pools based on experimentally defined CD4 and CD8 epitopes from The Immune Epitope Database (IEDB) and predicted MPXV epitopes have been used to investigate T cell responses in MPXV-infected donors using the activation-induced markers assay and intracellular cytokine staining[12-14].

Early during infection, besides effector memory CD8+ T cells, the predominant population of CD8+ T cells in MPXV-infected individuals are $T_{EMRA}$ cells: terminally differentiated effector memory T cells re-expressing CD45RA, which also express PD1 and CD57 in the post-acute phase[11]. Furthermore, MPXV-convalescent patients generate a robust T-cell response[12].

The Modified vaccinia Ankara-Bavarian Nordic (MVA-BN) vaccine is a third-generation smallpox vaccine authorized for use against MPXV, previously shown to induce MPXV-neutralizing antibodies in healthy individuals[15]. MVA-BN vaccination has shown weak antibody and neutralization responses against MPXV. However, a robust CD4+ and CD8+ T cell response is evident[14,16]. Mazzotta et al. [16] show an increase in the T cell response following MVA-BN vaccination in both individuals with and without a previous smallpox vaccination, with the vaccination able to reduce Mpox infection[17]. Moreover, Collier et al. [18] have shown that antibody responses against MPXV after MVA-BN vaccination begin to wane 6-12 months after vaccination, highlighting the importance of investigating the T cell response following both vaccination and infection. This has led to MVA-BN being used extensively as part of the response to the mpox epidemic. However little evidence exists comparing the T-cell response elicited by MVA-BN to natural infection.

Given that there is an 84% nucleotide sequence identity between MPXV from the 2022 outbreak and vaccinia virus (VACV)[19], it has been demonstrated that MPXV-induced T cells are largely cross-reactive with the vast majority of VACV epitopes[13]. Griffoni et al. [13] were able to show that MPXV-derived predicted epitope pools were able to induce T-cell responses in individuals vaccinated against smallpox. However, due to the large viral genome sizes and highly variable HLA backgrounds[19,20], it can be difficult to effectively evaluate overall T cell responses.

In this study, we develop a T cell assay to evaluate ex vivo T cell responses to VACV-infected cells which allows us to evaluate the overall MPXV T cell immunity and T cell responses to naturally processed and presented epitopes in infected cells. We further validate our results using MPXV mega pools consisting of MPXV-specific peptides to known VACV peptide epitopes identified from IEDB with identical sequences to the MPXV genome, and via detailed single-cell RNAseq analysis on an immunodominant HLA-A*02:01-restricted epitope.

## Results
### Study participants
**Convalescent cohort.** A total of 13 individuals in the UK were recruited using a pre-positioned Urgent Public Health Research protocol

(ISARIC) following recovery from MPXV infection in August 2022. All participants were male, aged 23-60 years old. 3/13 had co-morbidities, including two with well-controlled HIV infection. The infection status of all participants was confirmed by a positive MPXV PCR test. 11 out of 13 presented with skin lesions, while 5 out of 13 also reported fever. Six participants reported bleeding from either their rectum (proctitis) or from the vesicles, with one participant (on a vitamin K antagonist for anticoagulation) requiring a transfusion of red cells. In addition, 10 control individuals naïve to MPXV infection were studied in parallel. Participant characteristics are summarized in Table 1 and their HLA allele coverage is shown in Supplementary Fig. 1 and Supplementary Data Table 1.

**Vaccination cohort.** A further 20 male individuals aged between 25 and 72 were recruited following vaccination with the MVA-BN vaccine in Oxford[21]. All 20 individuals recruited were HIV negative and were naïve to MPXV infection; however, two of these individuals had a prior smallpox vaccination pre-1980 and all of these were seropositive at baseline. One individual had a history of type II diabetes. Further information on the vaccinated cohort can be found by Drennan et al. [21], with a detailed description of this cohort and time of sample collection after vaccination dose can be found in Supplementary Data Table 2.

### Strong overall memory T cell responses specific to MPXV detected in convalescent individuals
Fresh PBMCs were isolated from 13 participants who had recovered from an MPXV infection and 10 healthy controls. A schematic of the study protocol is shown in Fig. 1A. In order to evaluate the overall MPXV memory T cell response, we set up an ex vivo IFNγ ELISpot assay by stimulating PBMCs with the VACV strain Lister as previously described[22,23]. VACV-reactive T cells were detected as IFNγ-producing cells by ELISpot (Fig. 1B). Individuals who recovered from MPXV infection were able to mount a significant antiviral T cell response compared with the healthy controls (Fig. 1C, $p < 0.001$), while the frequency of influenza, Epstein-Barr virus (EBV) and human cytomegalovirus (HCMV) (FEC)-specific T cells responses in the two groups were comparable (Supplementary Fig. 2A).

In parallel with the ELISpot assay, an activation-induced markers (AIMs) assay[24] was carried out with a few modifications (see Methods) to quantify the overall VACV-reactive CD8+ and CD4+ T cells. Activation of CD8+ T cells was marked by the upregulation of CD69 and 4-1BB (CD137), while OX40 and CD40L (CD154) were upregulated on activated CD4+ T cells (Fig.1D). Two participants were excluded from this analysis: Mpox009 due to insufficient sample availability, and Mpox011 due to high background levels in the assay. In 11 MPXV-convalescent individuals, there was a similar level of response in the magnitude of CD8+ VACV-reactive T cells compared to CD4+ VACV-reactive T cells (Fig. 1E). The relative proportion of the CD3+ T cell response attributable to VACV-reactive CD8+ and CD4+ were calculated for each individual. As shown in Fig. 1F, the proportion of CD4+/CD8+ reactive T cells varied between participants. While Mpox008 showed only a CD4+ T cell response and Mpox005 showed only a CD8+ T cell response, all other individuals showed a varied proportion of the T cell response elicited by CD4+ and CD8+ T cells; however, only a marginal difference was observed overall, with an average proportion of 42.5% CD8+-reactive T cells compared to 57.5% of CD4+-reactive T cells in this cohort.

### T cell responses against MPXV-specific epitope mega pool and early antigens
Having assessed the overall T cell response using the VACV system, we next validated the MPXV T cell response against experimentally defined poxvirus epitopes from IEDB and the published literature (see Methods for details)[25]. 232 CD4 human epitopes from 90 ORFs and 273 CD8 human epitopes covering 164 ORFs were included and synthesized (Supplementary Data Table 3). Alignment of the sequences of the

**Table 1 | Clinical characteristics of Mpox convalescent participants and HC**

| Study ID | Sex | Age | HIV status | Co-morbidities | Orthopoxvirus exposure | Severity | Sample collection (days post symptom onset) | Fever | Skin Rash with vesicles |
|---|---|---|---|---|---|---|---|---|---|
| Mpox001 | M | 21–30 | Uninfected | None | mpox infection | Mild | 75 | Yes | Yes |
| Mpox002 | M | 31–40 | Uninfected | None | mpox infection | Moderate/severe | 105 | Yes | Yes |
| Mpox003 | M | 31–40 | Uninfected | None | mpox infection | Moderate/severe | 105 | Yes | Yes |
| Mpox004 | M | 31–40 | Uninfected | None | mpox infection | Moderate/severe | 105 | Yes | Yes |
| Mpox005 | M | 31–40 | Uninfected | None | mpox infection | Mild | 75 | Yes | Yes |
| Mpox006 | M | 51–60 | Uninfected | None | mpox infection | Mild | 60 | No | Yes |
| Mpox007 | M | 31–40 | Infected and controlled | Ischaemic Heart Disease | mpox infection | Mild | 120 | No | Yes |
| Mpox008 | M | 31–40 | Uninfected | None | mpox infection | Mild | 120 | Yes | No |
| Mpox009 | M | 41–50 | Infected and controlled | Anti-phospholipid Syndrome Diabetes mellitus | mpox infection | Moderate/severe | 75 | Yes | No |
| Mpox010 | M | 51–60 | Uninfected | None | mpox infection | Mild | 90 | No | Yes |
| Mpox011 | M | 51–60 | Uninfected | None | mpox infection | Mild | 90 | No | Yes |
| Mpox012 | M | 31–40 | Uninfected | Crohn's disease (not immunosuppressed) | mpox infection | Mild | 120 | Yes | Yes |
| Mpox013 | M | 31–40 | Uninfected | None | mpox infection | Mild | 120 | No | Yes |
| HC 001 | F | 41–50 | | | vaccinia-naïve | | | | |
| HC 002 | F | 31–40 | | | vaccinia-naïve | | | | |
| HC 003 | F | 31–40 | | | vaccinia-naïve | | | | |
| HC 004 | F | 21–30 | | | vaccinia-naïve | | | | |
| HC 005 | F | 51–60 | | | vaccinia-naïve | | | | |
| HC 006 | M | 31–40 | | | vaccinia-naïve | | | | |
| HC 007 | F | 31–40 | | | vaccinia-naïve | | | | |
| HC 008 | M | 31–40 | | | vaccinia-naïve | | | | |
| HC 009 | M | 31–40 | | | vaccinia-naïve | | | | |
| HC 010 | F | 21–30 | | | vaccinia-naïve | | | | |

chosen epitopes between VACV strain Lister and a clade IIb MPXV strain isolated in Glasgow in 2022 showed that the majority of CD8 epitopes (78%) and CD4 epitopes (72%) were conserved between VACV and MPXV. The remaining 22% of CD8 and 28% of CD4 epitopes corresponded to VACV epitopes that contained non-synonymous mutations in the MPXV genome at the amino acid level. Based on the stage of protein expression[26,27] (early vs. intermediate/late), we divided the MPXV CD4 mega pool into two smaller pools: an early epitope pool and an intermediate/late epitope pool; whereas we divided the MPXV CD8 mega pool into four smaller pools: MPXV-CD8 HLA-A*02 restricted early epitope pool and intermediate/late epitope pool, and Non-HLA-A*02 restricted early epitope pool and intermediate/late epitope pool as shown in Fig. 2A.

Memory MPXV-specific T cell responses in MPXV-convalescent individuals were then assessed by ex vivo IFNγ ELISpot with thawed PBMCs and responses against early epitopes and intermediate/late epitopes were compared (Fig. 2B). Overall, there was a significant greater CD8+ T cell response to proteins expressed early compared to intermediate/late ones, consistent with earlier studies[25]. In contrast, we see minimal CD4+ T cell responses with only two individuals showing a positive response, significantly lower than the CD8+ T cell response ($p = 0.005$), with no difference between the early and intermediate/late epitope pools. Seven out of 12 (58%) MPXV-convalescent individuals showed positive CD8 T cell response to early antigens, whereas only 4 out of 12 (33%) participants responded to MPXV CD8 epitopes from intermediate/late antigens.

Given that we showed a greater response to early epitopes compared to intermediate/late epitopes, we next synthesized overlapping peptides spanning five early proteins of MPXV, including A9, F3, E12, D10, and D1, which have been reported to be the most immunogenic after orthopoxvirus infection[28–30]. Cell responses against these antigens were assessed by ex vivo IFNγ ELISpot (Fig. 2C). Seven out of twelve donors recognized at least one of the antigens tested here. F3 and D1 exhibited the greatest immunogenicity, with 5 participants responding to F3 and 6 participants responding to D1, ranging from 50 to 245 SFU/10⁶ PBMC, while E12 exhibited the least immunogenicity, with recognition by only 4 out of 12 donors tested. These results suggest that these immunogenic antigens could be potential candidate antigens for new poxvirus subunit vaccines in the future.

Finally, we compared the magnitude of T-cell responses detected by using VACV as an antigen and the response against the MPXV CD8 and CD4 epitope mega pool. As shown in Fig. 2D, E, there was no significant difference in the overall magnitude of the response detected by these two methods. However, we observed that 12 of 13 individuals exhibited a positive VACV reactive T cell response, whereas only 7 of 12 showed a positive response to the peptide mega pools. Three individuals (Mpox-001, 005, and 012) showed stronger T cell response detected by epitope mega pools when compared to VACV. These T cell responses were mainly elicited by CD8+ T cells (Fig. 2E) and in particular, HLA-A2 epitopes (Supplementary Fig. 2B), where HLA-A2 epitopes accounted for >50% of the overall T cell response.

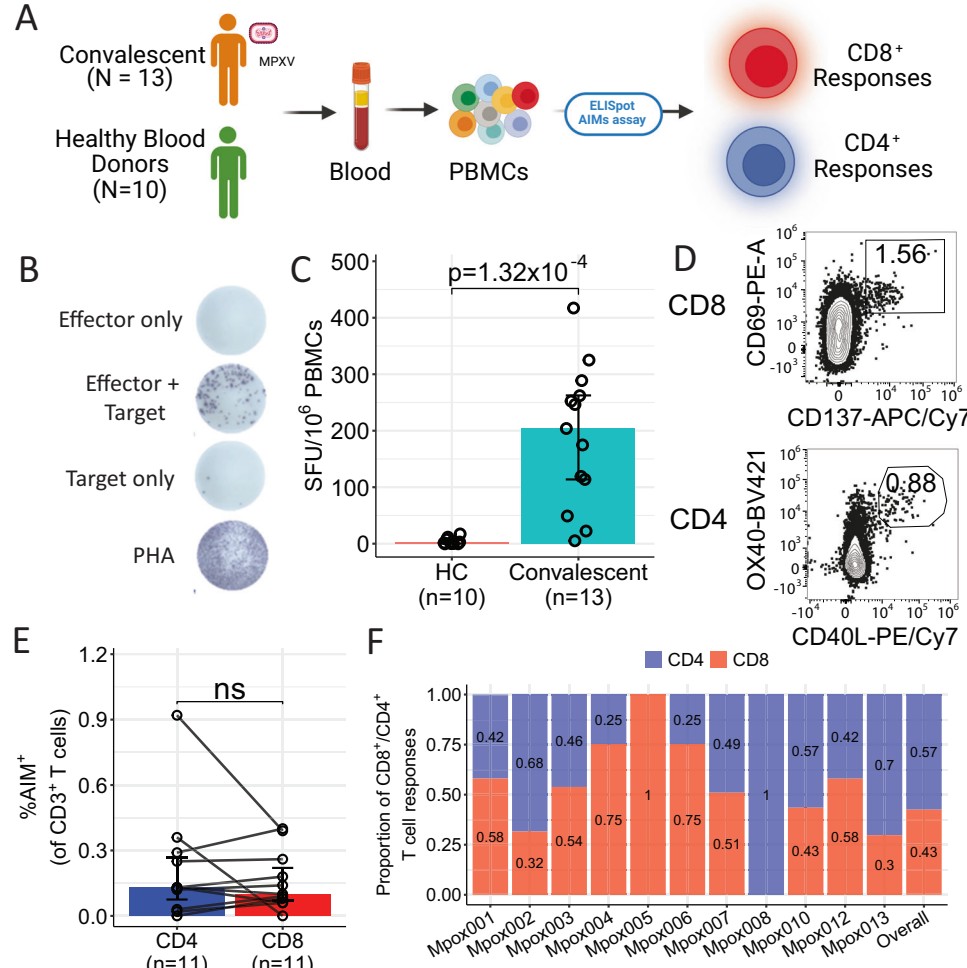

**Fig. 1 | MPXV-specific memory T cell response in mpox convalescent individuals and healthy controls (HC). A** Study design (Created in BioRender. Antoun, E. (2025) https://BioRender.com/a25q267). **B** Representative IFN-γ ELISpot response after stimulation with VACV. **C** Summary of VACV-induced memory T cell response in mpox convalescent (*N* = 13) and healthy control (HC, *N* = 10) participants. Data are presented as median ± IQR. **D** Representative flow cytometry plots measuring expression of activation-induced markers on VACV-reactive CD8⁺ and CD4⁺ T cells from mpox convalescent participants. **E** Overall percentage of VACV-reactive CD8⁺ and CD4⁺ T cells in total T cells. Data are presented as median ± IQR (*N* = 11). **F** Individual and overall relative proportion of VACV-reactive CD8⁺ (red) and CD4⁺ (blue) T cells, *N* = 11. The Mann−Whitney *U*-test was used for the analysis and two-tailed *P* values were calculated. ns not significant; IQR interquartile range. Source data are provided as a Source Data file.

## Identification of immunodominant HLA-A*02:01-restricted MPXV-specific T cell epitopes and functionality

Having identified that the strong overall CD8 T cell responses were heavily biased towards early-expressed antigens targeting HLA-A2-restricted epitopes, we next extended our studies to examine the HLA-A*02:01-restricted CD8 T cell response, given the fact that HLA-A*02 is the common Class I HLA allele in different populations around the world[31] and presents several immunodominant viral epitopes[32,33]. T cell responses to a pool of 71 HLA-A*02-restricted early epitopes and a pool of 68 HLA-A*02 intermediate/late epitopes were evaluated using an ex vivo ELISpot assay in six HLA-A*02:01⁺ convalescent individuals. Unsurprisingly, we observed 5 out of 6 donors with a strong memory A*02 CD8 T cell response against early antigens, whereas only 2 out of 6 displayed a response against intermediate/late expression antigens (Fig. 3A).

Next, we generated short-term T cell lines from four convalescent HLA-A*02:01⁺ individuals by stimulating PBMCs with an A*02-specific peptide pool of the early expressed antigens. The antigen-specificity of these short-term T cell lines was screened by individual peptides using the ELISpot assay. Seven epitopes were identified, as shown in Fig. 3B.

The sequences of these epitopes and name of the antigen identified were as follows: HLA-A*02:01-E1₈₃₃₋₈₄₁ (LLSYYVVYV, LLS), HLA-A*02:01-G5₁₈₋₂₆ (ILDDNLYKV, ILD)[23], HLA-A*02:01-B7₁₁₃₋₁₂₁ (IMYDIINSV, IMY), HLA-A*02:01-E1₄₄₅₋₄₅₃ (VLYNGVNYL, VLY), HLA-A*02:01-G5₂₂₉₋₂₃₇ (YLAKLTALV, YLA), HLA-A*02:01-E5₇₂₆₋₇₃₄ (ILYDNVVTL, ILY) and HLA-A*02:01-C17₃₄₆₋₃₅₃ (SLSNLDFRL, SLS). Of these epitopes, six are conserved between VACV and MPXV. HLA-A*02:01-B7₁₁₃₋₁₂₁ (IMYDIINSV, IMY) contains an L to I substitution at the start of the epitope. Among them, epitope ILD appeared as the greatest immunodominant epitope, with all four cell lines tested showing a detectable response to this epitope.

T cells specific to three epitopes (ILY, ILD, and SLS) were confirmed by HLA-A*02:01 tetramer staining (Fig. 3C). Six CD8⁺ bulk cell lines specific to either ILY, ILD, or SLS were then individually enriched by tetramers, and the functionality of these antigen-specific T cells was evaluated by intracellular cytokine staining (ICS) after co-culturing T cells with VACV-infected EBV-transformed B cell lines (BCLs) (Fig. 3D). All the bulk CD8⁺ T cell lines showed responses to VACV-infected BCLs, expressing cytokines (IFNγ, TNFα, MIP1β, and IL2) and the degranulation marker CD107a (Fig. 3D and E), with >50% of cells

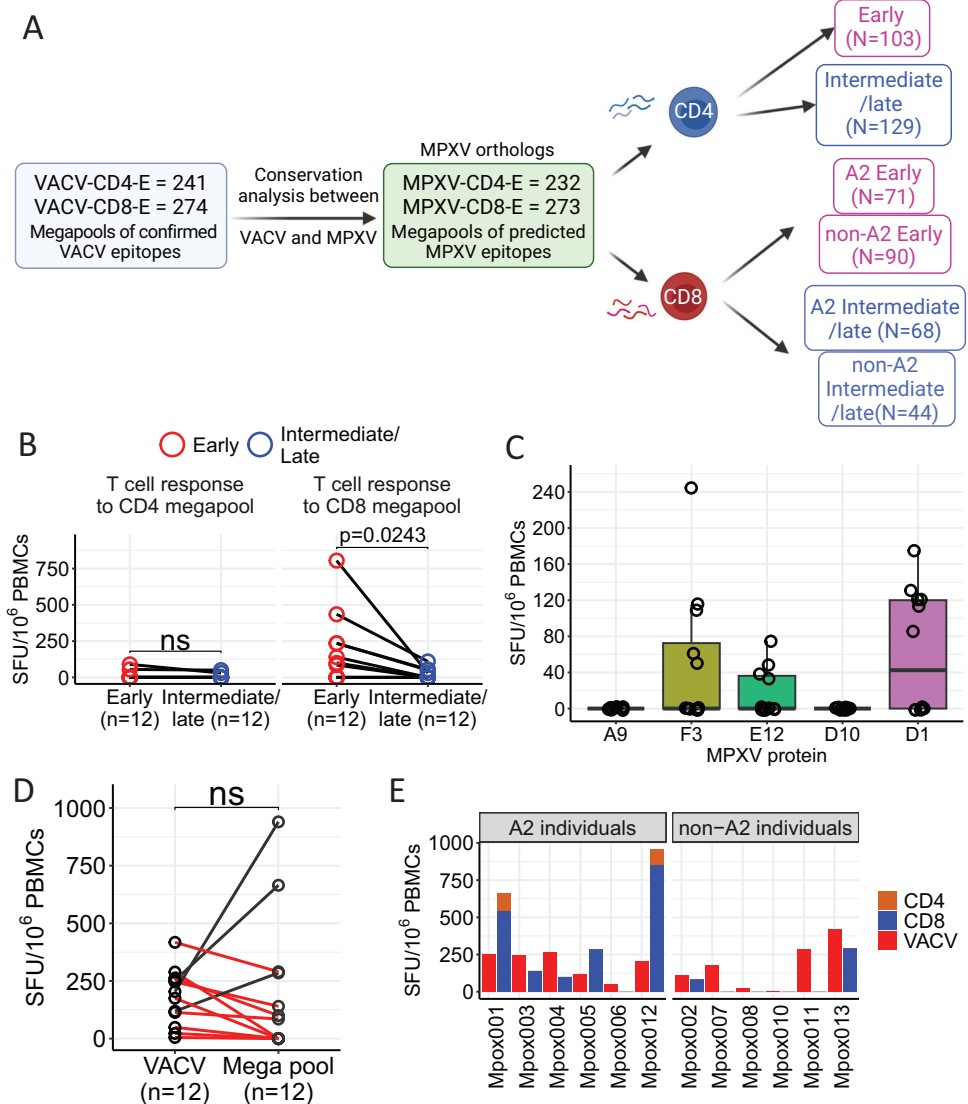

**Fig. 2 | Memory T cell responses in MPXV-convalescent individuals against epitope mega pools of early and intermediate/late antigens. A** Schematic graph of MPXV epitope mega pool design (see the "Methods" section for more details, Created in BioRender. Antoun, E. (2025) https://BioRender.com/a25q267). **B** T cell responses against MPXV CD4+/CD8+ early proteins (red) or intermediate/late proteins (blue), $N = 12$. **C** Overall T cell responses against overlapping peptide pools covering the full length of five selected MPXV antigens, $N = 12$. Boxplots represent the 25th and 75th percentiles with the median marked with whiskers at ±1.5*IQR. **D, E** Comparison of overall T cell response against VACV and that against MPXV mega pools ($N = 12$). A paired Wilcoxon signed-rank test for **B** and **D** was used and two-tailed $P$ values were calculated. ns not significant, SFU spot forming units, MPXV Mpox virus, VACV vaccinia virus. Source data are provided as a Source Data file.

showing polyfunctionality, expressing multiple of the investigated effector molecules (Fig. 3E). Some cell lines exhibited a strongly dominant TNFα response compared to the IFNγ response. Further investigation is needed to determine the factors contributing to this, including whether this is influenced by antigen load on infected cells, the functional avidity of the T cells, or other factors that may be at play. Moreover, these CD8+ epitope-specific bulk T cell lines could kill VACV-infected target cells (Fig. 3F), indicating their important role in controlling virus infection.

**Similar MPXV-specific T cell frequency and phenotype in convalescent and vaccinated individuals**

Given that MVA-BN is currently being administered to protect against MPVX infection, we next used a second cohort of individuals[21] vaccinated with one or two doses of MVA-BN vaccines to compare the T cell response between convalescence and vaccination. Samples were collected 28 days after the first dose of the MVA-BN vaccine and

28 days after the second dose (Fig. 4A, Supplementary Data Table 2). Overall T cell response was evaluated with ex vivo IFNγ ELISpot assay by stimulating PBMCs with the VACV strain Lister. Both individuals who recovered from MPXV infection and vaccinated individuals were able to mount a significant antiviral T cell response (Fig. 4B). We found no statistically significant difference in the T cell response between convalescent and vaccinated individuals ($p > 0.05$); however, there is a slightly stronger T cell response in convalescent individuals compared to those 28 days after dose 1 and dose 2 of the vaccine.

We next carried out ex vivo phenotypical analysis of antigen-specific T cells. Given the evidence of a high frequency of A*02:01-restricted ILD-specific T cells being detected amongst the HLA-A*02:01+ participants, we analysed the frequency, phenotype, and differentiation status of memory MPXV-specific CD8+ T cell using the HLA-A*02:01-ILD tetramer in both convalescent samples and samples 28 after a second vaccination.

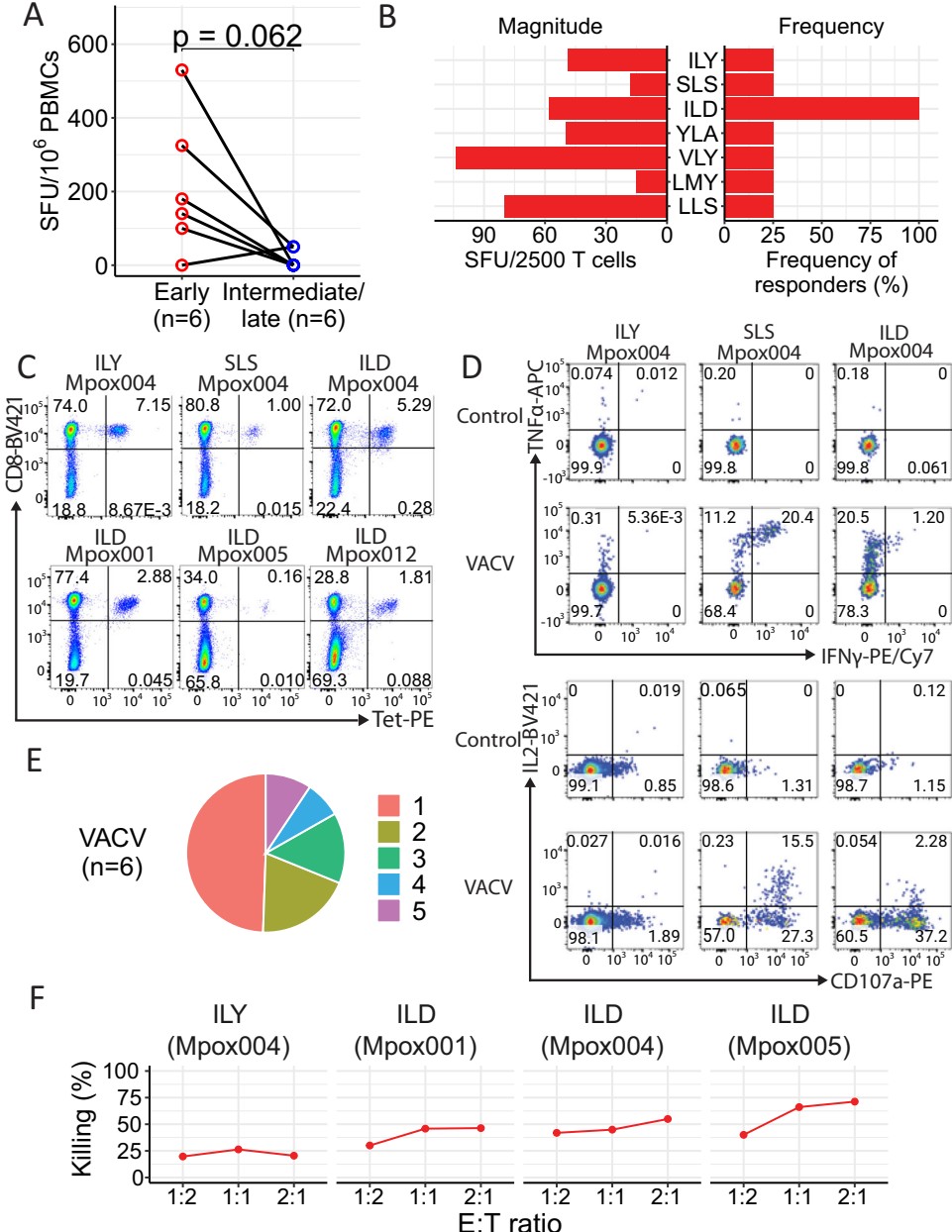

**Fig. 3 | Identification and functional evaluation of immunodominant A2-restricted MPXV-specific T cells. A** Comparison of HLA-A*02:01-restricted T cell responses against MPXV-CD8-early (red) and MPXV-CD8-intermediate/late (blue) peptide pools using ex vivo IFNγ ELISpot, $N = 6$ participants. A paired Wilcox-signed rank test was carried out to compare between groups and two-sided $p$-values calculated. **B** Identification of dominant HLA-A*02:01-restricted MPXV-specific T cell epitopes using short-term T cells lines from $N = 4$ donors. **C** Specific T cells recognizing the three epitopes ILYDNVVTL (ILY), SLSNLDFRL (SLS), and ILDDNLYKV (ILD) were confirmed by tetramers. **D** Representative cytokine responses of different HLA-A*02:01-restricted MPXV-specific T cell lines from Mpox004. **E** Proportion of T cells against VACV expressing 1, 2, 3, 4, or all 5 of the following functional markers: IFNγ, TNFα, MIP1β, IL2 and CD107a expression, $N = 6$. **F** Percentage of specific lysis of VACV-infected HLA-A*02:01+ B cell lymphoblastoid cell lines (BCLs) by different epitope-specific CD8+ bulk lines. VACV vaccinia virus, SFU spot forming units. Source data are provided as a Source Data file.

HLA-A*02:01-positive donors showed clear detectable tetramer+ staining (Fig. 4C) and the frequency of tetramer+CD8+ T cells from convalescent individuals was significantly higher than vaccinated individuals ($p = 0.032$, Fig. 4D). In convalescent samples and samples 28 days post vaccine dose 2, the majority (75%) of HLA-A*02:01-ILD tetramer-positive CD8+ T cells were CD45RA+CCR7− effector cells, with minimal levels of CD45RA+CCR7+ naïve cells (Fig. 4E). There was no significant difference in the memory phenotype of the tetramer+ T cells between convalescent individuals and after vaccination. Furthermore, the CD45RA+CCR7− effector cells expressed high levels of CD27 and minimal expression of the senescence marker CD57 (Fig. 4F), indicating an early/intermediate differentiation phenotype. Interestingly, this effector population showed heterogenous expression of KLRG1 and CX3CR1 (Fig. 4F), suggesting their potential to form different memory subpopulations, with a significantly greater level of KLRG+tetramer+CD45RA+CCR7−CD8+ T cells 28 days after a second vaccination compared to convalescence ($p = 0.03$, Fig. 4F).

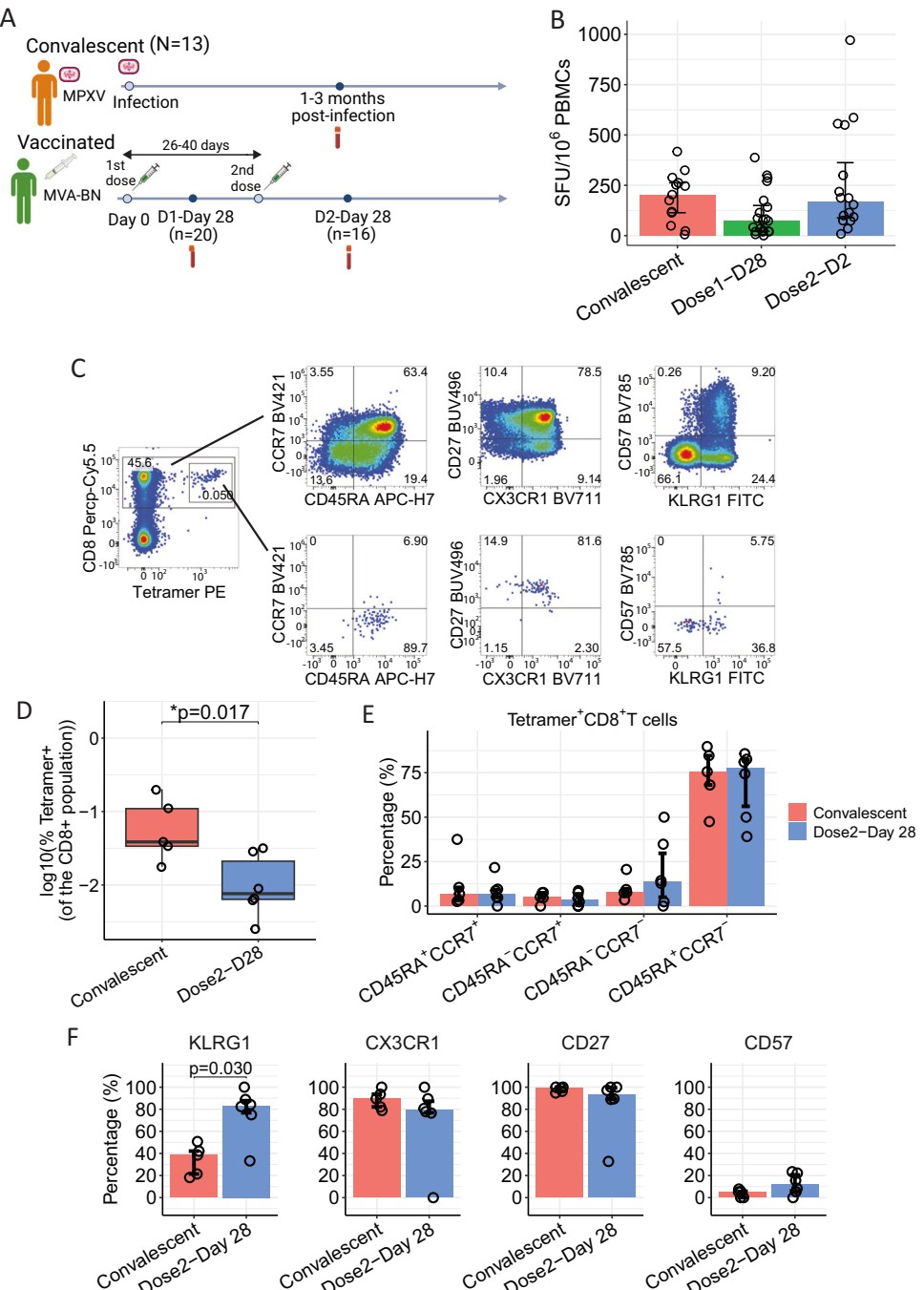

**Fig. 4 | Frequency and phenotype of ex vivo MPXV-specific T cell response in convalescent and vaccinated donors. A** Study design for MPXV-convalescent and vaccinated individuals (created in BioRender. Antoun, E. (2025) https://BioRender. com/a25q267). **B** Summary of VACV-induced memory T cell response in mpox convalescent (*N* = 13) and in individuals 28 days after 1 vaccination (*N* = 20) and 28 days after 2 vaccinations (*N* = 16). **C** Representative plot of tetramer⁺CD8⁺ and overall CD8⁺ T cells. **D** Comparison of frequency of HLA-A*02:01 ILD-specific Tetramer⁺ T cells between convalescent (*N* = 5) and vaccinated donors (*N* = 6). Boxplots represent the 25th and 75th percentiles with the median marked with

whiskers at ±1.5*IQR. **E** Percentage of different memory and differentiation phenotype marker expression on CD8⁺tetramer⁺ T cells from convalescent participants (red, *N* = 5) and samples 28 days after 2 vaccinations (blue, *N* = 6). **F** Expression of memory and differentiation markers on CD45RA⁺CCR7⁻CD8⁺tetramer⁺ T cells from convalescent participants (red, *N* = 5) and samples 28 days after dose 2 of vaccine (blue, *N* = 6). Data are presented as median±IQR. The Mann–Whitney *U*-test was used and two-tailed *P* values were calculated. SFU spot forming units. Source data are provided as a Source Data file.

## scRNAseq identifies transcriptomic and clonal differences of ex vivo CD8⁺ ILD-specific T cells between convalescent and vaccinated individuals

To further investigate these CD8⁺ MPXV-specific T cells in convalescent individuals, we carried out SmartSeq2 scRNAseq analysis to investigate the transcriptome and TCR repertoire of ILD-specific T cells from

convalescent individuals compared to samples 28 days after vaccination, the timepoint when sufficient tetramer⁺ T cells were isolated. Our dataset comprised 561 ILD-specific CD8⁺ T cells from 10 individuals, with 401 cells from convalescence and 160 cells after vaccination. Cells were integrated at a per-individual level and cells were clustered using nearest-neighbour analysis (Fig. 5A, Supplementary Fig. 5A),

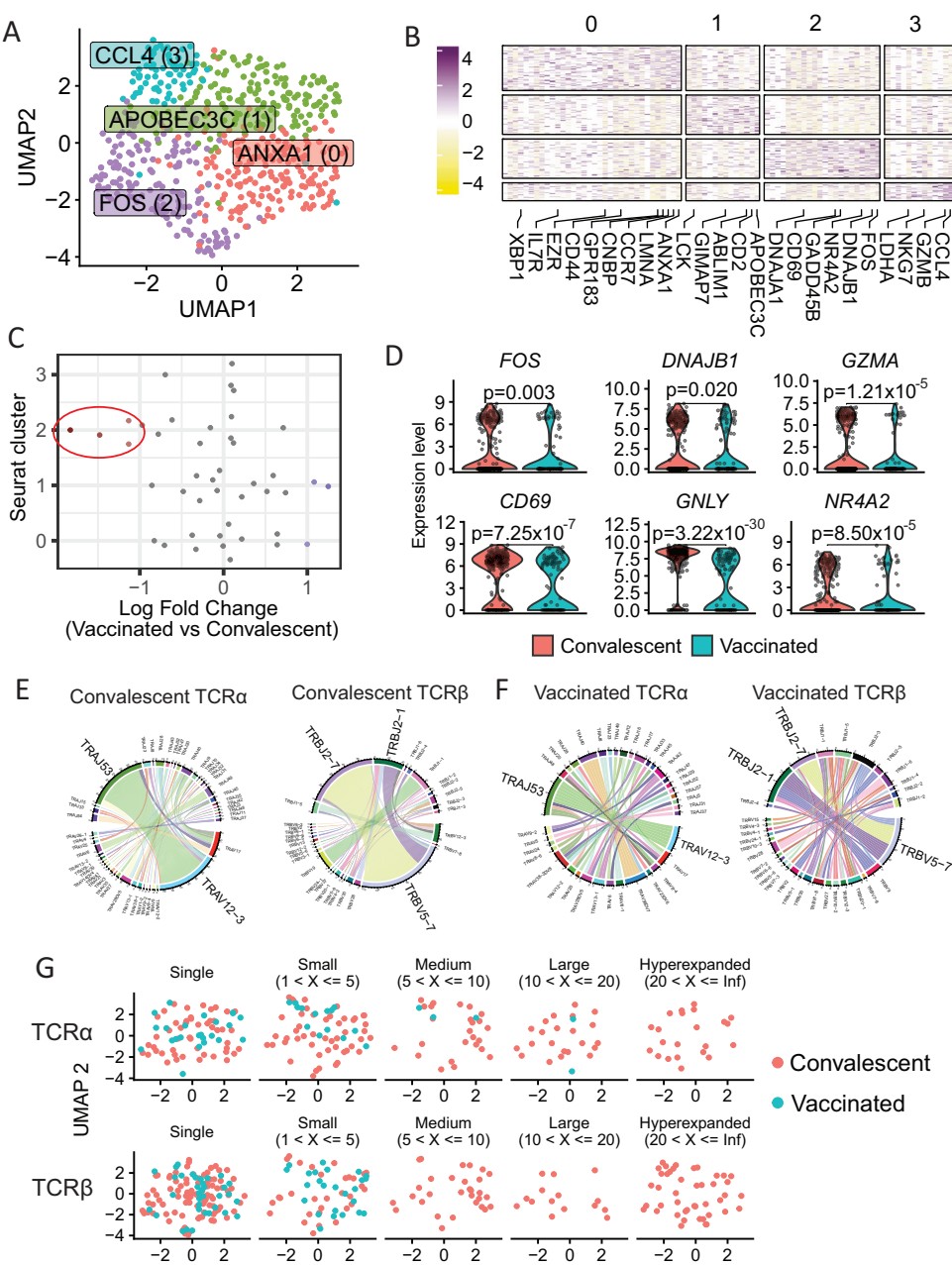

**Fig. 5 | scRNAseq transcriptomic and TCR repertoire analysis of ex vivo MPXV-specific CD8⁺ T cells from vaccinated (n = 6) and convalescent (n = 4) individuals.** Tetramer⁺CD8⁺ T cells were used for single-cell transcriptomic and TCR repertoire analysis. **A** UMAP plot of tetramer⁺CD8⁺ T cells integrated based on patient of origin, identifying 4 clusters of cells. **B** Heatmap of the differentially expressed marker genes of the 4 cell clusters. **C** Beeswarm plot showing the log-fold distribution of cell abundance changes between convalescent and vaccinated samples. Neighbourhoods overlapping the same cell population are grouped together, neighbourhoods exhibiting differential abundance are coloured in red and are circled. **D** Violin plots of cluster 2 marker genes between vaccinated and convalescent samples. **E** and **F** Circos plots showing the TRAV-TRAJ and TRBV-TRBJ gene pairing for convalescent (**E**) and vaccinated (**F**) samples. **G** UMAP plot of the tetramer⁺CD8⁺ T cells, split by whether their TCR alpha (top) and beta (bottom) clonotypes are expanded. Differential expression between clusters was carried out using the MAST test. The Wilcoxon signed-rank test was used to test differences between convalescent and vaccinated samples and two-tailed $P$ values were calculated. ****$p < 0.0001$, ***$p < 0.001$, **$p < 0.01$, *$p < 0.05$.

identifying 4 cell clusters. Comparison of the gene expression of the cell clusters identified differences in the overall gene signatures (Fig. 5B), including a cytotoxic cluster with high expression of *CCL4*, *GZMB*, and *NKG7* (cluster 3), as well as a more activated cluster with high expression of *FOS* and *CD69* (cluster 2).

To investigate whether there was a different abundance of cells from convalescent samples or from samples after vaccination in the identified clusters, we used Milo, a framework for identifying differences in cellular density without reliance on cellular labels (Fig. 5C)[34].

We observed mild changes in the cellular density of cluster 2 cells between samples from convalescent and vaccinated individuals, with an increased proportion of cells from convalescent samples in cluster 2. Comparing the cells from convalescence and after vaccination, we observed significantly increased expression of the cluster 2 marker genes *FOS*, *DNAJA1*, *NR4A2*, and *CD69*, as well as *GNLY* and *GZMA* in cells from convalescence (Fig. 5D, Supplementary Fig. 5B).

Furthermore, examining the TCR repertoire of the ILD-specific T cells revealed a diverse TCRα and TCRβ repertoire. For both TCRα

and TCRβ, there appeared to be a slight bias in V-J gene usage for cells after vaccination (Fig. 5F). However, for cells from convalescence, *TRBV5-7-TRBJ2-7* was the most dominant beta gene used, while *TRAV12-3-TRAJ53* was the most dominant alpha gene used by the MPXV-specific T cells (Fig. 5E), showing greater expansion in cells from convalescence than after vaccination. Investigating the expansion of the TCRα and TCRβ clonotypes, all the expanded TCRα clonotypes are from convalescent samples, whereas 56% of cells from vaccines had an alpha clonotype found in just one cell, compared to 29% of cells from convalescent individuals which had a clonotype found in just one cell (Fig. 5G). Similarly, the expanded TCRβ clonotypes were also from convalescent samples, with the majority of beta clonotypes from vaccinated individuals also only identified in one cell. Furthermore, we investigated whether any of the identified clonotypes were public (found in >1 individual). Four beta clonotypes were found to be public in our dataset, of which two were found in samples from both convalescence and after vaccination (CASSLASGWNEQFF_*TRBV5-7* and CASSSSGSWYEQYF_*TRBV5-7*). Nine alpha clonotypes appeared to be public and found in more than one sample, of which five were found in both convalescent and vaccinated individuals, including CAMSANSGGSNYKLTF_*TRAV12-3* which was identified in all four convalescent individuals and 2 of the vaccinated individuals.

## MPXV-specific CD8⁺ T cells after infection are more activated and cytotoxic and show greater migratory potential compared to vaccination and SARS-CoV-2-specific CD8⁺ T cells

As we identified overall differences between MPXV-specific T cells during convalescence and after vaccination, we investigated these differences further to explore how MPXV-specific CD8⁺ T cells in convalescence compared to antigen-specific CD8⁺ T cells in another viral infection, namely SARS-CoV-2-specific CD8⁺ T cells[35]. Briefly, 912 SARS-CoV-2 $NP_{105-113}$-B*07:02-specific T cells were isolated using peptide-MHC pentamers from 10 individuals, 1–3 months after initial infection, and single-cell RNAseq was carried out to investigate gene expression and TCR repertoire differences.

MPXV-specific T cells from convalescent individuals showed greater expression of several integrin and migratory genes, including *ITGB1* ($p = 3.22 \times 10^{-2}$), *ITGA4* ($p = 3.89 \times 10^{-3}$), and *GPR183* ($p = 2.02 \times 10^{-6}$), and a suggestive increase in *ITGB7* and *ITGA6* (Fig. 6A). Furthermore, CD44 plays a role in skin migration of T cells, as well as in T cell activation, and showed a significant increase in expression in convalescence compared to after vaccination ($p = 1.79 \times 10^{-4}$), together with *CD69* ($p = 5.21 \times 10^{-3}$), a marker of T cell activation (Fig. 6A). Moreover, the increased cytotoxicity profiles of the MPXV-specific T cells in convalescent individuals compared to vaccinated individuals is due to the increased expression of *GNLY* ($p = 5.61 \times 10^{-16}$), *GZMK* ($p = 3.69 \times 10^{-3}$) and *GZMA* ($p = 3.95 \times 10^{-2}$) (Fig. 6A).

We next generated module scores for cytokine gene expression, integrins, cytotoxicity, activation and chemokines using gene signatures obtained from the literature (Fig. 6B, Supplemental Data Table 4) and compared these between the MPXV-specific T cells in convalescence with MPXV-specific T cells after vaccination and SARS-CoV-2-specific CD8⁺ T cells. The overall cytokine and chemokine expression of MPXV-specific T cells was lower than that of the SARS-CoV-2-specific T cells ($p = 8.64 \times 10^{-11}$ and $1.6 \times 10^{-27}$, respectively). However, ILD-specific cells from convalescence showed a greater cytokine signature compared to ILD-specific T cells from vaccinated individuals ($p = 5 \times 10^{-3}$) with no difference in the chemokine signature between MPXV convalescence and vaccination ($p > 0.05$). Moreover, MPXV-specific T cells appear more activated ($p = 1.26 \times 10^{-5}$ and $p = 4.89 \times 10^{-7}$, respectively) and show greater cytotoxicity ($p = 3.56 \times 10^{-25}$ and $5.86 \times 10^{-8}$, respectively) than SARS-CoV-2-specific T cells and MPXV-specific T cells after vaccination. Investigating the individual genes, MPXV-specific T cells show increased expression of several cytotoxic molecules (Fig. 6C) compared to SARS-CoV-2-specific T cells, including *GZMA* and *GNLY* (log2FC = 1.93 and 1.27 and $p = 0.002$ and $5.48 \times 10^{-71}$, respectively), as well as genes upregulated following T cell activation, including *CD44* (log2FC = 0.15, $p = 0.049$) and *CD69* (log2FC = 1.07, $p = 5.86 \times 10^{-12}$) (Fig. 6C). Furthermore, T cell entry into the skin is initiated by interactions with blood vessel endothelial cells, including the interaction between CD162 (*SELPG*) with E-selectin; and the interaction between CD44 and very late activation Ag-4 (VLA-4) (*ITGA4/ITGB1*) with VCAM-1[36]. *ITGB1* (log2FC = 0.38, $p = 4 \times 10^{-6}$) shows increased expression in MPXV-specific T cells compared to SARS-CoV-2-specific T cells while *ITGB1* and *ITGA4* show increased expression in MPXV-convalescence compared to vaccination (log2FC = 0.62, $p = 1.6 \times 10^{-3}$ and log2FC = 1.20, $p = 0.042$ respectively) (Fig. 6C). Moreover, MPXV-specific T cells express higher levels of *GPR183* (log2FC = 1.27, $p = 1.07 \times 10^{-15}$), previously shown to promote skin-infiltration of Tγδ17 cells[37].

To validate these findings, we assessed the protein expression of several cytotoxic (Granzyme A and Granulysin) and migratory (CD44, CD49d and CD29) molecules on ILD-Teramer⁺CD8⁺ T cells with flow cytometry (Figs. 6D and S5). In agreement with the scRNAseq analysis, cells from convalescent individuals show increased expression of CD29 (*ITGB1*, $p = 0.047$) CD44 ($p = 0.008$) and CD49d (*ITGA4*, $p = 0.008$). Furthermore, there is an increased percentage of cells expressing GNLY ($p = 0.047$) and GZMA ($p = 0.0278$) in convalescence.

Taken together, these data suggest that compared to MVA-BN vaccination, MPXV-specific memory T cells in convalescence are more cytotoxic and activated, with increased expression of migratory molecules, suggesting these T cells may be poised and ready to more effectively act upon re-infection with a greater likelihood to be acting in the skin, the primary site of MPXV infection.

## Discussion

In this study, we demonstrated that memory T cells elicited by mpox infection showed a strong response against naturally processed antigens when PBMCs from MPXV convalescent donors were co-cultured with autologous PBMCs infected with the VACV by ex vivo ELISpot. We were able to achieve a response in 93% of individuals (12 out of 13) with this approach compared to 58% (7 out of 12) using the MPXV mega pool. Furthermore, we reported HLA-A*02:01 restricted MPXV-specific epitopes which have been reported as VACV epitopes, where six of them are conserved between VACV and MPOX, one is specific for MPXV. Moreover, we provide an overall comparison of T cell responses between individuals after natural MPXV infection and after MVA-BN vaccination. Compared to vaccination, T cells elicited by mpox showed increased cytotoxicity and activation, a more expanded TCR repertoire and high potential to migrate to the site of infection.

The use of virus-infected cells to quantify CD8⁺ and CD4⁺ T cell responses is important, as it integrates the natural processing of antigens, which cannot be ascertained using peptide pools. Using this approach, we observed a similar magnitude of CD8⁺ and CD4⁺ T cell responses from MPXV recovered donors. This finding contradicts recent reports that shows a greater presence of CD4⁺ T cells compared to CD8⁺ T responses in MPXV convalescence[12,14] when using peptide mega pools. One reason for this may be the limitation in the coverage of the mega pools. In our study, immune responses were measured against the whole VACV, resulting in a response more similar to natural infection, with a potentially broader epitope coverage compared to peptide-based assays[38]. This may also explain why a much lower response was detected to our CD4 mega pool compared to our CD8 mega pool, which only covers 90 ORFs for CD4 epitopes, fewer epitopes than in the mega pools used in previous studies. Moreover, different assays were used for detection of T cell responses capture different aspects of the response. The IFNγ ELISpot assay used in our study may have missed non-IFNγ producing CD4⁺ T cell populations, which can be captured by AIMs assay.

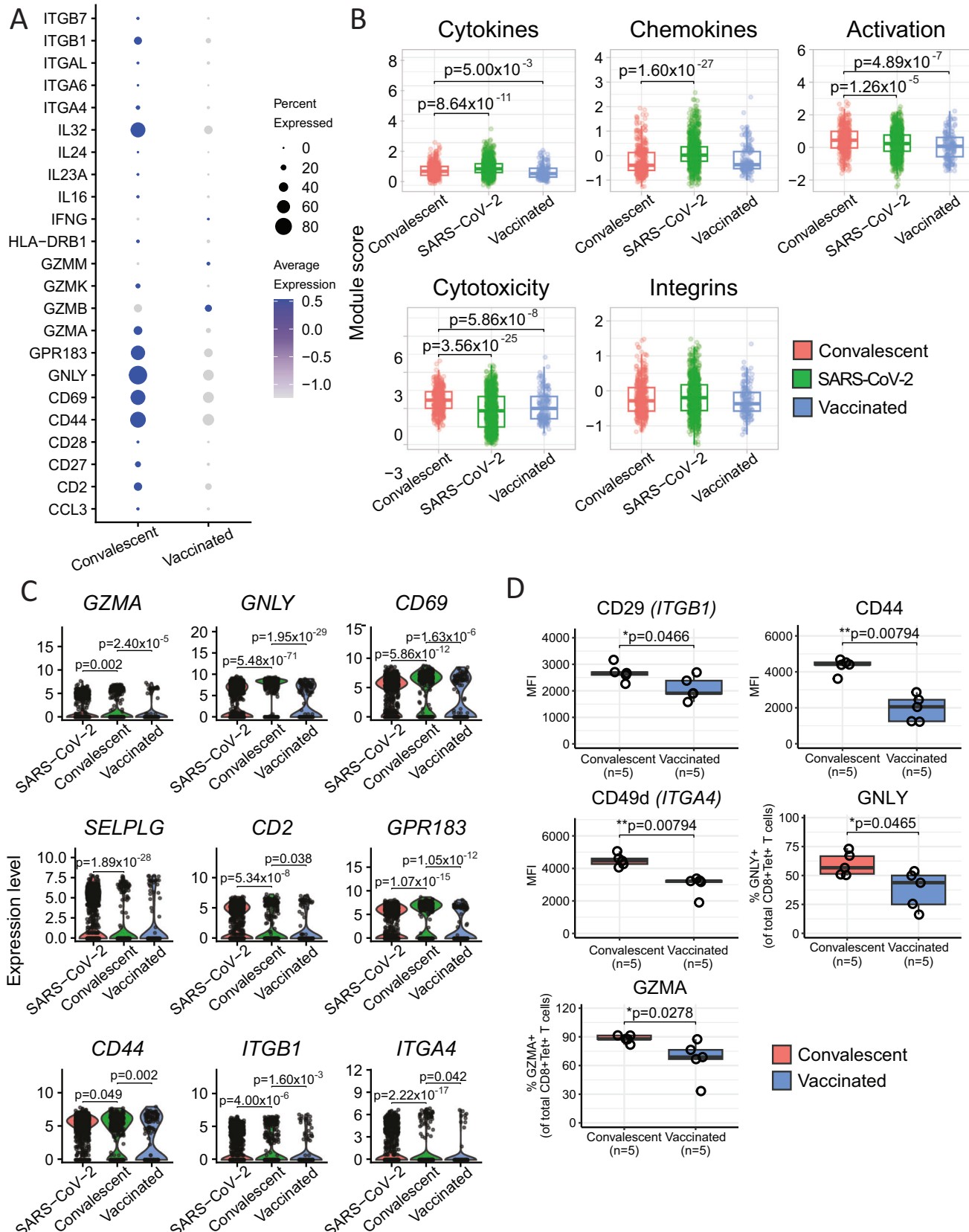

Differences in the phenotype and functionality of T cells primed by either vaccination or infection have been previously reported in SARS-CoV-2[39–42]. It is therefore important to consider whether similar differences in T cell responses would emerge between mpox infection and vaccination. Although both natural infection and vaccination can

induce a comparable overall MPXV-reactive T cell response, MPXV-reactive CD8+ T cells in convalescent individuals exhibit a T$_{EMRA}$ (CD45RA+CCR7-) phenotype[12], showing greater activation and effector profiles, with increased CD45RA, CD38, HLA-DR and GZMB expression. Consistent with these findings, our data showed MPXV-specific CD8+

**Fig. 6 | scRNAseq transcriptomic comparison between MPXV-specific and SARS-CoV-2-specific CD8[+] T cells. A** Dot plot of the gene expression of cells from convalescent and vaccinated individuals for cytokines, cytotoxicity molecules, integrins and activation markers. **B** Comparison of module scores for cytokines, integrins, cytotoxicity genes, activation-related genes and chemokines between MPXV-specific (401 and 160 cells from convalescence and vaccination, respectively) and SARS-CoV-2-specific CD8[+] T cells (912 cells). **C** Violin plots depicting the expression of select genes between MPXV-specific and SARS-CoV-2-specific CD8[+] T cells. **D** Comparison of cytotoxicity molecules and integrins expressed on ILD-tetramer[+]CD8[+] T cells between convalescent ($N = 5$) and vaccinated ($N = 5$) individuals by flow cytometry. The Wilcoxon signed-rank test was used in (**B**) and (**C**). In **D**, the Kolmogorov–Simonov test was used to test for normality and an unpaired *t*-test was used for CD29 and the Mann–Whitney *U*-test was used for the rest. Box-plots in **B** and **D** represent the 25th and 75th percentiles with the median marked with whiskers at ±1.5*IQR. Two-tailed *p* values were calculated. ****$p < 0.0001$, ***$p < 0.001$, **$p < 0.01$, *$p < 0.05$. Source data are provided as a Source Data file.

T cells elicited from both natural infection and vaccination displayed a dominant $T_{EMRA}$ phenotype. However, in this cell population, significantly lower KLRG1 expression was observed in convalescent individuals. CD8[+]CD45RA[+]CCR7[−] cells have been previously reported in SARS-CoV-2[40], EBV[43], and CMV infection[44]. These cells were shown to exhibit enhanced proliferation and differentiation potential in vivo compared to other memory and non-virus-specific cell subsets[45]. Herndler-Brandstetter et al. [45] have previously identified a population of exKLRG1 memory T cells, which are KLRG1[+] T cells that receive intermediate amounts of activating signals during the contraction phase, resulting in a downregulation of KLRG1 in a Bach2-dependent manner and differentiate into all memory T cell lineages with varying CX3CR1 expression levels. These cells retain high cytotoxicity and proliferation, contributing to effective anti-viral immunity. Therefore, the reduction in KLRG1 levels detected in convalescence compared to post-vaccination suggests that in convalescence, long-term MPXV-specific memory T cells with high cytotoxicity may be generated. These observations are further confirmed by our scRNAseq analysis, in which MPXV-specific T cells from convalescent individuals exhibited greater cytotoxicity and activation signatures compared to vaccinated individuals. Moreover, investigating the TCR repertoire of T cells from vaccinated and convalescent individuals showed that the T cells from convalescent individuals have more expanded TCR clonotypes, as well as a greater bias in gene usage, compared to T cells from vaccinated individuals. With a more diverse TCR repertoire, T cells after vaccination may be able to contain variants more effectively.

In addition, our single-cell and flow cytometry analysis revealed that MPXV-reactive CD8[+] T cells in the convalescence showed distinguished integrin and migratory gene profiles with the high expression level of *ITGA4* (CD49d), *ITGB1* (CD29), *SELPLG*, *CD44*, and *GPR183*, compared to after MVA-BN vaccination and SARS-CoV-2 infection. T cell entry into the skin involves lymphocyte rolling on vascular endothelial cells, mediated by very late activation Ag-4 (VLA-4) ($\alpha_4\beta_1$, CD49d/CD29) and CD162 (*SELPLG*)[36]. The increased expression of these molecules may contribute to increased migratory potential of MPXV-specific T cells to the skin lesions and inflammation seen in mpox. GPR183, also known as Epstein-Barr virus-induced G-protein coupled receptor 2 (EBI2), has been shown to play a role in the positioning of various immune cells[46–49], and is also a mediator of T cell migration. Frascoli et al. [37] show that dermal Tγδ17 cells require oxysterol sensing through GPR183 to maintain the skin-barrier homoeostasis and migrate and accumulate at the dermal barrier. Local dermal inflammation as a result of MPXV infection may result in an increased level of oxysterols, promoting the migration of GPR183[+] MPXV-specific CD8[+] T cells to the site of infection. These results suggest that T cells primed at different sites show different transcriptomic profiles, antigen-specific T cells primed by MPXV infection may have greater migratory potential to inflammatory skin infection sites, compared to vaccination and SARS-CoV-2 infection.

Differences in the TCR repertoire and expression profiles of ILD-specific T cells after natural infection and vaccination suggest that variations in T cell priming may lead to different longer-term outcomes. Our data showed that MPXV infection could elicit memory T cells with greater effector function and migratory potential to the

site of infection. Therefore, improving vaccine design to elicit a T cell response similar to natural infection may provide better long-term outcomes, with potentially inducing a T cell memory with a greater homing potential to the skin, the primary site of infection for mpox. Although studies have shown no difference in circulating antibody titers between subcutaneous and intradermal vaccination for mpox[50], further investigation is needed to understand the impact of different vaccine administration methods on the T cells response. It is likely that intradermal vaccination may better prime T cells at the infection site and induce a tissue-resident T ($T_{rm}$) cell memory. Furthermore, it would be interesting to investigate whether individuals with a historical smallpox vaccine can also generate long-lasting T-cell responses to immunodominant MPXV epitopes and how these responses compare to those induced by current MVA-BN vaccinations and natural infection.

To our knowledge, this is the only study currently investigating the overall T cell responses following natural MPXV infection and following vaccination. While similar studies have been conducted in other viral infections (e.g. SARS-CoV-2), our work provides insight into the differences seen following orthopoxvirus infection. In addition to providing an overview of the overall CD4[+] and CD8[+] T cell responses and phenotypes, we characterize the transcriptome and TCR repertoire of immunodominant HLA-A*02:01-G5$_{18-26}$ (ILDDNLYKV)-specific T cells at a single-cell level, identifying key differences in potential function of the antigen-specific cells being primed differently. These differences could be strengthened with more robust single-cell analysis at the transcription level and protein level with high-dimensional flow cytometry phenotype. However, our study has several limitations. Firstly, the sample size is relatively small and warrants validation in a larger number of cells and patients. The frequency of antigen-specific cells in circulation is relatively low, and as such, we were only able to investigate transcriptional changes in a low number of ILD-specific and compare with a low number of NP$_{105-113}$-specific T cells from Peng et al. [35]. Although we have increased the number of participants from which NP$_{105-113}$-specific T cells were investigated to overcome the issues of patient heterogeneity, validation of these results is needed in a larger number of cells while reducing batch effects arising from different patients. Secondly, although most of the individuals in this study don't have a historical smallpox vaccination, two individuals from the vaccinated cohort had. While there was no evident effect of this in our data; however, further analysis is required in a larger number of individuals with historical smallpox vaccination, to determine its impact on the T cell response to MPXV. In addition, our detailed phenotypical analysis and the single-cell RNASeq in this study only focused on the single dominant HLA-A*02:01 restricted ILD-epitope with a limited number of cells from vaccinated donors. Further investigation into other dominant MPXV-specific T-cell responses with greater cell numbers is necessary to enhance the robustness of our analysis. Moreover, in this study, we present the characteristics of MPXV-specific T cells in the peripheral blood, it would be crucial to investigate the MPXV-specific T cells in infectious skin lesions alongside circulating T cell responses to fully understand the role of T cells in controlling of MPXV infection. Given our findings suggesting increased migratory potential of ILD-specific T cells to the skin, it would be valuable to examine the expression of these migratory

markers on T cells in the skin lesions which were not available for this cohort but provides an interesting avenue for further investigation.

In summary, we used a Vaccina virus-infected cell-mediated approach to measure the MPXV-specific memory T-cell response in samples collected from both a valuable pre-positioned Urgent Public Health Research protocol and a rapidly developed trial of the MVA-BN vaccine. Overall VACV-reactive T cell responses were significantly higher among convalescent individuals than healthy controls, and specific CD8[+] and CD4[+] T cells had a similar magnitude of response. Furthermore, we identified a more cytotoxic and migratory response of MPXV-specific T cells in convalescence. Our approach will be useful for the analyses of the immunogenicity of MVA vaccinia vaccines and for basic studies of human T cell memory for orthopoxviruses. This study has possible implications for further improvements in the design of newer generation of MPXV vaccines, as although MVA-BN vaccination may be sufficient to induce a T cell response, there may still be room to improve, to induce T cell responses more similar to natural infection.

## Methods

### Study participants
Individuals were invited to participate in the ISARIC Clinical Characterization Protocol (REC: 13/SC/0149) study if they had presented with a clinical syndrome compatible with the 2022 mpox outbreak, and had one or more molecular tests performed on throat, rectal, skin or vesicle swab(s) positive for Clade II MPXV infection. Following informed, written consent, participants underwent a blood sample at least 28 days following their initial presentation, and details regarding their clinical background, presentation, and management were collected.

### Clinical definitions
Severity of infection was classified as Mild or Moderate/Severe based on guidance from the WHO[51], patients who met any of the following criterion were classified as having moderate/severe infection:
- >25 vesicles/lesions.
- Significant and painful lymphadenopathy and swelling (requiring referral to specialist team e.g. surgical or anaesthetic team).
- Volume depletion requiring resuscitation (in this cohort this was exclusively linked to blood loss from proctitis).
- Clinical syndrome compatible with sepsis and/or septic shock.
- Evidence of end-organ damage.

None of the cases presented with severe neurological, ocular, or respiratory complications requiring intensive care input or organ support.

### Vaccinated cohort study participants
This was a prospective observational study of the immune responses to the MVA-BN vaccination[21]. Individuals attending the vaccination clinic in Oxford, UK, and receiving the MVA-BN vaccine were invited to participate in this study. As recommended by the United Kingdom Join Committee on Vaccination and Immunization guidelines, individuals were administered with an intradermal dose regimen of two doses of 0.1 ml MVA-BN, given between 26 and 40 days apart, as in Drennan et al. [21]. Participants had blood sampling at baseline (day 0), day 14 and day 28 after their first dose, and additional blood sampling 28 and 90 days after their second dose. The study was approved by the UK NHS Ethics Committee (London−Surrey Borders Research REC: 22/PR/1425. All participants provided written informed consent.

### Peptide synthesis and Mega pool construction
Mega pool peptides were based on identifying the corresponding conserved sequences in the MPXV genome of T cell assay-defined VACV CD4 and CD8 epitopes from the Immune Epitope Database

(IEDB, https://www.iedb.org). In brief, we utilized epitope data sourced from the IEDB on VACV to predict potential targets of MPXV that are recognized by human CD4[+] and CD8[+] T cells. Firstly, we compiled a comprehensive list of experimentally validated VACV epitopes in humans documented in the literature or curated by IEDB as of September 2022, against all strains of VACV (Western Reserve, Ankara, Copenhagen, New York City Board of Health [NYCBH]−Dryvax and Modified Vaccinia Ankara [MVA] virus strains). 273 CD8[+] VACV epitopes and 232 CD4[+] VACV epitopes were included, and alignment was performed against the MPXV (Clade IIb, Glasgow 2022 strain) genome to predict MPXV orthologues and synthesize predicted MPXV epitopes. We developed four MPXV mega pools MPXV-CD8-early and MPXV-CD8-intermediate/late, MPXV-CD4-early and MPXV-CD4-intermediate/late. We further divided the CD8 peptides into HLA-A*02 and non-HLA-A*02 peptide pools. Peptides were synthesized as crude materials (GenScript Biotech) (Supplementary Data Table 3). Pools of HCMV, Epstein-Barr virus, and influenza virus-specific epitope (FEC) peptides were used as positive controls. FEC peptide mix was supplied by AIDS reagents and contains 32 published epitopes recognized by CD8 positive T cells and presented by 12 class I HLA-A and HLA-B types.

In parallel, a total of 257 15- to 18-mer peptides overlapping by ten amino acid residues and spanning the full proteome of MPXV A9, F3, E12, D10 and D1 (Supplementary Data Table 3) were designed using the PeptGen software (http://www.hiv.lanl.gov/content/sequence/PEPTGEN/peptgen.html) and synthesized (GenScript Biotech).

### Evaluation of T cell responses to VACV infection
PBMCs were infected with the VACV strain Lister at an MOI of 3 for 90−120 min at 37 °C. Cell was washed twice to remove any excess virus and cocultured with autologous PBMCs at an effector:target (E:T) ratio of 4:1 for either 16−18 h (for ELISpot assay) or 20−24 h (for AIMs assay).

### Ex vivo ELISpot
IFN-γ ELISpot assays using VACV-infected PBMCs as target cells were performed using freshly isolated PBMCs, as described[22]. Briefly. $4 \times 10^5$ PBMCs were incubated in duplicate wells with $1 \times 10^5$ autologous VACV-infected cells. $4 \times 10^5$ uninfected PBMCs (no antigen) and $1 \times 10^5$ infected PBMCs with no effector cells were used as background controls, while PBMCs treated with influenza, EBV, and HCMV (FEC) peptide mix and PHA were used as positive controls. To evaluate the T cell response against the peptide mega pools or overlapping peptides, IFN-γ ELISpots were performed using cryopreserved PBMCs. Overlapping peptides or Mega pools were added to 200,000 PBMCs per well at a final concentration of 2 μg/ml for 16−18 h. To quantify antigen-specific responses, mean spots of the background control wells (uninfected PBMCs and infected PBMCs with no effector cells for VACV-infection experiments; and uninfected PBMCs for MPXV mega pool experiments) were subtracted from the sample wells, and the results were expressed as spot-forming units (SFU) per $10^6$ PBMCs. Responses were considered positive if the results were at least three times the mean of the negative control wells and >25 SFU per $10^6$ PBMCs. If negative control wells had >30 SFU per $10^6$ PBMCs or positive control wells (PHA stimulation) were negative, the results were excluded from further analysis.

### Ex vivo activation-induced markers (AIM) assay
Antigen-specific T cells were evaluated using the activation-induced markers (AIMs) assay, measuring the percentage of OX40[+]CD40L[+] CD4[+] and CD69[+]CD137[+] CD8[+] T cells after co-culturing with VACV-infected PBMCs. Before co-culturing with $0.3 \times 10^6$ VACV-infected PBMCs, $1.2 \times 10^6$ fresh or cryopreserved autologous PBMCs cryopreserved using freezing medium of 10% DMSO in FCS were blocked at 37 °C for 15 min with 0.5 μg/ml of anti-CD40 monoclonal antibody (mAb) (0.5 μg/ml, Miltenyi, 130-094-133), followed by the addition of anti-CD28/CD49a (1 μg/ml, BD Biosciences, 347690) at a final

concentration of 1 μg/ml[22,23]. Subsequently, cells were incubated in R10 medium (RPMI + 10% FCS + 1% penicillin/streptomycin + 1% ʟ-glutamine) at 37 °C for 20–24 h in 96-well U-bottom plates, as described[24]. In addition, PBMCs without antigen were used as background negative controls and with PHA at 20 μg/ml as an AIMs assay positive control. The next day, PBMCs were resuspended in phosphate-buffered saline (PBS), incubated with Biolegend human FC block (Biolegend, 422302) and a LIVE/DEAD marker (Invitrogen, L34965) in the dark for 15 min and washed with PBS containing 5% FBS (FACS buffer). An antibody mix containing the rest of the surface antibodies in Brilliant Stain Buffer (BD Biosciences, 563794) was added directly to cells and incubated for 50–60 min at 4 °C in the dark. After surface staining, cells were washed twice with PBS containing 5% FBS. All samples were acquired on Attune NxT flow Cytometer (software v.3.2.1) and analysed using FlowJo version 10 (FlowJo LLC). A list of antibodies used in this panel can be found in Supplementary Data Table 5, and a representative gating strategy of VACV-specific CD4[+] and CD8[+] T cells using the AIM assay is shown in Supplementary Fig. 3A. We analysed the data using FlowJo and performed a live cell gate using forward- and side-scatter characteristics. Antigen-specific CD4[+] and CD8[+] T cells were measured as background-subtracted data[13]. A response of >0.02% and a stimulation index[13] (SI, the ratio of signal to background) of >2 for CD4[+], and a response of >0.03% and SI of >3 for CD8[+] T cells, were considered positive. The limit of quantification (LOQ) for antigen-specific CD4[+] T cell responses (0.03%) and antigen-specific CD8[+] T cell responses (0.06%) was calculated as the median twofold standard deviation of all negative controls.

### Generation of short-term CD8[+] T cells lines and antigen-specific bulk cell lines

Short-term T-cell line was generated as described[22]. Briefly, PBMCs were pulsed with 4 μg/ml of CD8 peptide mega pool for 1 h, washed once with H10 media (RPMI + 10% human AB serum [Sigma, H4522]) and seeded at 0.25 × 10⁶ per well in a 96-well plate with 100 μl of H10. IL-2 was added to a final concentration of 100 U/ml on day 3. Cultures were split as needed, fed periodically with IL-2-H10 for a further 10–14 days and tetramer[+] cells were sorted and expanded as CD8 bulk lines by stimulation with PHA in the presence of irradiated PBMC feeders for a further 10–14 days.

### Synthesis of UV exchanged peptide-HLA-A*02:01 tetramers and staining

Generation of peptide-MHC class I monomers and tetramerization was performed as described[22]. Briefly, biotin-tagged HLA-A*02:01 complexes were folded with the UV-sensitive peptide KILGFVFJV. 2 μg of HLA-A2 complexes were UV-exchanged for 1 h with each screening peptide at a final concentration of 200 μg/ml in 20 μl. After centrifugation at 2250 g, 1.5 μg of complexes (15 μl supernatant) were collected and tetramerized with 1.5 μl of a 1:1 mix streptavidin-APC:streptavidin-PE (Invitrogen, 17.4317-82 and 12-4317-87 respectively). Free biotin was blocked by adding 20 μl of 50 μM D-biotin. 1 × 10⁵ stimulated PBMCs were incubated in 50 μl staining buffer (PBS + 0.5% BSA) containing 2.5 μl of multimers, for 30 min at 37 °C. Cells were washed twice with staining buffer and stained with LIVE/DEAD Fixable Aqua (Invitrogen, L34965) in the dark for 10 min and washed once with PBS containing 5% FBS (FACS buffer). Anti-CD3 FITC (1:25, clone SK7, BD Pharmingen, 345764) and anti-CD8 BV421 (1:50, Clone SK1, Biolegend, 344748) was added and incubated for 20–30 min at 4 °C. All samples were acquired on Attune NxT flow Cytometer (software v.3.2.1) and analysed using FlowJo version 10 (FlowJo LLC).

### Intracellular cytokine staining (ICS)

ICS was performed as described[22]. Briefly, 100,000 cultured bulk CD8[+] T cells were co-cultured with either 100,000 VACV-infected BCLs as described above or individual peptide-pulsed BCLs at a final concentration of 1 μg/ml, together with BD GolgiPlug (BD Biosciences, 555029), BD GolgiStop (BD Biosciences, 554724) and PE-anti-CD107a (1:20, BD Biosciences, 555801) for 5 h at 37 °C. Dead cells were labelled with LIVE/DEAD Fixable Aqua dye (Invitrogen, L34965). Cells were stained for surface markers with the following antibodies: FITC-anti-CD8 (1:33, BD Biosciences, 345772), BV510-anti-CD14 (1:33, BioLegend, 301842), BV510-anti-CD16 (1:33, BioLegend, 302048) and BV510-anti-CD19 (1:33, BioLegend, 302242). Cells were washed, fixed with Cytofix/Cytoperm (BD Biosciences, 51-2090KZ51) and stained with PE-Cy7-anti-IFNγ (1:50, BD Biosciences, 557643), APC-anti-TNFα (1:200, Thermofisher Scientific, 17-7349-82), BV421-anti-IL-2 (1:33, BioLegend, 500328) and APC-H7-anti-MIP1β (1:33, BioLegend, 561280). Negative controls comprising a mix of bulk T cells with BCLs without peptide stimulation were run for each sample. All samples were acquired on Attune NxT flow Cytometer (software v.3.2.1) and analysed using FlowJov.10 software (FlowJo LLC). Reactive CD4[+] or CD8[+] T cells with a frequency lower than 0.05% were excluded from the analysis. Cytokine responses were background subtracted individually before further analysis. To determine the frequency of different response patterns based on all possible combinations, Boolean gates were created using IFN-γ, TNF-α, IL-2, CD107a, and MIP1β. A representative gating strategy for the ICS assay is shown in Supplementary Fig. 3B.

### Ex vivo phenotyping with peptide-MHC class I tetramers

Cryopreserved PBMCs were thawed and rested for 3 h. A total of 3−5 × 10⁶ live PBMCs were stained with A*02:01-ILDDNLYKV tetramer produced in-house using standard methods[52] and incubated for 20 min at 37 °C. Dead cells were first labelled with LIVE/DEAD Fixable Aqua dye (Invitrogen, L34965) and CD14-BV510 (1:33, BioLegend, 301842), CD19-BV510 (1:33, BioLegend, 302242) dumping markers; then with the surface markers CD3-BUV395 (1:50, BD Biosciences, 564001), CD45RA-APC-H7 (1:33, BD Biosciences, 560674), CD8-PerCP.Cy5.5 (1:33, BD Biosciences, 565310), CD27-BUV496 (1:50, BD Biosciences, 750168), CCR7-BV421 (1:33, BioLegend, 353208), CX3CR1-BV711 (1:33, BioLegend, 341630), CD57-BV785 (1:33, BioLegend, 393330), and KLRG1-AF488 (1:33, ThermoFisher Scientific, 53-9488-42) for phenotyping or CD3-BUV805 (1:66, BD Biosciences, 612895), CD8-BUV395 (1:50, BD Biosciences, 563795), CD49d-BV786 (1:100, BD Biosciences, 744588), CD44-BUV496 (1:100, ThermoFisher Scientific, 364-0441-80), CCR7-BV421 (1:33, BioLegend, 353208), CD45RA-BV711 (1:100, BioLegend, 304138), CD29-PE-Dazzel (1:50, BioLegend, 303032), CD69-APC-Cy7 (1:33, BioLegend, 310914) for the panel to verify single-cell RNASeq data. Staining of intracellular markers Peforin-AF488 (1:50, BioLegend, 353320), Granulysin-AF647 (1:33, BioLegend, 348006) and Granzyme A-PerCP-Cy5.5 (1:33, BioLegend, 507216) was carried out after the cells were fixed with Cytofix/Cytoperm (BD Biosciences, 51-2090KZ), for 20 min. All samples were acquired on a BD LSRFortessa X50 (BD Biosciences) flow cytometer and analysed using FlowJo version 10. A representative gating strategy for the ex vivo phenotyping of tetramer[+] T cells is shown in Supplementary Fig. 3C (immunophenotyping) and Supplementary Fig. 6 (scRNAseq validation).

### CFSE-based cytotoxic T lymphocyte killing assay

We carried out a carboxyfluorescein diacetate succinimidyl ester (CFSE, Invitrogen, C34554)-based cytotoxic T cell killing assay as previously described[53]. Briefly, HLA-A*02:01-positive BCLs were infected with the VACV strain Lister at an MOI of 3 overnight, Cells were washed, counted before being labelled with 0.5 μM CFSE. Subsequently, cells were co-cultured with T cells at an E:T ratio of 2:1, 1:1, and 1:2 at 37 °C for 6 h. Samples were then stained with 7-AAD (eBioscience, 00-6993-50) and CD19-BV421 (1:33, BioLegend, 302234). Cell death was assessed based on the presence of CFSE[+]CD19[+]7-AAD[-] (live) cells. Control wells containing non-infected BCLs and T cells at corresponding E:T

ratio were included for each sample. Samples at the same volume were run on Attune NxT Flow Cytometer (software v.3.2.1) and analysed using FlowJo v.10 software (FlowJo LLC). Calculation of killing (%) was carried by the following formula: 100* (Live non-infected BCLs with T cells−live VACV-infected BCL with T cells at same E:T ratio)/Live non-infected BCLs with T cells. A representative gating strategy for the killing assay is shown in Supplementary Fig. 4.

### Cell sorting for single-cell RNA sequencing (scRNAseq)

HLA-A*02:01-G5R$_{18-26}$ (ILDDNLYKV, ILD)-specific CD8$^+$ T cells were sorted with peptide-MHC-class I tetramers. In brief, $1-3 \times 10^6$ cells were stained with PE-conjugated HLA-A*02:01-ILD tetramer. Live/dead fixable Aqua dye (Invitrogen, L34965) was used to exclude nonviable cells from the analysis. Cells were washed and stained with the following surface antibodies: CD3-FITC (1:25, BD Pharmingen, 345764), CD8-PerCp-Cy5.5 (1:33, BD Biosciences, 565310), CD45RA-APC-H7 (1:33, BD Biosciences, 560674), CD14-BV510 (1:33, BioLegend, 301842), CD19-BV510 (1:33, BioLegend, 302242), CD16-BV510 (1:33, BioLegend, 302048), CCR7-BV421 (1:33, BioLegend, 353208), CD27-PE-Cy7 (1:50, BioLegend, 356412), CD57-BV785 (1:33, BioLegend, 393330) and KLRG1-APC (1:33, Thermofisher Scientific, 17-9488-42). After the exclusion of nonviable/CD14$^+$/CD19$^+$/CD16$^+$cells, CD3$^+$CD8$^+$tetramer$^+$ were sorted for single-cell RNAseq using a BD FACSAria Fusion sorter (BD Biosciences). Single cells were directly sorted into 96-well PCR plates (Thermo Fisher Scientific) containing cell lysis buffer and stored at −80 °C for further analysis. A representative gating strategy for the single cell sorting of ex vivo tetramer$^+$ T cells is shown in Supplementary Fig. 3D.

### SmartSeq2 scRNAseq

ScRNA-seq with ex vivo sorted tetramer+ cells was performed using SmartSeq2 analysis as described[54]. Reverse-transcription (RT) and PCR amplification were performed as described with the exception of using ISPCR primer with biotin tagged at the 5′ end and increasing the number of cycles to 25. Sequencing libraries were prepared using the Nextera XT Library Preparation Kit (Illumina, FC-131-1096) and sequencing was performed on Illumina NextSeq sequencing platform with NextSeq Control Software v.4.

### SmartSeq2 scRNAseq data processing and analysis

BCL files were converted to FASTQ format using bcl2fastq v2.20.0.422 (Illumina). FASTQ files were aligned to human genome hg19 using STAR v2.6.1d. Reads were counted using featureCounts (subread v2.0.0). The resulting counts matrix was analysed in R v4.3.1 using Seurat v4.0.1. Cells were filtered using the following criteria: minimum number of cells expressing specific gene = 3, minimum number of genes expressed by cell = 200, and maximum number of genes expressed by cell = 4000. Cells were excluded if they expressed more than 15% mitochondrial genes. Patient-specific cells were integrated using Harmony v.1.0 to remove batch effects. The FindMarkers function (Seurat) was used to evaluate differentially expressed genes (DEGs) between two conditions using model-based analysis of single-cell transcriptomics (MAST) statistical test, with different sequencing batches as latent variables. Module scores were generated using the AddModuleScore function from the Seurat package and compared between groups using the Wilcoxon signed-rank test. Genes used to generate the module scores can be found in the Supplementary Data Table 4. The expression of *CD3E*, *CD8A,* and *PTPRC* of the cells from convalescent and vaccinated individuals is shown in Supplementary Fig. 5.

### SmartSeq2 TCR processing

TCR sequences were reconstructed from SmartSeq2 scRNAseq FASTQ files using MiXCR v.3.0.13 to produce TRB output files for analysis. The output files were parsed into R using immunarch v0.9. TCRs were filtered to retain 1β. Clonotypes were defined as CDRβ amino acid + TRBV. Public clonotypes were defined as shared clonotypes between 2 or more patients. Circos plots were plotted using circlize (v.0.4.12) showing TRB gene usage. All other plots were generated using ggplot2 (v3.5.1) and ggpubr (v.0.4.0).

### Statistical analysis

Statistical analysis was performed with IBM SPSS Statistics 25 or in R v4.3.1 and figures were made with the ggplot2 package in R v4.3.1. Chi-squared tests were used to compare ratio differences between two groups. After testing for normality using the Kolmogorov–Smirnov test, the independent-samples *t*-test or Mann–Whitney *U*-test was employed to compare variables between two groups. Correlations were performed via Spearman's rank correlation. Statistical significance was set at *$P < 0.05$, **$P < 0.01$, ***$P < 0.001$ and ****$P < 0.0001$. All statistical tests were two-tailed.

### Reporting summary

Further information on research design is available in the Nature Portfolio Reporting Summary linked to this article.

## Data availability

The single-cell RNAseq data generated in this study has been deposited in the ArrayExpress database under accession code E-MTAB-14244. Source data for all other figures is provided in a Source Data file. Source data are provided with this paper.

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

## Acknowledgements

We are grateful to all the participants for donating their samples and data for these analyses, and to the research teams involved in the consenting, recruitment and sampling of these participants. We thank K. Clark and S.-A. Clark from the WIMM Flow Cytometry facility for their help with cell sorting and the WIMM Sequencing facility for sequencing. We acknowledge the support of ISARIC4C Investigators. Figures 1a, 2a and 4a were created with BioRender.com. This work is funded by UKRI to the UK Monkeypox Research Consortium (T.D.); Chinese Academy of Medical Sciences Innovation Fund for Medical Sciences 2018-I2M-2-002 (T.D., B.W., Y.P., Y.L., E.A., Z.Y., G.L., X.Y., D.J., J.W. and J.C.K.) and the UK Medical Research Council (T.D., J.-L.C., M.P., A.B., T.R., P.S. and C.W.); The present study is also funded by the NIHR Oxford Biomedical Research Centre (A.J.M., O.B., B.A., S.B., A.E., J.K.B., M.G.S. and J.C.K.); J.C.K. is also supported by a Wellcome Trust Investigator Award (204969/Z/16/Z); P.G.D. is supported by a Kennedy Trust Prize Studentship (Kennedy Trust for Rheumatology Research, KENN 20 21 02). G.L.S. and Y.L. are supported by BBSRC (BB/X011542/1) and the MRC (MR/W025590/1). M.C. and J.N.F. are supported by grants from the Medical Research Council (UKRI) and the Kennedy Trust for Rheumatological Research. The ISARIC4C investigators are listed in the supplementary information file. The views expressed are those of the author(s) and not necessarily those of the NHS, the NIHR or the UK Department of Health. This work uses data provided by patients and collected by the NHS as part of their care and supports #DataSavesLives.

## Author contributions

T.D. conceptualized the project. T.D. and Y.P. designed and supervised the study. A.J.M. supervised the clinical cohort. G.L.S. supervised the rVACV work. J.-L.C., B.W. and Z.Y. performed all the T-cell experiments. Y.L. produced the vaccinia virus. Y.P., G.L., X.Y. and E.A. performed Smartseq2 single-cell RNASeq experiments. A.J.M., O.B., B.A., S.B., A.E., J.K.B., M.G.S. and ISARIC4 Investigators established the clinical cohort and collected clinical samples and data. P.G.D., J.N.F. and M.C. established the MVA-BN clinical trial, collected clinical samples and data. J.-L.C., B.W., Z.Y., Y.P., M.P. and A.B. processed the clinical samples. D.J. and T.R. performed HLA typing. J.-L.C. and J.W. made the MHC-class I tetramers. C.W. and P.S. provided technical assistance with single-cell sorting. J.-L.C., B.W. and G.L. analysed T-cell data. E.A. performed single-cell data analysis. J.-L.C. and E.A. wrote the original draft. T.D., Y.P., P.G.D., J.C.K., J.N.F., M.C., G.L.S. and A.J.M. reviewed and edited the manuscript and figures.

## Competing interests

The authors declare no competing interests.

## Additional information

[1]Chinese Academy of Medical Science (CAMS) Oxford Institute (COI), University of Oxford, Oxford, UK. [2]MRC Translational Immune Discovery Unit, MRC Weatherall Institute of Molecular Medicine, Radcliffe Department of Medicine, University of Oxford, Oxford, UK. [3]Sir William Dunn School of Pathology, University of Oxford, Oxford, UK. [4]Centre for Human Genetics, Nuffield Department of Medicine, University of Oxford, Oxford, UK. [5]Oxford University Hospitals NHS Foundation Trust, Oxford, UK. [6]Kennedy Institute of Rheumatology, Nuffield Department of Orthopaedics, Rheumatology and Musculoskeletal Medicine, University of Oxford, Oxford, UK. [7]NDM Centre for Global Health Research, Nuffield Department of Medicine, University of Oxford, Oxford, UK. [8]Baillie Gifford Pandemic Science Hub, Centre for Inflammation Research, University of Edinburgh, Edinburgh, UK. [9]Roslin Institute, University of Edinburgh, Easter Bush, Midlothian Edinburgh, UK. [10]MRC Human Genetics Unit, Institute of Genetics and Cancer, University of Edinburgh, Western General Hospital, Crewe Road, Edinburgh, UK. [11]Intensive Care Unit, Royal Infirmary Edinburgh, Edinburgh, UK. [12]NIHR Health Protection Research Unit, Institute of Infection, Veterinary and Ecological Sciences, University of Liverpool, Liverpool, UK. [13]Sequencing Facility, MRC Weatherall Institute of Molecular Medicine, University of Oxford, Oxford, UK. [14]Flow Cytometry Facility, MRC Weatherall Institute of Molecular Medicine, University of Oxford, Oxford, UK. [15]Botnar Institute for Musculoskeletal Sciences, Nuffield Department of Orthopaedics, Rheumatology and Musculoskeletal Medicine, University of Oxford, Oxford, UK. [72]These authors contributed equally: Ji-Li Chen, Beibei Wang, Yongxu Lu, Elie Antoun, Olivia Bird, Philip G. Drennan, Zixi Yin. [73]These authors jointly supervised this work: Geoffrey L. Smith, Alexander J. Mentzer, Yanchun Peng, Tao Dong. ✉e-mail: tao.dong@ndm.ox.ac.uk

## ISARIC4C Investigators

J. Kenneth Baillie ⬡ [8,9,10,11], Peter JM Openshaw[16,17], Malcolm G. Semple ⬡ [12], Beatrice Alex[18], Petros Andrikopoulos[19,20], Benjamin Bach[18], Wendy S. Barclay[21], Debby Bogaert[22], Meera Chand[23], Kanta Chechi[19,24], Graham S. Cooke[25], Ana da Silva Filipe[26], Thushan de Silva[27], Annemarie B. Docherty[11,28], Gonçalo dos Santos Correia[29,30], Marc-Emmanuel Dumas[19,20,31,32],

Jake Dunning[16,33], Tom Fletcher[34], Christoper A. Green[35], William Greenhalf[36], Julian L. Griffin[19], Rishi K. Gupta[37], Ewen M. Harrison[28], Antonia Y. W. Ho[26,38], Karl Holden[39], Peter W. Horby[40], Samreen Ijaz[41], Saye Khoo[42], Paul Klenerman[43,44], Andrew Law[9], Matthew R. Lewis[29,30], Sonia Liggi[19], Wei Shen Lim[45], Lynn Maslen[29,30], Alexander J. Mentzer [1,4,73], Laura Merson[46], Alison M. Meynert[10], Shona C. Moore[12], Mahdad Noursadeghi[47], Michael Olanipekun[19,20], Anthonia Osagie[19,20], Massimo Palmarini[26], Carlo Palmieri[48,49], William A. Paxton[12,50], Georgios Pollakis[12,50], Nicholas Price[51,52], Andrew Rambaut[53], David L. Robertson[26], Clark D. Russell[22], Vanessa Sancho-Shimizu[54], Caroline J. Sands[29,30], Janet T. Scott[26,55], Louise Sigfrid[46], Tom Solomon[12,56], Shiranee Sriskandan[25,57], David Stuart[58], Charlotte Summers[59], Olivia V. Swann[60], Zoltan Takats[19,29], Panteleimon Takis[29,30], Richard S. Tedder[61,62], A. A. Roger Thompson[63], Emma C. Thomson[26], Ryan S. Thwaites[16], Lance CW Turtle[12,64], Maria Zambon[33], Thomas M. Drake[28], Cameron J. Fairfield[28], Stephen R. Knight[28], Kenneth A. Mclean[28], Derek Murphy[28], Lisa Norman[28], Riinu Pius[28], Catherine A. Shaw[28], Marie Connor[65], Jo Dalton[65], Carrol Gamble[65], Michelle Girvan[65], Sophie Halpin[65], Janet Harrison[65], Clare Jackson[65], James Lee[46], Laura Marsh[65], Daniel Plotkin[46], Stephanie Roberts[65], Egle Saviciute[65], Sara Clohisey[9], Ross Hendry[9], Susan Knight[66], Andrew Law[9], Gary Leeming[67], Lucy Norris[68], James Scott-Brown[18], Sarah Tait[66], Murray Wham[10], Gail Carson[46], Tassos Grammatikopoulos[69,70], Sarah E. McDonald[26], Victoria Shaw[71], Seán Keating[11], Cara Donegan[12], Rebecca G. Spencer[12], Chloe Donohue[65], Fiona Griffiths[9], Hayley Hardwick[12] & Wilna Oosthuyzen[9]

[16]National Heart and Lung Institute, Imperial College London, London, UK. [17]Imperial College Healthcare NHS Trust: London, London, UK. [18]School of Informatics, University of Edinburgh, Edinburgh, UK. [19]Section of Biomolecular Medicine, Division of Systems Medicine, Department of Metabolism, Digestion and Reproduction, Sir Alexander Fleming Building, Exhibition Rd, London, UK. [20]Section of Genomic and Environmental Medicines, Respiratory Division, National Heart and Lung Institute, Guy Scadding Building, Dovehouse St, London, UK. [21]Section of Molecular Virology, Imperial College London, London, UK. [22]Centre for Inflammation Research, The Queen's Medical Research Institute, University of Edinburgh, 47 Little France Crescent, Edinburgh, UK. [23]Antimicrobial Resistance and Hospital Acquired Infection Department, Public Health England, London, UK. [24]Department of Epidemiology and Biostatistics, School of Public Health, Faculty of Medicine, Imperial College London, 2 Norfolk St, London, UK. [25]Department of Infectious Disease, Imperial College London, London, UK. [26]MRC-University of Glasgow Centre for Virus Research, 464 Bearsden Road, Glasgow, UK. [27]The Florey Institute for Host–Pathogen Interactions, Department of Infection, Immunity and Cardiovascular Disease, University of Sheffield, Sheffield, UK. [28]Centre for Medical Informatics, The Usher Institute, University of Edinburgh, Edinburgh, UK. [29]National Phenome Centre, Department of Metabolism, Digestion and Reproduction, Imperial College London, London, UK. [30]Section of Bioanalytical Chemistry, Department of Metabolism, Digestion and Reproduction, Imperial College London, London, UK. [31]European Genomic Institute for Diabetes, CNRS UMR 8199, INSERM UMR 1283, Institut Pasteur de Lille, Lille University Hospital, University of Lille, 59045 Lille, France. [32]McGill University and Genome Quebec Innovation Centre, 740 Doctor Penfield Avenue, Montréal, QC H3A 0G1, Canada. [33]National Infection Service, Public Health England, London, UK. [34]Liverpool School of Tropical Medicine, Liverpool, UK. [35]Institute of Microbiology and Infection, University of Birmingham, Birmingham, UK. [36]Department of Molecular and Clinical Cancer Medicine, University of Liverpool, Liverpool, UK. [37]Institute for Global Health, University College London, London, UK. [38]Department of Infectious Diseases, Queen Elizabeth University Hospital, Glasgow, UK. [39]University of Liverpool, Liverpool, UK. [40]Nuffield Department of Medicine, Centre for Tropical Medicine and Global Health, University of Oxford, Old Road Campus, Roosevelt Drive, Oxford, UK. [41]Virology Reference Department, National Infection Service, Public Health England, Colindale Avenue, London, UK. [42]Department of Pharmacology, University of Liverpool, Liverpool, UK. [43]Nuffield Department of Medicine, Peter Medawar Building for Pathogen Research, University of Oxford, Oxford, UK. [44]Translational Gastroenterology Unit, Nuffield Department of Medicine, University of Oxford, Oxford, UK. [45]Nottingham University Hospitals NHS Trust:, Nottingham, UK. [46]ISARIC Global Support Centre, Centre for Tropical Medicine and Global Health, Nuffield Department of Medicine, University of Oxford, Oxford, UK. [47]Division of Infection and Immunity, University College London, London, UK. [48]Molecular and Clinical Cancer Medicine, Institute of Systems, Molecular and Integrative Biology, University of Liverpool, Liverpool, UK. [49]Clatterbridge Cancer Centre NHS Foundation Trust, Liverpool, UK. [50]NIHR Health Protection Research Unit in Emerging and Zoonotic Infections, Liverpool, UK. [51]Department of Infectious Diseases, Centre for Clinical Infection and Diagnostics Research, School of Immunology and Microbial Sciences, King's College London, London, UK. [52]Department of Infectious Diseases, Guy's and St Thomas' NHS Foundation Trust, London, UK. [53]Institute of Evolutionary Biology, University of Edinburgh, Edinburgh, UK. [54]Department of Pediatrics and Virology, St Mary's Medical School Bldg, Imperial College London, London, UK. [55]NHS Greater Glasgow & Clyde, Glasgow, UK. [56]Walton Centre NHS Foundation Trust, Liverpool, UK. [57]MRC Centre for Molecular Bacteriology and Infection, Imperial College London, London, UK. [58]Division of Structural Biology, The Wellcome Centre for Human Genetics, University of Oxford, Headington Oxford, UK. [59]Department of Medicine, University of Cambridge, Cambridge, Cambridgeshire, UK. [60]Department of Child Life and Health, University of Edinburgh, Edinburgh, UK. [61]Blood Borne Virus Unit, Virus Reference Department, National Infection Service, Public Health England, London, UK. [62]Transfusion Microbiology, National Health Service Blood and Transplant, London, UK. [63]Department of Infection, Immunity and Cardiovascular Disease, University of Sheffield, Sheffield, UK. [64]Tropical & Infectious Disease Unit, Royal Liverpool University Hospital, Liverpool, UK. [65]Liverpool Clinical Trials Centre, University of Liverpool, Liverpool, UK. [66]Public Health Scotland, London, UK. [67]Centre for Health Informatics, Division of Informatics, Imaging and Data Science, School of Health Sciences, Faculty of Biology, Medicine and Health, University of Manchester, Manchester Academic Health Science Centre, Manchester, UK. [68]EPCC, University of Edinburgh, Edinburgh, UK. [69]Paediatric Liver, GI & Nutrition Centre and MowatLabs, King's College Hospital, London, UK. [70]Institute of Liver Studies, King's College London, London, UK. [71]Institute of Translational Medicine, University of Liverpool, Liverpool, Merseyside, UK.

