## [Transparent Peer Review file · Nature Communications]

T cell memory response to MPXV infection exhibits greater effector function and migratory potential compared to MVA-BN vaccination

Corresponding Author: Professor Tao Dong

Version 0:

Reviewer comments:

Reviewer #1

(Remarks to the Author)

General remarks

The present manuscript by Chen et al. has important and interesting aspects. It employs a useful assay (ELISPOT with VACV-infected cells) as a readout to analyse virus-specific T cell responses ex vivo. Some responses in convalescents were surprisingly low; it would have been interesting to correlate response strength with available information on patient cofactors such as HIV infection.

It is also interesting and informative that the authors directly compare specific immune responses in MPOX-infected and MVA-vaccinated individuals, although those differences between these responses which they identify based on single-cell analysis alone, and which they strongly emphasize in their presentation, might not be sufficiently robust.

A focused analysis of epitope-specific T cells was undertaken, but limited to a single A2 epitope and to a small number of subjects. Direct quantitation of epitope-specific T cells was limited to expanded T cell cultures, although it should be expected that several immunodominant VACV/MPXV epitopes are expanded enough in order to quantitate them directly ex vivo by tetramer/multimer staining. Well-characterized epitopes with other HLA restrictions that might be of higher importance than A2 in populations other than those of European descent (but also of importance in Europeans) would have been available. The authors could also have been clearer in their acknowledgment that their chosen epitope was identified in dominant MVA vaccination responses in humans already in 2005 (ref. 25). Generally, they could have provided more detailed literature context for a discussion of MVA vaccination responses.

The scRNAseq analysis of T cells specific for this epitope is unfortunately hampered by small cell number and lack of information on input cells, and thus unlikely to be representative. It would be much better in general, in studies in this field, if such single cell analyses would be undertaken in a sufficiently comprehensive and documented way, and if their most important results were further examined using independent methods.

The comparison of MPOX and MVA vaccination responses to SARS-CoV-2-specific T cells is hard to interpret as long as it is not even stated what kind of (and how many) SARS-CoV-2-specific T cells were studied.

The manuscript is hampered by sometimes contradictory and incomplete information regarding methods and results, as specified below in detail (for which I apologize, but I do not know how to present my argument more succinctly).

Unfortunately, the interpretation of results as shown in Title, Abstract, and Discussion tends to make multiple statements that are insufficiently substantiated by the data, highly speculative, or incorrect, as explained below. Here I mention one example: the claim that T cells after MPOX infection show "greater migratory potential" than T cells after vaccination. This is entirely based on the single-cell analysis that was made, on the side of the vaccinated cohort, with only 160 cells in total, of unstated purity, from an unstated number of vaccinated donors in unstated proportions. In my opinion, generalizations and main messages of a paper should not be based on such numbers, and the authors' neglect to provide available background information is worrying. That said, the authors' interpretation does not even fit their data from this experiment, since

chemokines, which are paradigmatic "migratory" molecules, were more highly expressed after vaccination than MPOX infection according to Fig. 4B (but not according to some of the author's descriptions in the text). In contrast, certain integrins were more highly expressed after infection than vaccination, which may be the reason why the authors sometimes merge "integrins and migratory" genes together in a way that is not obviously justified and makes the discussion less clear.

Since the single-cell experiment is unlikely to be robust, I would suggest to eliminate it altogether. It could only be rescued if some of its main findings would be confirmed using robust methods such as analysis of protein expression.

Specific remarks in order of appearance

Results

Table 1 provides sample collection time as days post symptom onset, which is helpful. It would also be informative to add information for "days post symptom resolution" or duration of symptoms, if that is available.

Fig. S1 is visually pleasing, but it would be more informative to provide a table with HLA types per individual, such that HLA associations, homo-/heterozygosity etc. might be inferred and all allotypes can be provided. E.g. 1/3 of HLA-B is "other", which is a large information gap. The factual content of the figure is unclear as well: it is supposed to show the HLAs of 13 convalescent participants, so $n=26$ for alleles of a given locus. However, several of the diagrams have segments far smaller than 1/26th of a full circle. If more is known about the HLA type (HLA-C, DP, DRB3/4/5, DQA, DPA) that info might be provided, too.

line 113, "we set up an ex vivo IFN γ ELISpot assay by stimulating PBMCs with "the VACV strain Lister as previous [sic] described (18)" - I cannot find an ELISpot assay with VACV in the cited paper. Please make the statement more precise.

Fig. 1E legend, is $n=11$ or 12 here instead of $n=13$? Please state in text or legend. Were some convalescent individuals excluded because of sample availability or technical problems?

Fig. 1F contains the proportions of CD4/CD8 cells for $n=12$. It contains these proportions both for high and low T-cell responders, although the proportions would probably be more robust for high responders (there were two near-zero or zero responders for CD8 and CD4 each in Fig 1E). Please add some number to each bar of Fig. 1F that indicates the overall size of response to make the readers' assessment easier, or scale the bars according to response size. Adding lines that indicate paired values in Fig. 1E would fulfill a similar purpose.

(please correct y-axis spelling for Fig. 1F)

Did these reconvalescents still have elevated overall CD8/CD4 ratios in their peripheral blood as described by others?

line 139, "were selected". However, the Methods part suggests that all available epitopes were included. Is that correct, or if not, what were the selection criteria? The methods part (line 605) speaks of experimentally verified epitopes. IEDB makes the important distinction between T cell assays and MHC ligand assays. Were only epitopes with available T cell assay information included (an approach I would prefer)? Please clarify. It might be contested that peptides for which only MHC ligand assays but not T cell assays are available are "experimentally defined" T cell epitopes.

Please provide a reference for the categorization of antigens as early, intermediate, late.

Please use consistent terminology (intermediate/late vs. late).

line 142, "The remaining 22% of CD8 and 28% of CD4 epitopes corresponded to VACV epitopes that contained mutations in MPXV genome" - i.e. non-synonymous mutations, since we are at the amino acid level? It's possible that the 78% and 72% epitopes that were conserved at the aa level had synonymous mutations.

Line 162 and Fig. 2C, were these antigen-specific responses partially complementary? How many donors recognized at least one of these 5 antigens?

line 181, "HLA-A*02:01 is the most common HLA allele in human populations". This statement is not covered by the cited reference, and is incorrect as stated. The most common classical HLA alleles are probably HLA-DRA*01:01, or possibly DPA1*01:03 among the more polymorphic ones. If we limit consideration to classical HLA class I alleles, the statement seems correct for people of European descent but not others. Other candidates for the most frequent 4-digit class I allele worldwide are A*11:01, A*24:02, or HLA-C*07:02, which are much more frequent than A*02:01 both in India and China.

How many A2 or A*02:01 epitopes were individually tested in ELISpot assays, if any, and in how many donors? The text in lines 179-195 seems unclear in that respect. Subsequent text suggests that ex vivo ELISpot was done and short-term T cell lines were generated with peptide pools, not individual peptides. Please specify in line 187f. In that case, statements about immunodominance need a disclaimer, since not all epitope-specific subpopulations may expand equally well in vitro in such T cell lines.

line 182, "T cell responses to 71 A*02:01-restricted early epitopes and 69 intermediate/late epitopes" - This statement suggests that all A2-restricted epitopes in your pools are A*02:01-restricted epitopes. Were all of the A2 epitopes described to be A*02:01-restricted in IEDB or the literature sources? The legend to Fig. 3 speaks of "A2-restricted" responses. Please make the terminology consistent, even if it may seem tedious to always write "A*02:01".

line 187ff, please indicate whether all the peptides mentioned here were conserved between VACV and MPXV.

line 195, "all three participants tested showing a strong response to this epitope", but Fig. 3C shows ILD tetramer-specific T cells in four donors - supposedly four T cell cultures/lines - and one of them did not show a "strong" response. Please make it clearer in the text and legend where peptide-stimulated T cell lines were used for analysis. Fig. 3C shows tetramer vs same tetramer, and both correlate well, which is usually expected - such an experiment can only control for fluorochrome-based background staining but not other forms of background or interference. Another plot showing tetramer vs independent marker (e.g. CD8) or dual staining with different tetramers may give a better impression of potential background in this assay.

Please indicate the donors shown in Fig. 3D.

Fig. 3D, it is unexpected that there were many more cells staining for TNF-alpha than IFN-gamma. This is unusual for virus-specific CD8 T cells. The authors seem to disregard this finding by sweepingly saying that "all" of the T cell lines were "expressing multiple cytokines (IFN γ , TNF α , and IL2)". Is there an explanation for this unusual finding, was it previously observed for VACV or poxviruses? Or was there confusion about the fluorochromes used? Fig. 3D, upper part, shows IFN-gamma PE/Cy7, but the Methods section says that IFN-gamma FITC and IL-2 PE-Cy7 antibodies were used for ICS. This seems contradictory. Some other fluorochromes do not match either.

Fig. 3F, please show killing of the non-infected LCLs as a control, if the data exist. It is curious that the highest killing was observed in the T cell line with the lowest number of ILD-specific T cells (Mpx005). Why is that? What were possible confounders in this experiment?

Does Fig. 3F indeed show killing of infected cells as stated in text (line 203) and legend? The methods part mentions killing assays only in the context of "BCLs" (usually abbreviated LCLs or B-LCLs) loaded with high concentrations (10 μ M) of peptide. It makes little sense to label the diagrams in Fig. 3F with the abbreviation of a single peptide (ILY, ILD) if the effectors were in fact polyspecific T-cell lines previously stimulated with multiple peptides. What was done here?

line 208, "Given that we identified functional differences in MPXV-specific CD8+ T cells in convalescent individuals", what does this mean? Heterogeneity among convalescent individuals?

What was the time interval between the two doses of the MVA-BN vaccine? Cannot find this in the text or the graphics in Fig. 4A.

line 213, it may not be obviously correct to call this a "memory" T cell response if T cells were analyzed shortly (4 weeks) after vaccine doses.

I wonder why the differences in Fig. 4B were not significant. What were the p values, what test was used? Are the results non-significant in parametric and non-parametric tests?

line 218, "there is a slightly stronger T cell response" - The effect was not statistically significant but quantitatively strong: the median response after infection was apparently more than twice as high than after the first or second vaccine dose. It's interesting that a strong booster effect of the second dose was observed, but only in a minority of subjects. Is it possible that they were more likely to be immunocompromised? Who were the vaccinated individuals, what were the inclusion criteria? Table S1 tells us they were all-male with a median age of 36 and only two people over 50. Comorbidities or other factors are not listed. Might some of them have been HIV-positive, as were two of the 13 reconvalescents? What would be the outcome of this analysis if HIV-positive subjects were excluded, if that is possible?

Two high responders to the second dose in Fig. 4B stood out. Did these subjects overlap with those who had had a smallpox vaccination in their youth?

line 234, "with a significantly greater level of tetramer+CD45RA+CCR7-CD8+ T cells 28 days after a second vaccination compared to convalescence" - This contradicts Fig. 4D,E and the text above. The only significant difference between the two cohorts was found for KLRG1+ cells, according to Fig. 4E.

The scRNAseq experiment is unsatisfactory. On the vaccination side it contains only 160 cells from an unstated number of donors in unstated proportions, sorted with tetramers to an unstated purity. It seems that such experiments are now so fashionable that they have become mandatory in studies of this type. At the same time, they are performed with so much data processing and normalization procedures that small differences in gene expression may be represented as large ones, subclusters may be categorized in an arbitrary number of ways, etc.

It is stated that cells were "integrated based on patient of origin", or, as stated in Methods, "Patient-specific cells were integrated using Harmony v.1.0 to remove batch effects." Are the batch effects referred to here the differences between patients? Why is it considered valid to remove or reduce these differences by mathematical processing? This might be adequate for batches of samples that should be expected to be biologically equivalent. But this is not the situation when a variety of human subjects is studied.

The populations were divided into four clusters, but these clusters were not clearly separated in the UMAP plot (Fig. 5A). So I wonder how distinct these clusters actually are, and if alternative clustering would have been similarly justified.

Fig.5D legend says "cluster 0 marker genes" but Figure and main text suggest these are cluster 2 marker genes.

Please make it clearer in this section that the cells being analyzed in scRNAseq are specific for one epitope, and mention the epitope in the Results text (e.g. line 240, 258) and Fig. legends.

As the authors state, many clonotypes were identified in just one cell each (line 258ff). But that would imply that this analysis does not provide us with robust information about the clonal TCR composition and repertoire complexity of these samples.

The experiment lacks controls, so we cannot say which of the low-frequency TCRs may be abundant TCRs specific for something other than the MPXV epitope that ended up as contaminants in the tetramer-sorted cells. The purity of sorted cells isn't stated either.

However, it seems incorrect if the authors say that TCR alpha clonotypes from vaccinees were "all identified in just one cell" (line 265), compare Fig. 5F-G. Some clonotypes were clearly more frequent than others.

The statement in line 270 suggesting that even public clonotypes (found multiple times in this experiment) may in fact be specific for SARS-CoV-2 dramatically invalidates this experiment.

Line 271 refers to Table 2 supposedly comparing clonotypes found here to SARS-CoV-2-specific ones. I cannot find a Table 2 in the materials submitted.

Relating to Fig. 6, what kind of SARS-CoV-2-specific T cells were studied in these experiments? Cells specific for a particular antigen or epitope, and which one? I cannot find this in Results, Legend or Methods.

Regarding Fig. 6A, it would be good to see a few control genes (expected to be expressed or not expressed in CD8 T cells, such as CD3, CD45, CD8, TCR, etc.) in order to judge the significance of this analysis and the granularity of the results. What is the real quantitative difference of expression of the genes mentioned in line 284-291? Fig. 6B seems to suggest that highly significant group differences (****) can result even if there are very minor shifts of average expression patterns. Regarding Fig. 6C, please indicate the total number of genes that was studied in the experiment, and please apply a statistical correction for multiple testing before you report highly significant p values for 12 selected genes. It would have been helpful to verify differential expression of these genes by FACS or another direct method, preferably measuring protein expression.

The summarizing paragraph (line 311-314) compares "MPXV-specific memory T cells in convalescence" to something, but to what? For example, these cells are said to have "increased expression of migratory molecules", but Fig. 6B seems to show that their expression of chemokines is lower than SARS-CoV-2-specific cells and lower than in cells MVA-vaccinated subjects. In line 296ff there's still another statement, but that statement seems to combine cytokines and chemokines in some way; nonetheless it contradicts the chemokine panel in Fig. 6B.

Discussion:

line 335, "This finding contradicts recent reports that shows a greater persistence of CD4+ T cells compared to CD8+ T responses in MPXV convalescence (12,14)." How does ref. 14 (Cohn) show greater persistence of CD4+ than CD8+ T cells, and what is meant by persistence? It seems to me that CD8+ T cells were as frequent or more frequent than CD4+ T cells in ref. 14 - however that paper might not have studied enough donors to even make such an assessment.

line 337, "In our study, immune responses were measured against the whole VACV, resulting in a with broader [sic] epitope coverage compared to peptide-based assay." This is by no means clear and would require proof. In a cell infected with a very large virus such as VCAV, there may be competition between epitopes for presentation depending on various factors (HLA affinity, efficacy of processing, expression kinetics) and the peptide pools were also very complex (even called "mega pools" by the authors).

line 339, "lower response" compared to what?

line 346, "similar results seen in mpoX" - this statement is entirely unclear

line 353, Such cells are also very prominent in CMV infection.

line 354, "These cells were shown to exhibit enhanced proliferation and differentiation potential in vivo." Compared to what? Please make a precise statement and provide a reference.

line 359, "suggests that in convalescence, long-term MPXV-specific central memory T cells with high cytotoxicity may be generated." This statement is surprising since CCR7-CD45RA+ was by far the dominant phenotype, and this is not a central memory phenotype. There were few CCR7+CD45RA- T cells. Moreover, classical central memory T cells are not of high cytotoxicity. This seems to be an arbitrary and speculative statement that confuses different subsets.

line 365, "With a more diverse TCR repertoire, T cells after vaccination may be able to contain variants more effectively." As the authors partially say themselves, and I explain further above, it is not clear which of these TCRs are even VCAV/MPOX-specific.

line 393, "Our data showed that MPXV infection could elicit longer-term memory T cells with greater effector function and migratory potential to the site of infection, whereas T cells induced by MVA-BN vaccination may contain variants more effectively and show broader cross-reactivity to other orthopoxviruses due to their diverse TCR repertoire." The present data do not substantiate such a statement. T cells were studied only at one time point after the second dose of vaccination (about 4 weeks), and only at one time point after infection per patient. Timing of the two groups was not matched. Thus speculative comparisons such as "longer-term" cannot be substantiated. The claim about broader cross-reactivity to other orthopoxviruses is not substantiated either. I have already commented on the TCR repertoire.

line 407, "at a similar time point post antigen encounter." The time point was not similar. It was 4 weeks after the second dose vs. a median of 105 days after symptom onset.

line 416, "While there was no evident effect of this in our data" - since the question was not studied? Please provide information, as available, about the immune response observed in the two individuals with prior smallpox vaccination, compared to the two others. It would be easy to use different symbols for these two individuals in Fig. 4B, for example. You might also provide more information or evaluation of the MPOX convalescent cohort. It is of interest that each of the three MPOX convalescents older than fifty years had mild infection without fever (Table 1). Might some of these have received a smallpox vaccination?

Methods:

line 650, please add to the Methods part info about the incubation medium for overnight stimulation - that's important since FCS/FBS can stimulate strong background T cell responses. The cited reference (19, Mateus 2021) does not provide this info for the AIM assay. Please indicate as well whether the cryoconservation medium contained FCS.

line 653, "BD human FC block (Biolegend, 422302)" seems contradictory

line 658, "A list of antibodies used in this panel can be found in table S2" - that's Table S3, and it only lists target antigens, it does not indicate fluochromes and combinations of antibodies used in panels. Please provide the FACS panel(s) and combinations of antibodies that were actually used in this assay, with fluorochromes and clone identification.

line 663, the stimulation index (SI) mentioned, is this the ratio of signal to background?

provide info Table S1, NA means information not available or not applicable? Is there a chance that some of the NA subjects had a smallpox vaccine but that is unknown?

Fig. S3, only limited gating information is shown

line 667f "T cells lines", "T cell line" -> T cell lines

line 669, what is H10?

Table S3, why are some epitope sequences printed in light grey font? The footnote says "**bold and underlined residues indicate substitutions from the VACV genome to the MPXV genome**" - please state explicitly that it is the MPXV sequence that is displayed. For those peptides with divergent sequence, please provide the VACV sequence as well (for easier reference), since some peptides are heavily polymorphic among the two viruses.

Reviewer #2

(Remarks to the Author)

Chen et al. investigated in their manuscript 'T cell memory response to MPXV infection exhibits greater effector function and migratory potential compared to MVA-BN vaccination' T cell responses in mpox convalescence and MVA-BN vaccinated individuals. The authors found that magnitude of mpox-specific T cells are comparable between convalescent and MVA-BN vaccinated individuals, but that the T cells in convalescent patients are phenotypically different that from vaccinated individuals, as for example, have a more cytotoxic and migratory profile. These findings are in line with reports from SARS-CoV convalescent and vaccinated individuals.

Due to the recent outbreaks of mpox, this study is of high importance for the scientific community.

However, some major points need to be addressed to clarify parts of this manuscript and make it more understandable.

1. Cohorts:

The authors have described the cohorts of n=13 convalescent mpox individuals (Mpx001-13) well, including basic information for the healthy cohort (HC). Regarding the vaccination cohort, the basic characteristics of this cohort (supplementary material) are not connected with the main text (starting line 105). The authors need to link this information. Neither the methods section nor the results section of the manuscript mentions the characteristics of the SARS-CoV-2 cohort used and how the data were generated (Figure 6). The authors need to clarify where this data comes from. If this data comes

from a publicly available dataset, authors need to clearly state this so that the reader can understand their workflow. This is especially important considering that they have integrated this data into the mpox data (figure 6). E.g. which SARS-specific Tetramer was used for cell isolation? Are the authors sure that this potentially publicly available dataset was generated the same way as the authors analyzed their scRNAseq data?

In addition, there is no consistency in the manuscript in naming the cohorts in the figures. Sometimes convalescent and vaccinated and sometimes sometimes MPXV and MVA-BN was used. The authors need to streamline this.

2. The authors need to streamline the results part, as in some parts they even included detailed methods sections (e.g. line 112-114) and in some parts they barely describe how they came to their results.

3. The authors need to explain in more detail what the FEC peptide mix contains.

4. In line 110-130, the authors describe their results from the ELISPOT using VACV infected PBMCs to investigate the MPXV-specific response. Followed by an AIMs assay. It is not clear why the authors performed this AIM experiment. Wouldn't a flow cytometry analysis with a read-out of IFN γ + CD4 or IFN γ + CD8 T cells be more specific? The authors need to clarify this.

5. In figure 1, the authors need to check whether the results presented in figure 1e and f are correct. It seems they do not match. In figure 1E they show two individuals without CD8 response, while in figure 1f only one individual (Mpx008) is shown.

6. For the experiment shown in figure 2, for example, it is not clear how the data shown in 2b was generated for CD8 early antigen peptide pool. Are both mega-pools combined as one stimulus (A2 early plus non-A2 early) or are the data are generated by separately stimulating PBMC with one of the pools and then adding them together? In other words, did the authors stimulate the PBMCs with the peptide pools separately or together (71 or 90 peptide containing pool or 161 peptide containing pool)?

7. Are the data shown in figure 2B from stimulated sorted CD4 and CD8 T cells? Or are the authors assuming that they are CD4 or CD8 responses based on the predicted epitopes used? This could be misleading.

8. In figure and figure legend 2, the authors need to indicate what the x-axis means in figure 2c, and what the red lines mean in figure 2d. It would also be helpful if the authors would emphasize that the convalescent cohort was used to generate the data.

9. In line 153-155: The authors need to report the mean + SEM or SD values before the p-value. In addition, this sentence is linked to figure 2b, where 7 (CD8) to 3 (CD4) responses were compared, which is borderline for statistical analysis. The authors should also check if the reported n=13 for figure 2b is correct, since less than 13 dots are shown in the graph.

10. The authors should indicate that the convalescent cohort, if correct, is used from line 177.

11. The authors must indicate in figure legend 3 how many HLA-A2 individuals were used for this study, please.

12. If the identified epitopes (line 191-194) have not been previously described, the authors should highlight this in the abstract.

13. Figure 3B: it is not clear whether the 'magnitude' is a result of pooled responses from all investigated individuals. Please clarify.

14. Figure 3B, if n=6 donors are investigated, I do not understand who a frequency of 25% responders can appear. Or are only 4 individuals investigated?

15. Figure 3C: Why did the authors used double staining with PE and APC labeled tetramers of the same peptide?

16. Figure 3D: The response to SLS in the VACV-stimulated A2-positive individual shows a nice memory recall response, however, for ILD it looks like an unwanted extremely dominate TNF α response after restimulation. Do the authors have an explanation for this?

17. In figure 4 the authors need to show the frequency of tetramer+ mpox-specific T cells for all investigated individuals.

18. Figure 5: The labeling of the clusters are not readable in A. For D, the mean is not visible in the violin plots. Please correct this.

19. The authors need to rearrange the order of the graphs in figure 6b according to the text.

20. It is not traceable how the authors come to '93% of individuals with...' in line 324. The authors need to explain this.

21. In line 325, the authors mentioned that an ELISPOT to detect mpox-specific T cell responses might be interesting for diagnosis. This statement seems to be a bit ambitious. This assay is time consuming and does not indicate the clinician if the patient is acutely infected or convalescent. The authors need to down-tune this statement or need to explain this in more

detail.

22. In general, the discussion needs to be strengthened, since some parts are hardly discussed whereas others are too long. Especially, the section (line 332-343) needs to be rewritten. It is not clear what the authors want to state here, especially in the light that they also performed many data with peptide pools.

23. Line 355: The term ex in exKLRG1 is not introduced. Please do so.

24. The manuscript would improve by confirming some of the transcriptome data by proteome analysis (e.g. flow cytometry).

25. Regarding the described migration profile of the mpox-specific T cells, analysis from skin lesions might support the findings.

26. Figure Legends: The authors need to check their figure legends. In general, often no dots or comma are included in the figure legends, makes it hard to read. In addition, the authors need to report the number of individuals investigated in each graph as well as the investigated number of cells (e.g. 6c) and they need to make sure that the numbers mentioned in the figure legend are in to the individual dots shown in the graphs. In this regard, the number of dots displayed e.g. in figure 1c for Mpox do not match with the indicated number in the figure legend (n=13).

27. Figure: Fine tuning of the figures would dramatically improve the manuscript. This includes bigger dots in the graphs like fig 1c, more distance between different graphs and labeling like fig4c and d as well as in figure 3c.

28. Supplementary tables: The formatting of the uploaded supplementary tables are disrupted.

Reviewer #3

(Remarks to the Author)

The manuscript by Chen et al described a study of convalescent monkeypox virus (MPXV)-infected and MVA-BN (a new small pox vaccine) vaccinated individuals. The authors found substantial CD8+ and CD4+ T cell responses that shared similar differentiation and activation phenotypes. scRNAseq analysis of convalescent individuals seems to have a somewhat increased cytotoxicity and migratory potential. Given the recent outbreaks and relatively little knowledge on human T cell immunity to MPXV infection the current work provides additional insight. Overall the work has been performed well but validation of the main conclusions based on only the scRNAseq data are required.

Comments and suggestions.

-The main issue with this manuscript is the claim that the functional profile (with respect to cytotoxicity and migration) of MPXV-specific memory T cells induced by natural infection is claimed to be "better" than the vaccine (Figure 6). This conclusion is based on a limited set of scRNAseq data (only 161 cells of the vaccination group) and no validation at the protein level or by functional assays (e.g. by flow cytometry, cytotoxic and migration assays). In fact, the assays performed as displayed in Figure 4, indicated no differences in cytokine profiles and if anything a only a different phenotype with respect to KLRG1 but not to other markers. As such the conclusions are currently based on scarce data, and may therefore not reflect actual differences or resemblance.

Maybe needless to emphasize but whether (functional) differences between T cells elicited by natural infection or vaccination are significant or insignificant the results are of interest and contribute to improved understanding.

-Further understanding could be from determining TCF-1 levels as described by Adamo et al, 2023. The TCF-1 levels could give insight into longevity/stemness of the viral-specific T cell population.

Minor: Abstract: explain the abbreviations MPXV and MVA-BN in the abstract for clarity and instant understanding.

Version 1:

Reviewer comments:

Reviewer #1

(Remarks to the Author)

The authors have undertaken a thorough and thoughtful revision of their manuscript, and have clarified many points. However, some points of contention remain, some of which I consider critical (as indicated).

Responses to the authors' rebuttal

R10, I am not sure if the sample should really be excluded due to high background. If background is as high or higher as the signal in the presence of antigen, it might be better to include this in the dataset as a zero result. If this was a single and strong outlier in terms of background, exclusion might be justified though.

R11-1, I thank the authors for their response and additional information. Sorry I didn't manage to make it clear what my question was. My point was that if a low absolute number of specific T cells was obtained, then the ratio of CD4 to CD8 is less certain. So the different CD4/CD8 ratios may have a very different degree of certainty that is not obvious from this kind of graphical representation. I wanted to suggest that the absolute number of T cells is indicated for each sample in Fig. 1F, in order to make it easier to judge the certainty/precision of each of these CD4/CD8 ratios.

R16-1, In their reply here the authors now provide the important information that 7/12 donors recognized at least one of the antigens tested here. Conversely, that means that 5/12 donors did not respond to any of the antigens. This seems important information and should be included in the text. It will help readers (such as me) understand Fig. 2B correctly.

By "complementary" I meant that some donors not responding to E antigens might respond to int./L antigens instead or vice versa. The authors' response makes it clear that this is not the case. Since 7 donors had a CD8 response to E antigens as already stated in the text, donors responding to int./L were a subset of donors responding to E.

R17, I thank the authors for these details. Sorry I was nitpicking so much about this HLA frequency issue. But it is still not correct to say that HLA-A*02:01 is the most frequent HLA allele in Caucasians, HLA-DPB1*04:01 is more frequent. However, it would be correct to say that HLA-A*02:01 is the most frequent HLA class I allele in Caucasians.

R18, the revised text 194ff is a bit unclear, "T cell responses to 71 A*02-restricted early epitope pool and 68 intermediate/late epitope pool", I think this means "T cell responses to a pool of 71 A*02-restricted early epitopes and a pool of 68 A*02-restricted intermediate/late epitopes"?

R19, I understand that the authors wished to test all epitope candidates reported to be restricted through A*02:01 or "A2", and that makes sense to me. I'm not sure though whether they included epitopes that were known to be restricted through another A2 allele such as A*02:03 but not through A*02:01. This is just an aside.

However, in line 674 they still write "We further divided the CD8 peptides into HLA-A*02:01 and non-HLA-A*02:01 peptide pools", which is inconsistent with their response. Please state the criterion for this "division" clearly here. Please check again if statements about A2 vs A*02:01 are consistent and error-free.

R22 (critical) The reason for my skepticism is that I have repeatedly encountered such results (TNF α + cells >> IFN γ + cells) in large and important datasets from partners, and in each case these observations were demonstrably due to experimental errors (and did not match a preponderance of results in the literature). In one case the problem was FACS compensation, in another it was problems with separation of fluorescence signals in multicolor ELISPOT. Therefore I am very wary of this specific issue.

I thank the authors for providing several relevant references that claim %TNF α + > %IFN γ + for virus-specific T cells ex vivo. I have checked Shahbaz 2021. The IFN γ staining in that paper (Fig. 2A) seems weak, smear-like, and substandard, even in the anti-CD3/28 control. Bad antibody, staining protocol, or compensation? Information about fluorochromes is not even provided in that paper. Those authors' job might have been a bit more difficult since some of their patients were acutely infected, severely ill and in ICU, which may lead to suboptimal T cell quality or generalized activation. In the present manuscript, however, patients were convalescent.

In the case of the present manuscript, I am worried by the unusual appearance of the populations in Fig 3D (TNF vs IFN), and that there is not a single IFN γ + TNF α - cell anywhere, although such cells are regularly observed in diverse viral infections. The one sample with significant numbers of IFN γ + cells shows only double-positives in some diagonal staining pattern. I am not convinced everything is OK with these FACS analyses and their compensation. Unfortunately they do not show a full set of, or any, cytokine-vs-cytokine stainings in their gating example in Fig S3B, they just show the higher-level gating. However, correct compensation is just as important as correct gating, and usually more difficult.

In the authors' earlier publication (Yin 2023), there are many diagonal distributions of cells in cytokine-vs-cytokine plots, which are especially clear in a combination of the same fluorochromes as here (PE-Cy7 vs APC, Fig. 3C in Yin 2023). Although in that earlier population the antibodies were IL-2-PE-Cy7 vs TNF-APC, the distributions are actually very similar to the ones the present manuscript for IFN-g-PE-Cy7 vs TNF-APC: the double-positive cells are entirely found in a diagonally-shaped population, and there are no PE-Cy7 single-positive cells.

Therefore I strongly suspect a problem with fluorescence compensation. The authors should try and see what happens if they modify their compensation settings. I understand this might be tedious. But it seems important to me that fundamental functions of T cells are not erroneously reported. Such errors seem to be arising more and more often in the literature.

Reviewer 2 agrees with me that this seems to be an issue.

R23, I thank the authors for clarifying their killing assay, which makes sense to me.

R26, There is a difference between "28 days" (revised Fig. 4A) and "no less than 28 days" (revised Methods). The new preprint cited actually says it was between 26 and 40 days in the cohort. Please be so kind and report this correctly here for the subset of donors studied here.

R32-1 (critical) If authors insist to show and interpret these single-cell results from as few as 160 cells after vaccination, they should at least make it very clear how many donors were involved in these 160 cells, and how many cells were from each donor. I think it's unacceptable that this information still seems not to be provided. My apologies if I overlooked anything. I also request an UMAP plot such as in Fig. 5A but color-coded by donor. At least there should be full transparency about the origin of these cells, and about the impact of their donor origin on subset distribution.

R32-1, R37 (critical) In spite of their claim in their replies, authors have not provided proof that their tetramer-sorted cells are 100% pure. 100% purity is a strong claim, and it seems the authors have not added this explicit claim to their manuscript? Fig. S3 does not prove that. Indeed, the CD8+CD3+ gate contains both CD8-high and CD8-low-positive cells. The tetramer+ gate contains tetramer-high and tetramer-dim cells and even a few cells that are both tetramer-dim and CD8-dim. Such cells might be more likely to be something non-specific. A much stricter gate might have been used for sorting in this example. Some control staining that shows tetramer background in non-immune donors would be helpful. Tetramer staining sometimes does have significant background. Can original data be provided that substantiate 100% purity?

The frequency of specific cells is not indicated in this example in Fig. S3; it might be even more difficult to sort epitope-specific CD8 T cells to high purity in donors with very low frequency of tet+ cells (see new Fig. 4D). Proportions of tet+ cells were at 0.01% or lower in 4 of the 6 vaccinated donors in that figure. How high were these numbers? Can a log plot be provided, or the actual numbers?

Considering Fig. 4D, we can't even exclude that group differences in single cell analysis between multimer-sorted single cells from convalescents and vaccinated donors are due to different degrees of contamination by non-specific T cells, because responses in some vaccinated donors are so much lower than in convalescents.

Other

Line 353 has a new claim, "Furthermore, we identified 7 HLA-A*02:01 restricted MPXV-specific epitopes where six of them are conserved between VACV and MPOX, one is specific for MPXV" - "we identified epitopes" is usually understood to mean that they were not known before. Please modify.

Reviewer #2

(Remarks to the Author)

The authors have sufficiently answered most of my questions/comments.

Regarding the Sars-Cov2 data from Peng et al Nature Immunology 2022, I would ask the authors for a statement again.

C1: In the paper 'Peng et al Nature Immunology 2022', scRNAseq data from 4 patients are shown. Figure 2 'a, Gene sets scored based on single-cell gene expression from a SmartSeq2 RNA-seq dataset comprising two mild and two severe convalescent HLA-B*07:02-positive patients with COVID-19 (n = 208 cells from mild cases, n = 140 cells from severe cases).' The authors should comment on how they arrive at 10 individuals. Furthermore, 912 cells from 10 individuals means a mean of 90 cells per individual. Can the authors discuss the limitations and problems arising from these low cell numbers and the heterogeneity between individuals (batch effect)?

In addition, it would be good if the authors would insert the number of vaccines and convalescent individuals into the figure caption of Figure 5.

Reviewer #3

(Remarks to the Author)

The authors have addressed my comments and suggestions.

Version 2:

Reviewer comments:

Reviewer #1

(Remarks to the Author)

I thank the authors for their additional thoughtful and informative comments.

R8: I thank the authors for their patience in dealing with my inquiries and for providing additional explanations and examples of stainings with peptide-stimulated cells, which seem to me to demonstrate adequate compensation between TNFa-APC and IFNg-PE-Cy7.

R11: Thank you for this graph, which suggests that the cells are roughly equally distributed in the UMAP space independent of their donor origin, and that similar numbers of cells were analyzed within each donor group. I think the graph is informative and should be included in the paper, not just in the rebuttal letter, maybe as a supplement. This would strengthen the message in spite of the relatively small number of cells. I was asking my question in behalf of future readers of the paper, not just for my own information.

R13: Thank you for this graph. I think it should be shown in the paper, e.g. replacing Fig. 4D, not just in the rebuttal letter.

Reviewer #2

(Remarks to the Author)

The authors answer all my questions satisfactorily.

REVIEWER COMMENTS

Reviewer #1 (Remarks to the Author):

We appreciate the reviewer's positive feedback and thorough and constructive critique of our study. We apologise for the avoidable mistakes we made and revised manuscript accordingly.

We appreciate the concerns raised regarding the limitation of single cell analysis in robustness. However, we would like to emphasize the importance of such unique data set for the field for the following reasons:

There are very little data available in the literature on the single cell analysis of specific responses due to the technical difficulties, limited sample availability and low frequency antigen-specific T cells in the circulation. Hence, such detailed analysis can yield meaningful correlations with disease outcome, because our approach controls many confounding factors caused by differences in antigen specificity, targeting viral protein, HLA restriction, antigen epitope load on antigen presenting cells etc. As demonstrated by our group previously using smart-seq2 cells¹ instead of 10X Genomics technologies, we have shown that even small number of single cells can produce high quality data. In this study, Tetramer+CD8+ T cells were sorted using stringent gating criteria, and their purity was verified by confirming the Tetramer-PE index, ensuring 100% purity of the tetramer+CD8+ T cell population¹. Therefore, we remain confident in the quality of our data. Although the cell input from vaccinated donors is limited, such data are rare in the field due to limited sample availability, technical challenges posed by low frequency of antigen-specific T cells. Therefore, we believe our findings are valuable for advancing current knowledge. In particular, we are the first group to reveal paired alpha and beta TCR sequences specific for Mpox immunodominant T cell epitopes. In addition, we have now performed additional flow cytometry analysis and validated some key molecules identified from single cell data such as CD29 (*ITGB1*), CD44 and GNLY. We have also added the limitation of single cell analysis performed in this study in the revised discussion section.

General remarks

C1-1. The present manuscript by Chen et al. has important and interesting aspects. It employs a useful assay (ELISPOT with VACV-infected cells) as a readout to analyse virus-specific T cell responses *ex vivo*.

R1: We thank the reviewer's positive comments regarding our study and assays performed.

C1-2: Some responses in convalescents were surprisingly low; it would have been interesting to correlate response strength with available information on patient cofactors such as HIV infection.

R1-2: This is a very good point. We checked clinical information of those with very low response, they are not HIV+ and without comorbidities, and our cohort is too small to carry out any further meaningful analysis.

C2. It is also interesting and informative that the authors directly compare specific immune responses in MPOX-infected and MVA-vaccinated individuals, although those differences between

these responses which they identify based on single-cell analysis alone, and which they strongly emphasize in their presentation, might not be sufficiently robust.

R2: We appreciate the positive comments regarding comparing specific immune responses in MPOX-infected and MVA-vaccinated individuals, and concerns regarding the robustness of our conclusion, we have now tuned this down.

C3-1: Focused analysis of epitope-specific T cells was undertaken, but limited to a single A2 epitope and to a small number of subjects.

R3-1: We agree about this comment, however, due to the limitation in availability of samples, and considering HLA-A2 is one of the most common HLA type in our cohort, and most of the Caucasian population; and as the epitope is one of most dominant HLA-A2-restricted epitopes which is verified in our cohort, and there are no any single cell analysis in the literature in Mpox infected and Vaccinated individuals, with all these limitation in mind, we feel the data we generated still provides valuable information for the field. We have now discussed this limitation on line 444-447 in our revised manuscript.

C3-2: Direct quantitation of epitope-specific T cells was limited to expanded T cell cultures, although it should be expected that several immunodominant VACV/MPXV epitopes are expanded enough in order to quantitate them directly *ex vivo* by tetramer/multimer staining.

R3-2: We would like to emphasise that ELISPOT assays for evaluation of total T cell responses were conducted *ex vivo* (figure 1, figure 4 and figure 6). The expanded cell lines were only used for epitope screening, confirmation, and functional assays exemplified by us and others in previous studies¹⁻³. Direct quantification of epitope-specific T cells, particularly ILD-specific responses, was performed through *ex vivo* tetramer staining, as illustrated in the immunophenotype in Figure 4 and the single-cell analysis in Figure 5. The frequency of *ex vivo* tetramer-specific CD8⁺ T cells for each donor has now been incorporated into the main Figure 4 as new figure 4d.

C3-3: Well-characterized epitopes with other HLA restrictions that might be of higher importance than A2 in populations other than those of European descent (but also of importance in Europeans) would have been available.

R3-3: We fully agree about the importance of immunodominance and HLA association and its impact on disease outcome. However, our cohort is too small to answer such question properly. As mentioned in R3-1, there are very limited data in the literature on detailed analysis of Mpox-specific T cells in Mpox-infected individuals and none at a single cell level. Given the limited sample availability and considering HLA-A2 is one of the most common HLA types in our cohort and the Caucasian population, we feel the data we generated still provides valuable information for the field. As such, we have now discussed this limitation on lines 444-447 in our revised manuscript.

C3-4: The authors could also have been clearer in their acknowledgment that their chosen epitope was identified in dominant MVA vaccination responses in humans already in 2005 (ref. 25). Generally, they could have provided more detailed literature context for a discussion of MVA vaccination responses.

R3-4: We agree with this comment and have clarified the epitopes identified in ref 23 (old ref 25), and added more literature regarding to MVA vaccination responses has now been added on lines 78-83.

C4-1: The scRNAseq analysis of T cells specific for this epitope is unfortunately hampered by small cell number and lack of information on input cells, and thus unlikely to be representative. It would be much better in general, in studies in this field, if such single cell analyses would be undertaken in a sufficiently comprehensive and documented way, and if their most important results were further examined using independent methods.

R4-1: We highly appreciate this reviewer's concerns about our single cell analysis approach, therefore provided our response at beginning.

C4-2: The comparison of MPOX and MVA vaccination responses to SARS-CoV-2-specific T cells is hard to interpret as long as it is not even stated what kind of (and how many) SARS-CoV-2-specific T cells were studied.

R4-2: We apologise for not making it clear which SARS-CoV-2-specific cells were used in this analysis. The paper¹ from which the cells were used is referenced and we have now included the following description of the cells: 'Briefly, 912 SARS-CoV-2 NP₁₀₅₋₁₁₃-B*07:02-specific T cells were isolated using peptide-MHC pentamers from 10 individuals, 1-3 months after initial infection, and single-cell RNAseq was carried out to investigate gene expression and TCR repertoire differences.' On lines 300-302.

The SARS-CoV-2-specific T cell dataset was included in this study because both datasets were generated using the same method and obtained from T cells at a similar convalescent-stage timepoint. This presented a unique opportunity to compare these datasets with different virus infection offering valuable insights into the composition of MPOX-specific T cells and highlighting their unique as well as common characteristics relative to T cells from other viral infections. We also added a sentence in discussion regarding the limitation of just comparing single epitope specific responses and small data set, on lines 444-447.

C5: The manuscript is hampered by sometimes contradictory and incomplete information regarding methods and results, as specified below in detail (for which I apologize, but I do not know how to present my argument more succinctly).

R5: We apologize for unclarity of the method and result section and have now addressed all the comments point by point.

C6-1: Unfortunately, the interpretation of results as shown in Title, Abstract, and Discussion tends to make multiple statements that are insufficiently substantiated by the data, highly speculative, or incorrect, as explained below.

R6-1: We appreciate the comments made by this reviewer and rewrote the abstract taking the consideration this reviewer's concerns.

C6-2: Here I mention one example: the claim that T cells after MPOX infection show "greater migratory potential" than T cells after vaccination. This is entirely based on the single-cell analysis that was made, on the side of the vaccinated cohort, with only 160 cells in total, of unstated purity, from an unstated number of vaccinated donors in unstated proportions. In my opinion, generalizations and main messages of a paper should not be based on such numbers, and the authors' neglect to provide available background information is worrying.

R6-2: We have now revised the manuscript and highlighted limitation of our single-cell analysis with small number of input cells from vaccinated donors. We also emphasised that our confidence in these 160 cells with extremely high purity by tetramer staining with very stringent gating, shown in new Fig. S3D.

C6-3: That said, the authors' interpretation does not even fit their data from this experiment, since chemokines, which are paradigmatic "migratory" molecules, were more highly expressed after vaccination than MPOX infection according to Fig. 4B (but not according to some of the author's descriptions in the text). In contrast, certain integrins were more highly expressed after infection than vaccination, which may be the reason why the authors sometimes merge "integrins and migratory" genes together in a way that is not obviously justified and makes the discussion less clear.

R6-3: We agree that chemokines are quintessential "migratory" molecules. It is well established that integrins also play a crucial role in T cell migration⁴. Therefore, we considering T cells with high expression in both the chemokines and integrins as having greater migratory potential. We have now made our justification clearer in our revised manuscript.

C6-4: Since the single-cell experiment is unlikely to be robust, I would suggest to eliminate it altogether. It could only be rescued if some of its main findings would be confirmed using robust methods such as analysis of protein expression.

R6-4: We respectfully but could not agree, for the reasons highlighted in above responses. We recognise 160 is a small number, and we have now emphasised the limitations in our discussion section and verified the single cell results with flow cytometry, which are presented in new figure 6D. We believe our dataset although small, and with the confidence of the purity of our sorted cells and well established methodology¹, we purposely choose this path after considering the noise and confounding factors, so we could have the "cleanest" dataset we could have, therefore very low number cell was expected in the study design after considering all the pros and cons.

Result

C7: Table 1 provides sample collection time as days post symptom onset, which is helpful. It would also be informative to add information for "days post symptom resolution" or duration of symptoms, if that is available.

R7: Unfortunately, this information is not available as it was not collected from the individuals in the cohort, which is why it has not been included in the table. We have available date of symptom onset, from which 'Sample collection time (days post symptom onset)' was calculated.

C8: Fig. S1 is visually pleasing, but it would be more informative to provide a table with HLA types per individual, such that HLA associations, homo-/heterozygosity etc. might be inferred and all allotypes can be provided. E.g. 1/3 of HLA-B is "other", which is a large information gap. The factual content of the figure is unclear as well: it is supposed to show the HLAs of 13 convalescent participants, so n=26 for alleles of a given locus. However, several of the diagrams have segments far smaller than 1/26th of a full circle. If more is known about the HLA type (HLA-C, DP, DRB3/4/5, DQA, DPA) that info might be provided, too.

R8: We thank the reviewer for pointing this out. We have corrected Fig. S1 to show the proportion of each HLA allele amongst the 26 alleles in the 13 convalescent participants. The smallest segment now should be $1/26^{\text{th}}$ of a full circle as you would expect. We have also included a supplementary table with the HLA-typing results for each convalescent individual (Supplemental table 1).

C9: line 113, "we set up an ex vivo IFN γ ELISpot assay by stimulating PBMCs with "the VACV strain Lister as previous [sic] described (18)" - I cannot find an ELISpot assay with VACV in the cited paper. Please make the statement more precise.

R9: We apologise for the wrong reference being cited here. We have updated the citation to the correct references. The correct reference used recombinant MVA-infected PBMCs, however we used the same method and optimised it to work with VACV-infected PBMCs.

C10: Fig. 1E legend, is n=11 or 12 here instead of n=13? Please state in text or legend. Were some convalescent individuals excluded because of sample availability or technical problems?

R10: We apologise for not making this clear. The cohort consisted of 13 convalescent individuals, however due to limited sample availability, PBMCs for Mpx009 were only enough for the ELISpot assay but samples were not available for any further assays and so were not used in the AIMS

experiment. In addition, due to its high background in the AIMS assay, Mpox011 was excluded from the analysis in Fig 1E and F. Please see the FACS plots below for Mpox011 AIMS assay showing the high background in the PBMC only samples. This has been updated in Figure. 1E and the legend has now been updated to make it clearer. This has further been included in the text on lines 131-133.

C11-1: Fig. 1F contains the proportions of CD4/CD8 cells for n=12. It contains these proportions both for high and low T-cell responders, although the proportions would probably be more robust for high responders (there were two near-zero or zero responders for CD8 and CD4 each in Fig 1E). Please add some number to each bar of Fig. 1F that indicates the overall size of response to make the readers' assessment easier, or scale the bars according to response size. Adding lines that indicate paired values in Fig.

R11-1: We thank the reviewer for this comment. In response to comment 10, we have corrected the sample size to N=11 for Fig 1E and excluded Mpox011 due to its high background in AIMS assay (Please see response R10). We have added lines to figure 1E to show the CD4 and CD8 responses for the paired samples. Similarly, we have added the proportions as text to the bars in figure 1F. The bars are scaled to be the proportion of the T cell response that is a result of CD4+ or CD8+ T cells (e.g. Mpox002, 15.5% of the AIM+ T cell response was elicited by CD8+ T cells while 84.5% was elicited by CD4+ T cells, whereas for Mpox008, all the response was elicited by CD4+ T cells). Furthermore, we have re-worded the text on lines 135-141:

'The relative proportion of the CD3+ T cell response attributable to VACV-reactive CD8+ and CD4+ were calculated for each individual. As shown at Fig. 1F, the proportion of CD4+/CD8+ reactive T cells varied between participants. While Mpox008 showed only a CD4+ T cell response and Mpox005 showed only a CD8+ T cell response, all other individuals showed a varied proportion the T cell response elicited by CD4+ and CD8+ T cells; however, only a marginal difference was observed overall, with an average proportion of 51.7% CD8+-reactive T cells compared to 48.3% of CD4+-reactive T cells in this cohort.'

C11-2: (please correct y-axis spelling for Fig. 1F)

R11-2: We have corrected the spelling mistake in the y-axis title for Fig 1F.

C12: Did these reconvalescents still have elevated overall CD8/CD4 ratios in their peripheral blood as described by others?

R12: We thank the reviewer for this comment. We have looked at the overall ratios of CD8/CD4 proportions in the peripheral blood of donors and indeed we found that convalescent individuals had a significantly higher CD8:CD4 ratio compared to healthy donors.

C13: line 139, "were selected". However, the Methods part suggests that all available epitopes were included. Is that correct, or if not, what were the selection criteria? The methods part (line 605) speaks of experimentally verified epitopes. IEDB makes the important distinction between T cell assays and MHC ligand assays. Were only epitopes with available T cell assay information included (an approach I would prefer)? Please clarify. It might be contested that peptides for which only MHC ligand assays but not T cell assays are available are "experimentally defined" T cell epitopes.

R13: Thanks for the comment. We have included all the available epitopes which have been tested by T cell assay into the mega pools. In the revised manuscript, 'were selected' has been replaced by 'were included' on line 148 and 'T cell assay' was added into method section on line 664.

C14-1: Please provide a reference for the categorization of antigens as early, intermediate, late.

R14-1: Thanks for the comments. The categorization of antigens as early, intermediate, late was made based on this two References: Protein: PMID: 31067474; Deep sequencing: PMID: 20534518, which have been added to the manuscript on line 153.

C14-2: Please use consistent terminology (intermediate/late vs. late).

R14-2: Thanks for pointing it out. We have checked the manuscript and used the consistent terminology. Proteins should now all be classified as either early or intermediate/late based on the references in response R14-1.

C15: line 142, "The remaining 22% of CD8 and 28% of CD4 epitopes corresponded to VACV epitopes that contained mutations in MPXV genome" - i.e. non-synonymous mutations, since we are at the amino acid level? It's possible that the 78% and 72% epitopes that were conserved at the aa level had synonymous mutations.

R15: Thanks for picking this up. The mutations here are at amino acid level, and the text has been updated accordingly in the manuscript on lines 152-153.

C16: Line 162 and Fig. 2C, were these antigen-specific responses partially complementary? How many donors recognized at least one of these 5 antigens?

R16-1: We not quite sure what “partially complementary” means. However, the point of this data (figure 2C) is to take a step further with the observation by focusing on the early proteins based on the differences observed between “early” vs “intermediate/late” protein epitope pools (Fig 2B). We synthesised and tested overlapping peptides spanning five early proteins of MPXV, including A9, F3, E12, D10 and D1, which have been reported to be the most immunogenic after orthopoxvirus infection. We found D1 is the most dominant antigen followed by F3 then E12. A9 and D10 were not recognised by any of the donors. 7 of the 12 donors recognised at least 1 of the antigens tested here.

C17: line 181, "HLA-A*02:01 is the most common HLA allele in human populations". This statement is not covered by the cited reference and is incorrect as stated. The most common classical HLA alleles are probably HLA-DRA*01:01, or possibly DPA1*01:03 among the more polymorphic ones. If we limit consideration to classical HLA class I alleles, the statement seems correct for people of European descent but not others. Other candidates for the most frequent 4-digit class I allele worldwide are A*11:01, A*24:02, or HLA-C*07:02, which are much more frequent than A*02:01 both in India and China.

R17: Thanks for the comment. As mentioned in the response to your comment 3-3, in our small cohort, 10 are of European descent and two are of South Asian origin. Furthermore, almost 50% individuals (6 out of 13) are HLA-A2 positive. Although the reference (ref 24) doesn't explicitly investigate HLA allele frequencies, it is a reference for an online database from which global allele frequencies can be interrogated, and is what we used to determine allele frequency in the Caucasian population, showing HLA-A2 as most common allele. On **line 193**, we have updated the text by replacing ‘human population’ with ‘Caucasian population’.

C18-1: How many A2 or A*02:01 epitopes were individually tested in ELISpot assays, if any, and in how many donors? The text in lines 179-195 seems unclear in that respect. Subsequent text suggests that ex vivo ELISpot was done and short-term T cell lines were generated with peptide pools, not individual peptides. Please specify in line 187f. In that case, statements about immunodominance need a disclaimer, since not all epitope-specific subpopulations may expand equally well in vitro in such T cell lines.

R18: Due to limited sample availability, we were unable to test each individual A02:01 epitope by ex vivo ELISpot. We apologise for not explaining clearly in the text. We have now revised our sentences accordingly. The text now reads as:

‘T cell responses to 71 A*02-restricted early epitope pool and 68 intermediate/late epitope pool were evaluated using an ex vivo ELISpot assay in six HLA-A*02:01+ convalescent individuals.’ on **lines 194-196**

and

‘Next, we generated short-term T cell lines from four convalescent HLA-A*02:01+ individuals by stimulating PBMCs with A*02:01-specific peptide pool of the early expressed antigens’ on **lines 199-200.**

C19: line 182, "T cell responses to 71 A*02:01-restricted early epitopes and 69 intermediate/late epitopes" - This statement suggests that all A2-restricted epitopes in your pools are A*02:01-restricted epitopes. Were all of the A2 epitopes described to be A*02:01-restricted in IEDB or the literature sources? The legend to Fig. 3 speaks of "A2-restricted" responses. Please make the terminology consistent, even if it may seem tedious to always write "A*02:01".

R19: We apologise for this confusion. Since some of the literature, especially the old ones and IEDB database didn't use 4 digits HLA typing, some epitopes are listed as A2, not A*02:01. We have updated this terminology with "HLA-A2".

C20: line 187ff, please indicate whether all the peptides mentioned here were conserved between VACV and MPXV.

R10: 6 of the 7 peptides used are conserved 100% between VACV and MPXV. HLA-A*02:01-B7₁₁₃₋₁₂₁ (IMYDIINSV, IMY) contains a conserved L->I substitution at the start of the epitope. This has been added to the text on **lines 205-207.**

C21-1: line 195, "all three participants tested showing a strong response to this epitope", but Fig. 3C shows ILD tetramer-specific T cells in four donors - supposedly four T cell cultures/lines - and one of them did not show a "strong" response.

R21-1: Thanks for picking this up. Now we changed the sentence to ‘Among them, epitope ILD showed the greatest immunodominance, with all of four cell lines tested showing a detectable response to this epitope’ on **lines 208-209.**

C21-2: Please make it clearer in the text and legend where peptide-stimulated T cell lines were used for analysis.

R21-2: the figure legend and text have been updated accordingly on **lines 212-213.**

C21-3: Fig. 3C shows tetramer vs same tetramer, and both correlate well, which is usually expected - such an experiment can only control for fluorochrome-based background staining but not other forms of background or interference. Another plot showing tetramer vs independent marker (e.g. CD8) or dual staining with different tetramers may give a better impression of potential background in this assay.

R21-3: For this part, all peptide-MHC complexes were produced by UV-mediated MHC peptide exchange. To increase the specificity and sensitivity, we used dual labelled tetramers for the

screening, as noted in Anderson, R.S. et al 2012 Nat Protoc 2012 Vol. 7 Issue 5 Pages 891-902: Parallel detection of antigen-specific T cell responses by combinatorial encoding of MHC multimers. We have updated this figure with CD8 vs Tetramer.

C21-4: Please indicate the donors shown in Fig. 3D.

R21-4: We have updated the figure with the donor ID (Mpox004).

C22-1: Fig. 3D, it is unexpected that there were many more cells staining for TNF-alpha than IFN-gamma. This is unusual for virus-specific CD8 T cells. The authors seem to disregard this finding by sweepingly saying that "all" of the T cell lines were "expressing multiple cytokines (IFN γ , TNF α , and IL2)". Is there an explanation for this unusual finding, was it previously observed for VACV or poxviruses?

R22-1: In Figure 3D, we present ICS results from cell lines derived from Mpox 004, using target cells infected with vaccinia virus. Less IFN-gamma positive cells were observed than TNF-alpha+ cells. It is not unusual for virus-specific CD8 T cells. In fact, it has been observed in other virus infections, such as Dengue-specific T cells⁵ and SARS-CoV-2⁶. We believe this may be due to the TNF alpha being more sensitive to antigen stimulation than IFN-gamma, as well as many more complicated possibilities due to different virus infection, stage of the disease, amount of antigen exposure, antigen specificities etc⁷⁻⁹. Similar patterns were noted in our studies of SARS-CoV-2-specific T cells, where more cells produced TNF- α than IFN- γ when exposed to live SARS-CoV-2 infected cells¹⁰.

C22-2: Or was there confusion about the fluorochromes used? Fig. 3D, upper part, shows IFN-gamma PE/Cy7, but the Methods section says that IFN-gamma FITC and IL-2 PE-Cy7 antibodies were used for ICS. This seems contradictory. Some other fluorochromes do not match either.

R22-2: We apologise for the mistake here. The methods section has been corrected accordingly on **lines 760**.

C23: Fig. 3F, please show killing of the non-infected LCLs as a control, if the data exist. It is curious that the highest killing was observed in the T cell line with the lowest number of ILD-specific T cells (Mpox005). Why is that? What were possible confounders in this experiment?

R23: We thank the reviewer for this comment. We used non-infected LCLs with T cells at corresponding E:T ratios as a control, and the percentage of killing presented in the figure were calculated using the following formula: $100 * (\text{number of live LCLs in the control wells} - \text{number of live LCLs in the infected well with same E:T ratio of T cells}) / \text{number of live LCLs in the control wells}$. This has been updated in Method section on lines 795-798. A representative FACS plots of the killing assay has been added to the Supplementary Figure 4 and shown below. The frequency of ILD+ T cells shown in Fig3c is the frequency within the *in vitro* expanded short-term T cell line. Those tetramer+ T cells in the short-term cell lines were then enriched to single epitope specificity and expanded for subsequent functional assays. In the cytotoxicity assay, the same number of ILD+ T cells were used as effector cells to assess cytotoxicity., The greater killing of MpoX005 is likely due to the greater effector function of these cells.

C24-1: Does Fig. 3F indeed show killing of infected cells as stated in text (line 203) and legend? The methods part mentions killing assays only in the context of "BCLs" (usually abbreviated LCLs or B-LCLs) loaded with high concentrations (10 μ M) of peptide.

R24-1: we apologise for the confusion and the method section has been updated with correct information on line 221.

C24-2: It makes little sense to label the diagrams in Fig. 3F with the abbreviation of a single peptide (ILY, ILD) if the effectors were in fact polyspecific T-cell lines previously stimulated with multiple peptides. What was done here?

R24-2: T cells specific to a single epitope were tetramer sorted and grown as CD8+ bulk lines specific to a single epitope. Therefore, the bulk lines used in the killing assay are not polyspecific but are specific to the epitope indicated in the figure. This has been updated on lines 212-213.

C25: line 208, "Given that we identified functional differences in MPXV-specific CD8+ T cells in convalescent individuals", what does this mean? Heterogeneity among convalescent individuals?

R25: Thanks for picking this up, in line with the responses to other relevant comments by this reviewer, we are hoping the revised manuscript has made this point clearer. Please see lines 226-228.

C26: What was the time interval between the two doses of the MVA-BN vaccine? Cannot find this in the text or the graphics in Fig. 4A.

R26: The two doses of the MVA-BN vaccine were administered no less than 28 days apart as described in the pre-print describing the cohort submitted after our initial submission by Drennan et al.¹¹. This has now been referenced and is in the methods section on line 642. We have also corrected figure 4a to match.

C27: line 213, it may not be obviously correct to call this a "memory" T cell response if T cells were analyzed shortly (4 weeks) after vaccine doses.

R27: We apologise for this and agree this is not a memory response 4 weeks after vaccination. We have now removed this and called it an 'Overall T cell response' on lines 229-230.

C28: I wonder why the differences in Fig. 4B were not significant. What were the p values, what test was used? Are the results non-significant in parametric and non-parametric tests?

R28: Mann Whitney U was used with Bonferroni correction for multiple testing correction. After correction, adjusted $p=0.204$, 0.43 and 0.567 respectively for Convalescent vs D1-D28, Convalescent vs D2-D28 and D1-D28 vs D2-D2. Before correction, nominal $p=0.068$, 0.215 and 0.567 respectively. Similarly, after t test rather than Mann Whitney, results remain not significant

C29-1: line 218, "there is a slightly stronger T cell response" - The effect was not statistically significant but quantitatively strong: the median response after infection was apparently more than twice as high than after the first or second vaccine dose. It's interesting that a strong booster effect of the second dose was observed, but only in a minority of subjects. Is it possible that they were more likely to be immunocompromised? Who were the vaccinated individuals, what were the inclusion criteria?

R29-1: The vaccinated cohort formed part of a prospective single-centre observational study of immunological responses to MVA-BN vaccine administered to individuals attending a sexual health vaccination clinic in Oxford, United Kingdom (UK). Clinic attendees were approached by a member of the study team and invited to participate. HIV-infected individuals were excluded. This is as described in Drennan et al.¹¹

C29-2: Table S1 tells us they were all-male with a median age of 36 and only two people over 50. Comorbidities or other factors are not listed.

R29-2: The participants in this vaccinated cohort are healthy donors. None of them had HIV. In terms of comorbidities, one participant had a history of type II diabetes (IMO1000003), and three participants had a history of obesity (including one with a BMI of 61.1 - IMO000007, and two (IMO1000004 and IMO1000015) with BMIs in the 30s). These are the only relevant comorbidities in this cohort which has not impacted on the immune response as shown in the figure below (CMV status and BMI). Now the CMV infection status, BMI and Diabetes status for each participant has been added to the table in Supplementary Table 1.

C29-3: Might some of them have been HIV-positive, as were two of the 13 convalescents? What would be the outcome of this analysis if HIV-positive subjects were excluded, if that is possible?

R29-3: For the vaccinated cohort, HIV-infected individuals were excluded and not recruited. We repeated all analyses after excluding data from the two HIV+ convalescent individuals and observed no impact on the results.

C30: Two high responders to the second dose in Fig. 4b stood out. Did these subjects overlap with those who had had a smallpox vaccination in their youth?

R30: None of them had previous smallpox vaccine. Those two individuals that have had a previous smallpox vaccination were only administered 1 dose of MVA-BN, as mentioned in Drennan et al.¹¹, and as such had no sample collected at the D2-D28 timepoint. At the D1-D28 timepoint, these individuals had a relatively low response. The two individuals with the previous smallpox vaccination are highlighted in yellow in the figure below:

C31: line 234, "with a significantly greater level of tetramer+CD45RA+CCR7-CD8+ T cells 28 days after a second vaccination compared to convalescence" - This contradicts Fig. 4D,E and the text above. The only significant difference between the two cohorts was found for KLRG1+ cells, according to Fig. 4E.

R31: We apologise for this contradiction; this was a typo on our behalf. The significance stated on lines 234-236 with a $p=0.03$ corresponds to the significant difference in the KLRG1+tetramer+CD45RA+CCR7-CD8+ T cells as shown in figure 4E. This has now been corrected in the text on line 251.

C32-1: The scRNAseq experiment is unsatisfactory. On the vaccination side it contains only 160 cells from an unstated number of donors in unstated proportions, sorted with tetramers to an unstated purity.

R32-1: Thanks for the comment. We should have clarified in the main text/section of Method that all the tetramer-positive T cells sorted for single-cell RNASeq were at 100% purity with very stringent gating and checked by the index as shown in supplementary Figure 3d

C32-2: It seems that such experiments are now so fashionable that they have become mandatory in studies of this type. At the same time, they are performed with so much data processing and normalization procedures that small differences in gene expression may be represented as large ones, subclusters may be categorized in an arbitrary number of ways, etc. It is stated that cells were "integrated based on patient of origin", or, as stated in Methods, "Patient-specific cells were integrated using Harmony v.1.0 to remove batch effects." Are the batch effects referred here the differences between patients? Why is it considered valid to remove or reduce these differences by mathematical processing? This might be adequate for batches of samples that should be expected to be biologically equivalent. But this is not the situation when a variety of human subjects is studied.

R32-2: We thank the reviewer for bringing up this concern, and although we agree sometimes the data can become distorted by a large amount of mathematical processing of the data, we carried out standard data processing and normalisation for such experiments, methods which have been benchmarked and compared in multiple publications (e.g. <https://doi.org/10.3389/fgene.2020.00041>, <https://doi.org/10.1038/s41592-023-01814-1>, <https://doi.org/10.1186/s13059-021-02552-3>). Batch effect here is the differences between patients. Generally, with different patients, the heterogeneity between subjects can overshadow biological differences, therefore, it is generally accepted to integrate donors together mathematically. (<https://doi.org/10.1101/2020.09.17.301911>, <https://doi.org/10.1186/s13059-019-1850-9>). Although the differences between vaccinated vs convalescence may be masked by this integration, it is promising that we still see some differences after the integration. Re-doing the clustering without any integration there is a similar overall story. 4 clusters are formed in the analysis, with 2 clusters showing greater abundance of cells from convalescence, and 2 clusters showing greater abundance of cells from vaccination

C33: The populations were divided into four clusters, but these clusters were not clearly separated in the UMAP plot (Fig. 5A). So I wonder how distinct these clusters actually are, and if alternative clustering would have been similarly justified.

R33: Thanks for the comment. Given these are all antigen-specific cells, wouldn't expect very separate clusters on a UMAP as the cells are relatively homogenous but as we and others have previously published, subtle differences remain allowing for sub-clustering (e.g. Low-Avidity CD4+ T Cell Responses to SARS-CoV-2 in Unexposed Individuals and Humans with Severe COVID-19 <https://doi.org/10.1016/j.immuni.2020.11.016>).

The cells were clustered using both the Louvain and Leiden algorithms, the two most popular algorithms for cluster identification in single-cell RNAseq datasets. We observe similar clustering using both methods. Using the normalized mutual information (NMI) metric to compare the clusters assigned to cells using these two methods, the NMI=0.939 (NMI ranges from 0-1). A higher value indicates a greater degree of similarity between two, suggesting that an alternative clustering algorithm generates the same clusters in the data.

C34: Fig.5D legend says "cluster 0 marker genes" but Figure and main text suggest these are cluster 2 marker genes.

R34: We apologise for this mistake. This is meant to read 'cluster 2 marker genes' and has been corrected in the text **on lines 272-273**

C35: Please make it clearer in this section that the cells being analyzed in scRNAseq are specific for one epitope, and mention the epitope in the Results text (e.g. line 240, 258) and Fig. legends.

R35: We apologise for the confusion here. We have now changed the text to say ILD-specific cells rather than MPXV-specific in both the text (**lines 258 and 276**) and the figure legends.

C36: As the authors state, many clonotypes were identified in just one cell each (line 258ff). But that would imply that this analysis does not provide us with robust information about the clonal TCR composition and repertoire complexity of these samples.

R36: We are not investigating the overall clonal TCR composition but rather the clonal TCR composition of sorted antigen-specific T cells which will be relatively low in general due to the relatively low frequency of antigen-specific T cells in the whole T cell population (Peng et al NI 2022). Therefore, the fact that a lot of clonotypes only appear in very few numbers of cells isn't too surprising given this low frequency of cells. The overall frequency of tetramer+ cells can be seen in the new figure 4D

C37: The experiment lacks controls, so we cannot say which of the low-frequency TCRs may be abundant TCRs specific for something other than the MPXV epitope that ended up as contaminants in the tetramer-sorted cells. The purity of sorted cells isn't stated either.

R37: All the cells that were sequenced, and as such the TCR clonotypes identified, were CD8⁺tetramer+. Tetramer+ cells were sorted directly into individual wells of a 96-well plate containing cell lysis buffer after which cDNA conversion and library preparation was carried out. Therefore, of the cells that were sequenced, they were all 100% tetramer+ cells with very stringent gating shown in supplementary Figure 3d and therefore no contaminating cells were sequenced. Therefore, all the TCRs are specific for the MPXV-epitope and not something other than the ILD epitope.

C38: However, it seems incorrect if the authors say that TCR alpha clonotypes from vaccinees were "all identified in just one cell" (line 265), compare Fig. 5F-G. Some clonotypes were clearly more frequent than others.

R38: We apologise for the confusion here. We have corrected this sentence to say: '56% of cells from vaccinees had an alpha clonotype found in just one cell, compared to 29% of cells from convalescent individuals which had a clonotype found in just one cell', **on lines 282-284**

C39: The statement in line 270 suggesting that even public clonotypes (found multiple times in this experiment) may in fact be specific for SARS-CoV-2 dramatically invalidates this experiment.

R39: We thank the reviewer for this comment. Although 2 of the public beta clonotypes we identify are similar to previously reported SARS-CoV-2 TCRs with similar or identical CDR3 sequence, but none of the SARS-CoV-2 TCRs has the same TRBV gene usage to ILD-specific TCRs, therefore, they are unlikely SARS-CoV-2 specific. Secondly, as we clarified to previous comments, we are very confident that 100% ILD-tetramer+ CD8 T cells were sorted for our single-cell analysis, there would not be any

contamination of SARS-CoV-2 specific T cells in our analysis. To avoid future confusion, we deleted this sentence from the main text.

C40: Line 271 refers to Table 2 supposedly comparing clonotypes found here to SARS-CoV-2-specific ones. I cannot find a Table 2 in the materials submitted.

R40: Please see the reply for C39 above, this table was from our old version of manuscript and the sentence has been updated accordingly.

C41: Relating to Fig. 6, what kind of SARS-CoV-2-specific T cells were studied in these experiments? Cells specific for a particular antigen or epitope, and which one? I cannot find this in Results, Legend or Methods.

R41: We apologise for not making it clear which SARS-CoV-2-specific cells were used in this analysis. The paper from which the cells were used is referenced and we have now included the following description of the cells on lines 300-302: 'Briefly, 912 NP1₀₅₋₁₁₃-B*07:02-specific T cells were isolated using peptide-MHC Pentamers from 10 individuals, 1-3 months after initial infection, and Smartseq2 single-cell RNAseq was carried out to investigate gene expression and TCR repertoire differences.'

C42: Regarding Fig. 6A, it would be good to see a few control genes (expected to be expressed or not expressed in CD8 T cells, such as CD3, CD45, CD8, TCR, etc.) in order to judge the significance of this analysis and the granularity of the results.

R42: Thanks for the comment, now we added CD3, CD45, CD8 expression into **supple. Figure 5 and mentioned on lines 838-839**. We remove all TCR related genes (TRA[V/J]*, TRB[V/D/J]*) from the dataset to prevent any clonotype influence on clustering, as is standard for T cell scRNAseq analysis.

C43: What is the real quantitative difference of expression of the genes mentioned in line 284-291? Fig. 6B seems to suggest that highly significant group differences (****) can result even if there are very minor shifts of average expression patterns.

R43: We have added logFC for the comparison between cells from MPXV convalescence and those from SARS-CoV-2 convalescence, **on lines 324-332**. A Wilcoxon-signed rank test as used to compare between groups and two-tailed p-values adjusted for multiple testing using the Holm adjustment method.

C44: Regarding Fig. 6C, please indicate the total number of genes that was studied in the experiment, and please apply a statistical correction for multiple testing before you report highly significant p values for 12 selected genes.

R44-1: These 12 genes are all the ones tested for this analysis. The analysis in Fig 6C was a targeted analysis of selected genes of interest based on the results of the previous analyses. P-values were

calculated from a Wilcoxon signed-rank test and are corrected for multiple testing using the Bonferroni correction.

C44-2: It would have been helpful to verify differential expression of these genes by FACS or another direct method, preferably measuring protein expression.

R44-2: we have carried out ex vivo flow cytometry phenotype with some of the markers, we observed consistent results to the scRNASeq data that ILD-specific T cells from Mpox infected donors expressed higher level of migratory molecules CD44, CD29 (*ITGB1*) and CD49d (*ITGA4*) and cytotoxic molecule Granulysin and Granzyme A, as shown in the new Fig. 6D and in the text on **lines 335-340**.

C45-1: The summarizing paragraph (line 311-314) compares "MPXV-specific memory T cells in convalescence" to something, but to what?

R45-1: Thanks for picking this up, now the sentence has been updated on **line 342** with "compared to MVA-BN vaccination, MPXV-specific T cells in convalescence."

C45-2: For example, these cells are said to have "increased expression of migratory molecules", but Fig. 6B seems to show that their expression of chemokines is lower than SARS-CoV-2-specific cells and lower than in cells MVA-vaccinated subjects. In line 296ff there's still another statement, but that statement seems to combine cytokines and chemokines in some way; nonetheless it contradicts the chemokine panel in Fig. 6B.

R45-2: Although we see no significant differences between the chemokine and integrin modules scores from the single-cell RNAseq data, when investigating individual molecules at both the RNA and protein level, we do see significant differences in several migratory molecules between samples from MPXV-convalescence and vaccination. At the RNA level, *ITGB1*, *ITGA4* and *GPR183* appear to show greater expression in convalescence compared to after vaccination. This was confirmed at the protein level in our newly added figure 6D FACS validation of the scRNAseq data, in which CD29 (*ITGB1*) and CD49d (*ITGA4*) show increased protein expression in convalescence. However, we do agree it would be good to investigate the expression of these migratory molecules on T cells in skin lesions, given skin lesions are characteristic of Mpox disease. However, this was not possible in this study and is listed as a limitation in our discussion **on lines 447-453**.

Discussion:

C46: line 335, "This finding contradicts recent reports that shows a greater persistence of CD4+ T cells compared to CD8+ T responses in MPXV convalescence (12,14)." How does ref. 14 (Cohn) show greater persistence of CD4+ than CD8+ T cells, and what is meant by persistence? It seems to me that CD8+ T cells were as frequent or more frequent than CD4+ T cells in ref. 14 - however that paper might not have studied enough donors to even make such an assessment.

R46: Thanks for the comments, indeed this is a typing error with "persistence", now it has been corrected with "presence" on **line 363**.

C47: line 337, "In our study, immune responses were measured against the whole VACV, resulting in a with broader [sic] epitope coverage compared to peptide-based assay." This is by no means clear and would require proof. In a cell infected with a very large virus such as VCAV, there may be competition between epitopes for presentation depending on various factors (HLA affinity, efficacy

of processing, expression kinetics) and the peptide pools were also very complex (even called "mega pools" by the authors).

R47: We have added the fact that using the VACV virus rather than peptide stimulation allowed for a response more similar to natural infection for the precise reasons as this reviewer mentioned and also highlighted and discussed in detail in our previous publication¹⁰, as well as the fact that the broader epitope coverage compared to peptide pools may be due to having to select a set number of peptides/more common HLA to include in the pool rather than peptides covering the whole genome given the large size of the MPXV genome. See **lines 365-367**:

‘In our study, immune responses were measured against the whole VACV, resulting in a response more similar to natural infection with a potentially broader epitope coverage compared to peptide-based assays.’

C48: line 339, "lower response" compared to what?

R48: This is in comparison to the CD8 mega pool. This has been added **on line 368**.

C49: line 346, "similar results seen in mpox" - this statement is entirely unclear

R49: Thanks for picking this up. Now we have changed the text to ‘It is therefore important to explore whether similar differences in T cell responses might arise between mpox infection and vaccination’, **on lines 375-376**.

C50: line 353, Such cells are also very prominent in CMV infection.

R50: We thank the reviewer for this point and have added ‘CMV infection’ **to line 384**.

C51: line 354, "These cells were shown to exhibit enhanced proliferation and differentiation potential in vivo." Compared to what? Please make a precise statement and provide a reference.

R51: We apologise for this. This is compared to other memory cell subsets and non-virus-specific cell subsets, which we have now added **on line 385** and added the appropriate reference (ref 45).

C52: line 359, "suggests that in convalescence, long-term MPXV-specific central memory T cells with high cytotoxicity may be generated." This statement is surprising since CCR7-CD45RA+ was by far the dominant phenotype, and this is not a central memory phenotype. There were few CCR7+CD45RA- T cells. Moreover, classical central memory T cells are not of high cytotoxicity. This seems to be an arbitrary and speculative statement that confuses different subsets.

R52: Thanks for picking this up. Now we changed “central memory T cells” to “memory T cells”, **on lines 391-392**.

C53: line 365, "With a more diverse TCR repertoire, T cells after vaccination may be able to contain variants more effectively." As the authors partially say themselves, and I explain further above, it is not clear which of these TCRs are even VCAV/MPOX-specific.

R53: please see our overall response above regarding clarification of the single cell analysis and purity of the sorted cells. In brief, we are confident about the purity of sorted cells therefore the specificity of the TCRs identified.

C54: line 393, "Our data showed that MPXV infection could elicit **longer-term** memory T cells with

greater effector function and migratory potential to the site of infection, whereas T cells induced by MVA-BN vaccination may contain variants more effectively and show broader cross-reactivity to other orthopoxviruses due to their diverse TCR repertoire." The present data do not substantiate such a statement. T cells were studied only at one time point after the second dose of vaccination (about 4 weeks), and only at one time point after infection per patient. Timing of the two groups was not matched. Thus speculative comparisons such as "longer-term" cannot be substantiated. The claim about broader cross-reactivity to other orthopoxviruses is not substantiated either. I have already commented on the TCR repertoire.

R54: We agree this comment and revised the sentence and removed the comment '...whereas T cells induced by MVA-BN vaccination may contain variants more effectively and show broader cross-reactivity to other orthopoxviruses due to their diverse TCR repertoire' from updated manuscript.

C55: line 407, "at a similar time point post antigen encounter." The time point was not similar. It was 4 weeks after the second dose vs. a median of 105 days after symptom onset.

R55: We apologise for this mistake, this has now been removed from the discussion.

C56:line 416, "While there was no evident effect of this in our data" - since the question was not studied? Please provide information, as available, about the immune response observed in the two individuals with prior smallpox vaccination, compared to the two others. It would be easy to use different symbols for these two individuals in Fig. 4B, for example. You might also provide more information or evaluation of the MPOX convalescent cohort. It is of interest that each of the three MPOX convalescents older than fifty years had mild infection without fever (Table 1). Might some of these have received a smallpox vaccination?

R56: We have looked our raw data that these three convalescent candidates who is over 50 indeed show T cell response at the lower end in our cohort. This is also observed in the vaccinated cohort, the two candidates with previous smallpox vaccination (both received only one dose of MVA-BN vaccine) didn't elicit stronger T cell response after vaccination, compared to other donors. These five donors over 50 are highlighted in yellow in the figure below. It is possible that previous smallpox vaccination may have contributed to their milder symptoms, which needs to be investigated with larger cohorts.

Methods:

C57-1: line 650, please add to the Methods part info about the incubation medium for overnight stimulation - that's important since FCS/FBS can stimulate strong background T cell responses. The cited reference (19, Mateus 2021) does not provide this info for the AIM assay.

R57-1: The assay used R10 medium in co-culture. This information has been updated in Method section on lines 715-716. In our AIMs assay, only participant Mpox011 showed very strong background T cell response, which was excluded from the analysis.

C57-2: Please indicate as well whether the cryoconservation medium contained FCS.

R57-2: The cryopreservation medium was 90% FCS + 10% DMSO. This has been updated in Method section on line 712

C58: line 653, "BD human FC block (Biolegend, 422302)" seems contradictory

R58: Thanks for picking this up. It has now been updated with 'Biolegend FC block (Biolegend, 422302)' on line 719.

C59: line 658, "A list of antibodies used in this panel can be found in table S2" - that's Table S3, and it only lists target antigens, it does not indicate fluorochromes and combinations of antibodies used in panels. Please provide the FACS panel(s) and combinations of antibodies that were actually used in this assay, with fluorochromes and clone identification.

R59: Thanks for the comments. We have now included Table S5 which lists the antibodies used for AIMs assay. The details of antibodies used in this study, including fluorochromes, clone identification, lot number, can be found in the NP-reporting summary as required by the journal.

C60: line 663, the stimulation index (SI) mentioned, is this the ratio of signal to background?

R60: Yes, as noted in reference 13 by Grifoni, A. et al. The text has been updated accordingly along with the reference on line 729.

C61-1: provide info Table S1, NA means information not available or not applicable?

R61: NA means "Not Available". This has been added to the Table S1 footnote.

C61-2: Is there a chance that some of the NA subjects had a smallpox vaccine but that is unknown?

R61-2: These participants with an NA for smallpox vaccine most likely didn't receive anything because after 1980 it was no longer routine for smallpox vaccination to be carried out.

C62: Fig. S3, only limited gating information is shown

R62: For these three flow cytometry assays, the example FACS plots including the gating information with specific markers after CD3/CD4/CD8 gating has been shown in the main figures, therefore they are not included in the Fig. S3. But this figure has been updated with our tetramer gating for single-cell sorting.

C63: line 667f "T cells lines", "T cell line" -> T cell lines

R63: Thank you for pointing this out, this has now been corrected

C64: line 669, what is H10?

R64: We apologise for not defining this, H10 is RPMI media + 10% human serum (Sigma) + pen/strep + 1% L-glutamine. This has been added in the text on **line 735**.

C65-1: Table S3, why are some epitope sequences printed in light grey font?

R65-1: Thanks for picking this up, those sequences in light grey are the peptides failed in production **and then have been removed from the table**.

C65-2: The footnote says "*bold and underlined residues indicate substitutions from the VACV genome to the MPXV genome" - please state explicitly that it is the MPXV sequence that is displayed.

R65-2: Thanks for the comment, the label for 4th column of table has been updated with "Sequence in MPXV"

C65-3: For those peptides with divergent sequence, please provide the VACV sequence as well (for easier reference), since some peptides are heavily polymorphic among the two viruses.

R65-3: A column was added to the table with "Sequence in VACV".

Reviewer #2 (Remarks to the Author):

Chen et al. investigated in their manuscript 'T cell memory response to MPXV infection exhibits greater effector function and migratory potential compared to MVA-BN vaccination' T cell responses in mpox convalescence and MVA-BN vaccinated individuals. The authors found that magnitude of mpox-specific T cells are comparable between convalescent and MVA-BN vaccinated individuals, but that the T cells in convalescent patients are phenotypically different that from vaccinated individuals, as for example, have a more cytotoxic and migratory profile. These findings are in line with reports from SARS-CoV convalescent and vaccinated individuals.

Due to the recent outbreaks of mpox, this study is of high importance for the scientific community. However, some major points need to be addressed to clarify parts of this manuscript and make it more understandable.

We thank the reviewer for their positive feedback and their overall constructive criticism of our study, especially the reviewer's highlight of the importance of this and similar studies to the scientific community given the recent outbreaks of mpox. We have gone through the manuscript and addressed all the comments, hopefully making our study and presentation of the results more understandable. We have further addressed the individual comments below:

1. Cohorts:

C1-1: The authors have described the cohorts of n=13 convalescent mpox individuals (Mpx001-13) well, including basic information for the healthy cohort (HC). Regarding the vaccination cohort, the

basic characteristics of this cohort (supplementary material) are not connected with the main text (starting line 105). The authors need to link this information.

R1-1: We thank the reviewer for pointing this out. We have now linked supplementary table 1 to the text on **lines 112-117** and have added the following describing the basics of the vaccinated cohort:

‘A further 20 male individuals aged between 25-72 were recruited following vaccination with the MVA-BN vaccine in Oxford¹¹. All 20 individuals recruited were HIV negative and were naïve to MPXV infection; however, two of these individuals had a prior smallpox vaccination pre-1980 and all of these were seropositive at baseline. One individual had a history of type II diabetes. Further information on the vaccinated cohort can be found at Drennan et al.¹¹, with detailed description about this cohort and time of sample collection after vaccination dose can be found in Supplementary Table 2.’

C1-2: Neither the methods section nor the results section of the manuscript mentions the characteristics of the SARS-CoV-2 cohort used and how the data were generated (Figure 6). The authors need to clarify where this data comes from. If this data comes from a publicly available dataset, authors need to clearly state this so that the reader can understand their workflow. This is especially important considering that they have integrated this data into the mpox data (figure 6). E.g. which SARS-specific Tetramer was used for cell isolation? Are the author sure that this potentially publicly available dataset was generated the same way as the authors analyzed their scRNAseq data?

R1-2: We apologise for not making it clear which SARS-CoV-2-specific cells were used in this analysis. The paper from which the cells were used is referenced (Peng at al., Nature Immunology 2022, ref 35) and was data was produced by our own group. We have now included the following description of the cells: ‘Briefly, 912 NP₁₀₅₋₁₁₃-B*07:02-specific T cells were isolated from 10 individuals, 1-3 months after initial infection and scRNAseq carried out to investigate gene expression and TCR repertoire differences.’, **on lines 300-302.**

C1-3: In addition, there is no consistency in the manuscript in naming the cohorts in the figures. Sometimes convalescent and vaccinated and sometimes sometimes MPXV and MVA-BN was used. The authors need to streamline this.

R1-3: We apologise for the inconsistencies in the naming of the cohorts. We have now changed the naming so the two cohorts are either ‘Convalescent’ or ‘Vaccinated’.

C2. The authors need to streamline the results part, as in some parts they even included detailed methods sections (e.g. line 112-114) and in some parts they barely describe how they came to their results.

R2: We appreciate the comments from this reviewer and have revised our manuscript and results section, balancing the information in the results and methods regarding the relative classical/commonly used methods and the methods that are not commonly used which may need the further clarification in the result section (i.e. cell analysis on rare population).

C3. The authors need to explain in more detail what the FEC peptide mix contains.

R3: FEC peptide mix was supplied by AIDS reagents and contains 32 published epitopes recognized by CD8 positive T cells and presented by 12 class I HLA-A and HLA-B types. FEC pool are often used as

positive controls for ELISPOT assay in addition to PHA stimulation. This has been updated in **the methods on lines 677-679.**

C4. In line 110-130, the authors describe their results from the ELISPOT using VACV infected PBMCs to investigate the MPXV-specific response. Followed by an AIMs assay. It is not clear why the authors performed this AIM experiment. Wouldn't a flow cytometry analysis with a read-out of IFN γ + CD4 or IFN γ + CD8 T cells be more specific? The authors need to clarify this.

R4: Thanks for this comment. ELISPOT using VACV infected PBMCs was to investigate the overall MPXV-specific response. In order to understand the proportion of CD4 and CD8 responses, there are two well-accepted assays (ICS and AIMs assay) in the field to evaluate CD4 and CD8 antigen specific T cell responses. Like most of in-vitro assays, there are always limitations for different assays as this reviewer pointed out. It wasn't an easy decision for us, ideally we would like to do both, however, due the limitation of samples that available to us, our own experiences in dealing with limited sample availability and considering the assay sensitivity vs specificity (<https://doi.org/10.1038/s41590-020-0782-6>; <https://doi.org/10.1016/j.cell.2020.05.015>), we decided to use AIMS assay over IFN γ + ICS for this study. However, we agree, IFN γ + is likely more specific but from our own experiences maybe less sensitive, in particular to weaker stimulations, a comprehensive comparison is needed and merits further investigation.

C5. In figure 1, the authors need to check whether the results presented in figure 1e and f are correct. It seems they do not match. In figure 1E they show two individuals without CD8 response, while in figure 1f only one individual (Mpox008) is shown.

R5: We apologise for the confusion here. Although our cohort had 13 participants, 1 (Mpox009) did not have enough sample remaining for any further experiments. Furthermore, Mpox011 showed high background in the AIMS assay (see FACS plots below) and as a result data (0% of AIMs+CD8⁺T cells and 0% of AIMs+ CD4⁺ T cells) was therefore should excluded from the analysis, resulting in 11 samples for the AIMs assay shown in Fig 1E and Fig. 1F. This has been added to the text **on lines 131-141 and in** the figure legend for figure 1.

C6. For the experiment shown in figure 2, for example, it is not clear how the data shown in 2b was generated for CD8 early antigen peptide pool. Are both mega-pools combined as one stimulus (A2 early plus non-A2 early) or are the data are generated by separately stimulating PBMC with one of the pools and then adding them together? In other words, did the authors stimulate the PBMCs with the peptide pools separately or together (71 or 90 peptide containing pool or 161 peptide containing pool)?

R6: We apologized for the confusion here. The PBMCs were stimulated with 6 pools listed in Fig2A, separately. These 6 pools include two CD4 pools (CD4 early pool and intermediate/late pool) and four CD8 pools (A2 early pool, A2 intermediate/late pool, Non-A2 early pool, and Non-A2 intermediate/late pool). The data in Fig 2B are generated by adding the response from each of pool together. For example, response of CD8 early combines the response from both A2 early pool and non-A2 early pool.

C7. Are the data shown in figure 2B from stimulated sorted CD4 and CD8 T cells? Or are the authors assuming that they are CD4 or CD8 responses based on the predicted epitopes used? This could be misleading.

R7: Thanks for the comment. Data shown in Figure 2B were not from sorted CD4 and CD8 T cells. Now we changed “CD4/CD8 T cell response” to “T cell response to CD4/CD8 epitope pools”

C8. In figure and figure legend 2, the authors need to indicate what the x-axis means in figure 2c, and what the red lines mean in figure 2d. It would also be helpful if the authors would emphasize that the convalescent cohort was used to generate the data.

R8: We thank the reviewer for this comment and have amended appropriately. For figure 2C, the x-axis is the MPXV protein that were the overlapping peptides used was from. In figure 2d, the red lines are samples which have a higher response to the mega pool compared to the VACV-infection assay. And all the data generated in this figure is from convalescent individuals. The axis in figures 2C and 2D have been updated

We have also changed the title for figure 2 to be: ‘Fig. 2. Memory T cell responses in MPXV-convalescent individuals against epitope mega pools and early antigens in mpox convalescence.’ To highlight the use of convalescent samples and added this to the text on lines 159.

C9. In line 153-155: The authors need to report the mean + SEM or SD values before the p-value. In addition, this sentence is linked to figure 2b, where 7 (CD8) to 3 (CD4) responses were compared,

which is borderline for statistical analysis. The authors should also check if the reported n=13 for figure 2b is correct, since less than 13 dots are shown in the graph.

R9: 12 samples for both CD4 and CD8 mega pools were used here as one sample was not used in downstream analysis. There are 12 points and lines for all in this figure, but a lot of these are at 0 and so overlapping in the figure and so cannot be seen. The figure legend for figure 2b now has been updated: '(B) T cell responses against MPXV CD4⁺/CD8⁺ early proteins (red) or intermediate/late proteins (blue), N=12,

C10. The authors should indicate that the convalescent cohort, if correct, is used from line 177.

R10: We thank the reviewer for pointing this out. We have added the fact that the ELISpot assays in this section were carried out 'in the convalescent individuals' on **lines 159, 196 and 199.**

C11. The authors must indicate in figure legend 3 how many HLA-A2 individuals were used for this study, please.

R11: We apologise for not including this information in the figure legend. We have now added 'in 6 HLA-A*02:01⁺ individuals' to the legend of figure 3.

C12. If the identified epitopes (line 191-194) have not been previously described, the authors should highlight this in the abstract.

R12: Thanks for pointing this out. We have now updated the on line 204 and in the abstract.

C13. Figure 3B: it is not clear whether the 'magnitude' is a result of pooled responses from all investigated individuals. Please clarify.

R13: In figure 3B, magnitude corresponds to the average number of SFU/2500 T cells for cells from 4 individuals responded to with the appropriate peptide. Corresponding text has now been updated.

C14. Figure 3B, if n=6 donors are investigated, I do not understand who a frequency of 25% responders can appear. Or are only 4 individuals investigated?

R14: We apologise for the confusion. Although there are 6 HLA-A*02:01⁺ donors in our cohort, we were only able to generate short-term T cell lines from 4 donors for the screening. This has been added in the main **text on lines 199** and in the figure legend.

C15. Figure 3C: Why did the authors use double staining with PE and APC labeled tetramers of the same peptide?

R15: For this part, all 7 peptide-MHC complex was produced by UV-mediated MHC peptide exchange. To increase the specificity and sensitivity, we used dual labelled tetramers for the screening, as noted in reference by Anderson, R.S. et al 2012 Nat Protoc 2012 Vol. 7 Issue 5 Pages 891-902: Parallel detection of antigen-specific T cell responses by combinatorial encoding of MHC multimers We have now updated the figure to show CD8 vs tetramer.

C16. Figure 3D: The response to SLS in the VACV-stimulated A2-positive individual shows a nice memory recall response, however, for IL2 it looks like an unwanted extremely dominate TNFa response after restimulation. Do the authors have an explanation for this?

R16: Thanks for the comment. The possible explanation is TNFa production is more sensitive to antigen stimulation than IFN, the SLS-specific T cells may have higher functional avidity or higher antigen load on infected cells than IL2-specific T cells. Please also see the detailed responses to reviewer 1 (R22-1) regarding cytokine production profiles (TNF vs IFN). We have now added a sentence to highlight this observation which merits further investigation on lines 218-220.

C17. In figure 4 the authors need to show the frequency of tetramer+ mpxv-specific T cells for all investigated individuals.

R17: Thanks for the comment. The frequency of Tetramer+ MPXV-specific T cells for all the investigated individuals has been added to Figure 4d.

C18. Figure 5: The labeling of the clusters are not readable in A. For D, the mean is not visible in the violin plots. Please correct this.

R18: We have added boxes around the labelling on the UMAP plot which should hopefully make the cluster names more visible. For the violin plots in figure 5D, it is generally not standard to include the median/boxplot in the density plot of the violin plot for the expression of genes in standard scRNAseq datasets. Due to the generally high proportion of cells with 0 expression for a lot of genes, the median for a lot of genes will be 0 and the boxplot will get obscured by the points due to the large number of points (cells) being plotted. For this reason, we have plotted just the violin and the points for the expression of genes (e.g. Fig 5D) and plotted boxplots and points for module scores (e.g. Fig 6B).

C19. The authors need to rearrange the order of the graphs in figure 6b according to the text.

R19: Figure 6B has now been re-arranged to match the order the results are discussed in the text

C20. It is not traceable how the authors come to '93% of individuals with...' in line 324. The authors need to explain this.

R20: Thanks for the comment, 93% was calculated by 12 out of 13. Now the text has been updated with '93% of individuals (12 out of 13) with' on line 352.

C21. In line 325, the authors mentioned that an ELISPOT to detect mpox-specific T cell responses might be interesting for diagnosis. This statement seems to be a bit ambitious. This assay is time consuming and does not indicate the clinician if the patient is acutely infected or convalescent. The authors need to down-tune this statement or need to explain this in more detail.

R21: Agree and we have now deleted the sentence.

C22. In general, the discussion needs to be strengthened, since some parts are hardly discussed whereas others are too long. Especially, the section (line 332-343) needs to be rewritten. It is not clear what the authors want to state here, especially in the light that they also performed many data with peptide pools.

R22: We have now revised our discussion based on all reviewer's comments and concerns., including simplifying the discussion mentioned here (line 360-372).

C23. Line 355: The term ex in exKLRG1 is not introduced. Please do so.

R23: exKLRG1 is a terminology used by Herndler-Brandstetter et al. (ref 35) to identify a novel population of T cells. These cells are KLRG1+ cells that receive intermediate amounts of activating signals, resulting in a downregulation of KLRG1 during the contraction phase in a Bach2-dependent manner, and differentiate into all memory T cell lineages. These exKLRG1 memory cells retain high cytotoxic and proliferative capacity distinct from other populations. This has now been defined on lines 386-389

C24. The manuscript would improve by confirming some of the transcriptome data by proteome analysis (e.g. flow cytometry).

R24: We thank this very sensible suggestion and have now performed validation experiments (flow cytometry) and updated figure 6 and the corresponding text in the result (lines 335-3410) and discussion section (lines 400-405) accordingly

C25. Regarding the described migration profile of the mpox-specific T cells, analysis from skin lesions might support the findings.

R24: We thank the reviewer for this comment and agree that it would be very interesting to investigate MPXV-specific T cells in the skin lesions alongside with circulating T cells, especially the expression of migratory molecules to validate hypotheses generated from the data presented in this study. Unfortunately, samples from skin lesions were not collected for this study which requires highest level safety facility for sample handling. We have added this as a limitation in the discussion section and as an avenue for further investigation on lines 447-453:

‘Moreover, in this study, we present the characteristics of MPXV-specific T cells in the peripheral blood, it would be crucial to investigate the MPXV-specific T cells in the infectious skin lesions alongside circulating T cell responses to fully understand the role of T cells in controlling of MPXV infection. Given our findings suggesting increased migratory potential of ILD-specific T cells to the skin, it would be valuable to examine the expression of these migratory markers on T cells in the skin lesions which were not available for this cohort but provides an interesting avenue for further investigation.’

C26. Figure Legends: The authors need to check their figure legends. In general, often no dots or comma are included in the figure legends, makes it hard to read. In addition, the authors need to report the number of individuals investigated in each graph as well as the investigated number of cells (e.g. 6c) and they need to make sure that the numbers mentioned in the figure legend are in to the individual dots shown in the graphs. In this regard, the number of dots displayed e.g. in figure 1c for Mpox do not match with the indicated number in the figure legend (n=13).

R26: Thanks for the comment. We have carefully checked all the figure legends and corrected all the errors.

C27. Figure: Fine tuning of the figures would dramatically improve the manuscript. This includes bigger dots in the graphs like fig 1c, more distance between different graphs and labelling like fig4c and d as well as in figure 3c.

R27: Thanks for the comment. We have created new figures with bigger dots and more distance between different graphs and labelling as suggested

C28. Supplementary tables: The formatting of the uploaded supplementary tables are disrupted.

R28: Unfortunately, the journal requires all tables that are longer than a side of A4 to be submitted as Excel spreadsheets. In the generation of the PDF for the reviewers, these larger tables tend to become distorted as they do not fit well on A4 pages. However, we have checked the tables and the

submitted tables and the Excel spreadsheets appear as expected and we believe upon publication, these Excel spreadsheets should be properly formatted.

Reviewer #3 (Remarks to the Author):

The manuscript by Chen et al described a study of convalescent monkeypox virus (MPXV)-infected and MVA-BN (a new smallpox vaccine) vaccinated individuals. The authors found substantial CD8+ and CD4+ T cell responses that shared similar differentiation and activation phenotypes. scRNAseq analysis of convalescent individuals seems to have a somewhat increased cytotoxicity and migratory potential. Given the recent outbreaks and relatively little knowledge on human T cell immunity to MPXV infection the current work provides additional insight. Overall the work has been performed well but validation of the main conclusions based on only the scRNAseq data are required.

We thank the reviewer for their positive feedback and constructive critique of our study, especially the reviewer's highlight of the importance of studying T cell immunity to MPXV infection given the newly recent outbreaks around. As suggested by the reviewer, we have carried out further *ex vivo* flow cytometry analysis to validate the scRNAseq data, with the results of this now included in the updated figure 6. We have further addressed the individual comments below:

Comments and suggestions.

C1: The main issue with this manuscript is the claim that the functional profile (with respect to cytotoxicity and migration) of MPXV-specific memory T cells induced by natural infection is claimed to be "better" than the vaccine (Figure 6). This conclusion is based on a limited set of scRNAseq data (only 161 cells of the vaccination group) and no validation at the protein level or by functional assays (e.g. by flow cytometry, cytotoxic and migration assays). In fact, the assays performed as displayed in Figure 4, indicated no differences in cytokine profiles and if anything a only a different phenotype with respect to KLRG1 but not to other markers. As such the conclusions are currently based on scarce data, and may therefore not reflect actual differences or resemblance.

R1: We agree with the concerns. And have now tuned down the conclusion regarding the differences between natural infection and Vaccination, which merit for future study. We have also performed flow cytometry analysis to validate the single cell analysis, generating new Figure 6d and discussed these results on **lines 335-340 and 400-405**.

C2: Maybe needless to emphasize but whether (functional) differences between T cells elicited by natural infection or vaccination are significant or insignificant the results are of interest and contribute to improved understanding.

R2: We appreciate and agree with this reviewer's comment, and we have now tuned down the conclusion regarding the differences between T cells elicited by natural infection and Vaccination in the discussion and abstract.

C3: -Further understanding could be from determining TCF-1 levels as described by Adamo et al,

2023. The TCF-1 levels could give insight into longevity/stemness of the viral-specific T cell population.

R3: We investigated it and found no differences between MPXV and vaccinated in regard to the level of TCF7 expression as shown below.

C4: Minor: Abstract: explain the abbreviations MPXV and MVA-BN in the abstract for clarity and instant understanding.

R4: We apologise for not defining these in the abstract. Both MPXV and MVA-BN have now been defined in the abstract.

1. Peng, Y. *et al.* An immunodominant NP105-113-B*07:02 cytotoxic T cell response controls viral replication and is associated with less severe COVID-19 disease. *Nat Immunol* **23**, 50-61 (2022).
2. Schulien, I. *et al.* Characterization of pre-existing and induced SARS-CoV-2-specific CD8(+) T cells. *Nat Med* **27**, 78-85 (2021).
3. Peng, Y. *et al.* Broad and strong memory CD4(+) and CD8(+) T cells induced by SARS-CoV-2 in UK convalescent individuals following COVID-19. *Nat Immunol* **21**, 1336-1345 (2020).
4. Bertoni, A., Alabiso, O., Galetto, A.S. & Baldanzi, G. Integrins in T Cell Physiology. *Int J Mol Sci* **19** (2018).
5. Duangchinda, T. *et al.* Immunodominant T-cell responses to dengue virus NS3 are associated with DHF. *Proc Natl Acad Sci U S A* **107**, 16922-16927 (2010).
6. Shahbaz, S. *et al.* The Quality of SARS-CoV-2-Specific T Cell Functions Differs in Patients with Mild/Moderate versus Severe Disease, and T Cells Expressing Coinhibitory Receptors Are Highly Activated. *J Immunol* **207**, 1099-1111 (2021).
7. Takahama, S. *et al.* Hepatitis B surface antigen reduction is associated with hepatitis B core-specific CD8 T cell quality. *Frontiers in Immunology* **14** (2023).
8. Duvall, M.G. *et al.* Polyfunctional T cell responses are a hallmark of HIV-2 infection. *Eur J Immunol* **38**, 350-363 (2008).
9. Abd Hamid, M. *et al.* Defective Interferon Gamma Production by Tumor-Specific CD8 T Cells Is Associated With 5'Methylcytosine-Guanine Hypermethylation of Interferon Gamma Promoter. *Frontiers in Immunology* **11** (2020).
10. Yin, Z. *et al.* Evaluation of T cell responses to naturally processed variant SARS-CoV-2 spike antigens in individuals following infection or vaccination. *Cell Rep* **42**, 112470 (2023).
11. Drennan, P.G. *et al.* Immunogenicity of MVA-BN Vaccine Deployed as Mpox Prophylaxis: A Prospective Cohort Study and Analysis of Transcriptomic Predictors of Response. *Preprints with The Lancet* (2024).

Point-to-point responses to reviewer's comments:

Reviewer #1 (Remarks to the Author):

The authors have undertaken a thorough and thoughtful revision of their manuscript, and have clarified many points.

Thank you

C1: R10, I am not sure if the sample should really be excluded due to high background. If background is as high or higher as the signal in the presence of antigen, it might be better to include this in the dataset as a zero result. If this was a single and strong outlier in terms of background, exclusion might be justified though.

R1: This participant is the only sample with a strong background in the AIMs assay. In agreement with this reviewer, it is hard to interpret at this stage. Therefore, we believe it is best to exclude the data rather to include as zero which could be misleading.

C2: R11-1, I thank the authors for their response and additional information. Sorry I didn't manage to make it clear what my question was. My point was that if a low absolute number of specific T cells was obtained, then the ratio of CD4 to CD8 is less certain. So the different CD4/CD8 ratios may have a very different degree of certainty that is not obvious from this kind of graphical representation. I wanted to suggest that the absolute number of T cells is indicated for each sample in Fig. 1F, in order to make it easier to judge the certainty/precision of each of these CD4/CD8 ratios.

R2; Thanks for the comment. We have now updated Figure 1E and 1F with the percentage of **CD3+** T cells which are CD4+AIMs+ or CD8+AIMs+. So, the proportion of CD4+/CD8+ T cell response in the figure 1F was conculcated using these values.

We also included the absolute cell number here, but these numbers are recorded without deduction of negative controls. The CD4/CD8 proportion is in general consistent from both calculations, except the ones highlighted in yellow. As such, we classified the MPX005 CD4+ and the MPX008 CD8+ response as 0 (highlighted in yellow in the table below) because they didn't pass the thresholds determined for a positive response as described in the methods.

We didn't use the absolute number of cells in this analysis because it would not be appropriate to subtract the background number of cells from the actual sample, given the differing initial cell numbers used/acquired in the actual sample and negative controls. Therefore, using the % of CD3+ cells allows us to subtract the negative control and to correct for background staining in the assay.

SampleID	Absolute number of cells		CD8+CD69+CD137+% minus Negative control (of CD3 population)	CD4+CD40L+OX40+% minus Negative control (of CD3 population)
	CD8+CD69+ CD137+ Count	CD4+OX40+ CD40L+ count		
MPX-001	321	248	0.179	0.134
MPX-002	110	205	0.059	0.13139
MPX-003	303	228	0.138	0.118
MPX-004	160	44	0.092	0.02655
MPX-005	131	79	0.08217	0
MPX-006	114	44	0.057	0.018
MPX-007	503	626	0.259	0.25
MPX-008	4	26	0	0.364
MPX-010	365	473	0.104	0.125
MPX-012	747	761	0.4	0.29
MPX-013	550	1169	0.388	0.917

C3: R16-1, In their reply here the authors now provide the important information that 7/12 donors recognized at least one of the antigens tested here. Conversely, that means that 5/12 donors did not respond to any of the antigens. This seems important information and should be included in the text. It will help readers (such as me) understand Fig. 2B correctly.

R3: Thanks for the comments. The sentence "7/12 donors recognized at least one of the antigens tested here" has been added on Line 183-184

C4: By "complementary" I meant that some donors not responding to E antigens might respond to int./L antigens instead or vice versa. The authors' response makes it clear that this is not the case. Since 7 donors had a CD8 response to E antigens as already stated in the text, donors responding to int/L were a subset of donors responding to E.

R4: Thanks.

C5: R17, I thank the authors for these details. Sorry I was nitpicking so much about this HLA frequency issue. But it is still not correct to say that HLA-A*02:01 is the most frequent HLA allele in Caucasians, HLA-DPB1*04:01 is more frequent. However, it would be correct to say that HLA-A*02:01 is the most frequent HLA class I allele in Caucasians.

R5: Thanks for the comment, this has been updated on line 194, we have now changed the text to: "A*02 is very common class I allele in different populations around the world" - based on this reviewer's concerns and the Editor's suggestion".

C6: R18, the revised text 194ff is a bit unclear, "T cell responses to 71 A*02-restricted early epitope pool and 68 intermediate/late epitope pool", I think this means "T cell responses to a pool of 71 A*02-restricted early epitopes and a pool of 68 A*02-restricted intermediate/late epitopes"?

R6: This has been updated as requested on line 195-196

C7: R19, I understand that the authors wished to test all epitope candidates reported to be restricted through A*02:01 or "A2", and that makes sense to me. I'm not sure though whether they included epitopes that were known to be restricted through another A2 allele such as A*02:03 but not through A*02:01. This is just an aside.

R7: We included epitopes restricted by A*02:01, as well as those restricted by A*02 when A0*2:01 was not explicitly specified.

C8: However, in line 674 they still write "We further divided the CD8 peptides into HLA-A*02:01 and non-HLA-A*02:01 peptide pools", which is inconsistent with their response. Please state the criterion for this "division" clearly here. Please check again if statements about A2 vs A*02:01 are consistent and error-free.

R8: Thanks for picking this up. We have checked and updated accordingly.

C9: R22 (critical) The reason for my skepticism is that I have repeatedly encountered such results (TNFa+ cells >> IFNg+ cells) in large and important datasets from partners, and in each case these observations were demonstrably due to experimental errors (and did not match a preponderance of results in the literature). In one case the problem was FACS compensation, in another it was problems with separation of fluorescence signals in multicolor ELISPOT. Therefore I am very wary of this specific issue.

I thank the authors for providing several relevant references that claim %TNFa+ > %IFNg+ for virus-specific T cells ex vivo. I have checked Shahbaz 2021. The IFNg staining in that paper (Fig. 2A) seems weak, smear-like, and substandard, even in the anti-CD3/28 control. Bad antibody, staining protocol, or compensation? Information about fluorochromes is not even provided in that paper. Those authors' job might have been a bit more difficult since some of their patients were acutely infected, severely ill and in ICU, which may lead to suboptimal T cell quality or generalized activation. In the present manuscript, however, patients were convalescent.

In the case of the present manuscript, I am worried by the unusual appearance of the populations in Fig 3D (TNF vs IFN), and that there is not a single IFNg+ TNFa- cell anywhere, although such cells are regularly observed in diverse viral infections. The one sample with significant numbers of IFNg+ cells shows only double-positives in some diagonal staining pattern. I am not convinced everything is OK with these FACS analyses and their compensation. Unfortunately they do not show a full set of, or any, cytokine-vs-cytokine stainings in their gating example in Fig S3B, they just show the higher-level gating. However, correct compensation is just as important as correct gating, and usually more difficult.

In the authors' earlier publication (Yin 2023), there are many diagonal distributions of cells in cytokine-vs-cytokine plots, which are especially clear in a combination of the same fluorochromes as here (PE-Cy7 vs APC, Fig. 3C in Yin 2023). Although in that earlier population the antibodies were IL-2-PE-Cy7 vs TNF-APC, the distributions are actually very similar to the ones the present manuscript for IFN-g-PE-Cy7 vs TNF-APC: the double-positive cells are entirely found in a diagonally-shaped population, and there are no PE-Cy7 single-positive cells.

Therefore I strongly suspect a problem with fluorescence compensation. The authors should try and see what happens if they modify their compensation settings. I understand this might be tedious. But it seems important to me that fundamental functions of T cells are not erroneously reported. Such errors seem to be arising more and more often in the literature.

Reviewer 2 agrees with me that this seems to be an issue.

R8: 1) We understand this reviewer's concerns; however, we are very confident in the ICS data compensation for the following reasons:

- a) The ICS panel used in this study is outlined in the table below. This panel has been routinely employed in our lab to evaluate the functionality of antigen-specific T cell clones and lines.

Data generated using this panel have also supported several publications, including Yin 2023 (specifically, the data presented in Figure 2B).

	Marker	fluorochrome
1	CD8	FITC
2	Live/dead	BV510
3	IL2	BV421
4	IFN γ	PE-Cy7
5	TNF α	APC
6	CD107a	PE
7	MIP1b	APC-H7

- b) The configuration of the Attune flow cytometer for PE-Cy7 (green laser) and APC (red laser) is detailed below. Due to the use of different lasers and filters, significant spillover between these channels is unlikely, which doesn't require substantial compensation for IFN γ -PE-Cy7 and TNF α -APC.

Y4 - PE-Cy7	561-780/60
R1 - APC	640-670/14

Here is the compensation matrix for data analysis

	BL1-A :: CD...	YL4-A :: IFN...	YL1-A :: CD...	RL1-A :: TNF...	RL3-A :: MIP...	VL2-A :: LD...	VL1-A :: IL2...
BL1-A :: CD8-FITC-A	100	0	0.0393	0.2585	0.0272	2.2489	1.1931
YL4-A :: IFN γ -PE-Cy7-A	0.2667	100	7.3672	0	15.4039	0.0559	0.053
YL1-A :: CD107a-R-PE-A	1.3322	0.1887	100	0	0.2	0.2589	0.1822
RL1-A :: TNF α -APC-A	0.9407	1.8423	0.1321	100	12.8041	0	0.3174
RL3-A :: MIP1b-APC-H7-A	0.0276	12.6133	0.3421	1.6448	100	0	0
VL2-A :: LD-Fixable Aqua-A	0.169	0	0.0831	0	0	100	8.2721
VL1-A :: IL2-BV421-A	0	0.1233	0.0282	0.079	0.8843	9.9626	100

- c) Peptide-loaded target cells (at a concentration of 2 μ g/mL) were included as controls alongside VACV-infected target cells (naturally processed and presented peptides presumably much lower ag load than 2 μ g/ml). All samples were acquired on the same machine on the same day. Below are the FACS plots displaying the gating of all the effector markers across the tested cell lines with peptide stimulation.

2) We would like to highlight that:

- Figure 3D is to evaluate the cytokine production by **in vitro expanded Cell lines specific to individual A2 epitope peptides** (not ex-vivo stimulation of PBMCs – presumably the large data set in most of the literature)
- The target cells used for this experiment were **Vaccinia virus-infected cells (naturally processed antigen)** rather than peptide-pulsed (**saturating concentration of exogenous** and commonly used in most of the literature for ICS).

It is beyond the scope of this paper to figure out the precise reasons why we observed more TNF than IFN in our dataset (which is likely to be complicated and multifaceted). This has been discussed in the result part of first round of revised manuscript on line 219-222 after presenting the data from Fig3E **“Some cell lines exhibited a strongly dominant TNF α response compared to the IFN γ response. Further investigation is needed to determine the factors contributing to this, including whether this is influenced by antigen load on infected cells, or the functional avidity of the T cells, or other factors that may be at play.”**

R23, I think the authors for clarifying their killing assay, which makes sense to me.

R9: Thank you

C10: R26, There is a difference between "28 days" (revised Fig. 4A) and "no less than 28 days" (revised Methods). The new preprint cited actually says it was between 26 and 40 days in the cohort. Please be so kind and report this correctly here for the subset of donors studied here.

R9: Thanks for picking this up. Now this error is corrected in the figure and in the methods on line 663.

C 11: R32-1 (critical) If authors insist to show and interpret these single-cell results from as few as 160 cells after vaccination, they should at least make it very clear how many donors were involved in these 160 cells, and how many cells were from each donor. I think it's unacceptable that this

information still seems not to be provided. My apologies if I overlooked anything. I also request an UMAP plot such as in Fig. 5A but color-coded by donor. At least there should be full transparency about the origin of these cells, and about the impact of their donor origin on subset distribution.

R11:

C12: R32-1, R37 (critical) In spite of their claim in their replies, authors have not provided proof that their tetramer-sorted cells are 100% pure. 100% purity is a strong claim, and it seems the authors have not added this explicit claim to their manuscript? Fig. S3 does not prove that. Indeed, the CD8+CD3+ gate contains both CD8-high and CD8-low-positive cells. The tetramer+ gate contains tetramer-high and tetramer-dim cells and even a few cells that are both tetramer-dim and CD8-dim. Such cells might be more likely to be something non-specific. A much stricter gate might have been used for sorting in this example. Some control staining that shows tetramer background in non-immune donors would be helpful. Tetramer staining sometimes does have significant background. Can original data be provided that substantiate 100% purity?

R12: We totally understand the concerns and apologise for claim of 100% pure. Due to the strict gating strategy, we believe most (if not 100%) were antigen specific. We indeed used the PBMCs from healthy donor as negative control, same gating was applied to the samples sorted at the same day. The gating shown in the Figures3 was applied to other samples (see below figures including examples of very high percentage and very low percentage for both MPOX infected individuals at top panel and Vaccinated at bottom panel). We agree that accurately gating tetramer-positive cells becomes particularly challenging when their frequency is low, increasing the likelihood of contamination, and we could not prove it is “100% pure”. therefore, we have addressed this in the limitations section of our discussion: “While the gating and sorting strategy strongly suggests that the majority of tetramer-sorted cells were antigen-specific, we acknowledge the potential for contamination by non-specific T cells”.

Comment 13: The frequency of specific cells is not indicated in this example in Fig. S3; it might be even more difficult to sort epitope-specific CD8 T cells to high purity in donors with very low frequency of tet+ cells (see new Fig. 4D). Proportions of tet+ cells were at 0.01% or lower in 4 of the 6 vaccinated donors in that figure. How high were these numbers? Can a log plot be provided, or the actual numbers?

R13: We agree that it may be difficult to obtain epitope-specific CD8+ T cells in donors with a low frequency of tetramer+ cells. However, we used consistent gating for all our sorting strategies as shown above. For ease of interpretation, we have included figure 4D below with the y-axis on a log scale. Although the frequency of tetramer positive cells is very low from vaccinated donors, we have used large number of input cells (up to 12-15 x10⁶ PBMCs, 3 vials of PBMCs) to sort the cell numbers we need.

C14: Considering Fig. 4D, we can't even exclude that group differences in single cell analysis between multimer-sorted single cells from convalescents and vaccinated donors are due to different degrees of contamination by non-specific T cells, because responses in some vaccinated donors are so much lower than in convalescents.

R14: Please see the response from R12 above. We have added “the lower cell number” and “possible contamination” in the limitation section in our discussion.

Other

Line 353 has a new claim, "Furthermore, we identified 7 HLA-A*02:01 restricted MPXV-specific epitopes where six of them are conserved between VACV and MPOX, one is specific for MPXV" - "we identified epitopes" is usually understood to mean that they were not known before. Please modify.

Those epitopes were initially identified from VACV, there were no report of those epitopes being detected or identified in MPXV virus infected individuals. We have now changed the text to “Furthermore, we reported 7 HLA-A*02:01 restricted MPXV-specific epitopes which have been reported as VACV epitopes, where six of them are conserved between VACV and MPOX, one is specific for MPXV”

Reviewer #2 (Remarks to the Author):

The authors have sufficiently answered most of my questions/comments.

Regarding the Sars-Cov2 data from Peng et al Nature Immunology 2022, I would ask the authors for a statement again.

C1: In the paper ‘Peng et al Nature Immunology 2022’, scRNAseq data from 4 patients are shown. Figure 2 ‘a, Gene sets scored based on single-cell gene expression from a SmartSeq2 RNA-seq dataset comprising two mild and two severe convalescent HLA-B7*07:02-positive patients with COVID-19 (n = 208 cells from mild cases, n = 140 cells from severe cases).’ The authors should comment on how they arrive at 10 individuals. Furthermore, 912 cells from 10 individuals means a mean of 90 cells per individual. Can the authors discuss the limitations and problems arising from these low cell numbers and the heterogeneity between individuals (batch effect)?

We apologise for the confusion about the SARS-CoV-2 dataset used here and for not making it clear. Yes, in the Peng et al Nature Immunology 2022 paper, data from 4 patients was used and analysed. Since the publication of that paper, we have further increased the dataset to 10 patients total, sequencing NP105-113-B*07:02-specific cells from a further 6 patients, resulting in a total of 912 cells from 10 patients. Although the average number of cells per patient in this dataset is relatively low, the cells investigated here are antigen-specific and targeting the SARS-CoV-2 epitope. These cells are used for comparison with MPXV epitope ILD-specific cells from another disease context, where the overall frequency in circulation is generally low, too. Furthermore, increasing the number of patients to 10 patients allows us to examine overall differences while accounting for patient heterogeneity. Although the number of cells per patient is relatively low, analysing a larger number of patients allows us to identify differences that are conserved despite individual variability.

We have now discussed the limitations and problems arising from these low cell numbers and the heterogeneity between individuals on lines 441-447.

‘The frequency of antigen-specific cells in the circulation is relatively low, and as such, we were only able to investigate transcriptional changes in a low number of ILD-specific and compare with a low number of NP105-113-specific T cells from Peng et al. Although we have increased the number of participants from which NP105-113-specific T cells were investigated to overcome the issues of patient heterogeneity, validation of these results is needed in a larger number of cells while reducing batch effects arising from different patients.’

C2: In addition, it would be good if the authors would insert the number of vaccines and convalescent individuals into the figure caption of Figure 5.

We have included the number of vaccinated and convalescent individuals in the figure legend title for figure 5.

Point-to-point responses to reviewer's comments:

Reviewer #1 (Remarks to the Author):

I thank the authors for their additional thoughtful and informative comments.

R8: I thank the authors for their patience in dealing with my inquiries and for providing additional explanations and examples of stainings with peptide-stimulated cells, which seem to me to demonstrate adequate compensation between TNF α -APC and IFN γ -PE-Cy7.

We thank the reviewer for this comment and are pleased we have managed to demonstrate adequate compensation of our flow cytometry experiments.

R11: Thank you for this graph, which suggests that the cells are roughly equally distributed in the UMAP space independent of their donor origin, and that similar numbers of cells were analyzed within each donor group. I think the graph is informative and should be included in the paper, not just in the rebuttal letter, maybe as a supplement. This would strengthen the message in spite of the relatively small number of cells. I was asking my question in behalf of future readers of the paper, not just for my own information.

We thank the reviewer for this, and have now added this figure of the UMAP with cells colours by donor as a supplemental figure, Supplementary figure 5a.

R13: Thank you for this graph. I think it should be shown in the paper, e.g. replacing Fig. 4D, not just in the rebuttal letter.

We thank the reviewer for this comment, and have now replaced figure 4d with the new figure, with the y-axis on a log scale.

Reviewer #2 (Remarks to the Author):

The authors answer all my questions satisfactorily.

We are pleased we have managed to satisfactorily address all the reviewer's questions.